# A SIRT7-dependent acetylation switch regulates early B cell differentiation and lineage commitment through Pax5

Andres Gamez-Garcia [1], Maria Espinosa-Alcantud[1,13], Alberto Bueno-Costa[2,13], Elisenda Alari-Pahissa [3], Anna Marazuela-Duque[1], Joshua K. Thackray [4], Chandni Ray[4], Clara Berenguer[5], Poonam Kumari[6], Joan Josep Bech [7], Thomas Braun [6], Alessandro Ianni [1,6], Jay A. Tischfield[4], Lourdes Serrano[4], Manel Esteller[2,8,9,10], Jose L. Sardina [5], Carolina De La Torre[7], Mikael Sigvardsson[11], Berta N. Vazquez [1,12] ✉ & Alejandro Vaquero [1] ✉

B lymphopoiesis is orchestrated by lineage-specific transcription factors. In B cell progenitors, lineage commitment is mediated by Pax5, which is commonly mutated in B cell acute lymphoblastic leukemia. Despite its essential role in immunity, the mechanisms regulating Pax5 function remain largely unknown. Here, we found that the NAD$^+$-dependent enzyme SIRT7 coordinates B cell development through deacetylation of Pax5 at K198, which promotes Pax5 protein stability and transcriptional activity. Neither Pax5$^{K198}$ deacetylated nor acetylated mimics rescued B cell differentiation in $Pax5^{-/-}$ pro-B cells, suggesting that B cell development requires Pax5 dynamic deacetylation. The Pax5$^{K198}$ deacetylation mimic restored lineage commitment in $Pax5^{-/-}$ pro-B cells and B cell differentiation in $Sirt7^{-/-}$ pro-B cells, suggesting the uncoupling of differentiation from lineage commitment. The SIRT7–Pax5 interplay was conserved in B cell acute lymphoblastic leukemia, where SIRT7 expression correlated with good prognosis. Our findings reveal a crucial mechanism for B lymphopoiesis and highlight the relevance of sirtuins in immune function.

Lineage-instructive transcription factors establish the gene regulatory networks that drive differentiation of hematopoietic progenitors toward all mature immune cell types. Among these, the transcription factor Pax5 regulates B cell identity[1]. Following specification, Pax5 orchestrates downstream differentiation and drives lineage commitment by promoting simultaneous activation of B cell-specific programs and repression of alternative-lineage genes[2-4]. Pax5 deficiency in mice results in a complete block in B cell differentiation at the pro-B cell stage, leading to the accumulation of uncommitted cells[5-7]. In human B cell acute lymphoblastic leukemia (B-ALL), which is the most common cancer in children, Pax5 is a haploinsufficient tumor suppressor, and the deletion of one *Pax5* allele in mice cooperates with oncogenic mutations and with heterozygosity of the transcription factor *Ebf1* to drive malignant transformation[8-11].

In B cells, Pax5 controls the global genome architecture[12] while simultaneously limiting pro-B cell proliferation through repression of *Myc*[13], restricting cellular metabolism[14] and facilitating loop extrusion of the whole immunoglobulin heavy chain (*Igh*) locus[15-17]. Pax5 collaborates with chromatin remodeling coactivator and co-repressor complexes and with other B lineage transcription factors to establish a self-reinforcing network that drives differentiation[18,19]. Furthermore, Pax5 dynamically binds to various genomic regions and regulates distinct target genes in developing and mature B cells[20]. Although the importance of Pax5 for B lymphopoiesis and human leukemia is well

established, the mechanisms that control Pax5 biology and dynamics are still unclear.

Sirtuins are a family of conserved NAD[+]-dependent deacetylases implicated in mammalian immunity[21] that regulate inflammation, T cell polarization and responses and hematopoietic stem cell exhaustion through different mechanisms, including the deacetylation of histone and nonhistone substrates[22]. Here, we show that SIRT7, a sirtuin with central roles in genome stability and stress response[23], controls Pax5 activity during B cell lymphopoiesis and human B-ALL. SIRT7 directly promotes B cell development and commitment by establishing an acetylation switch at a single Pax5 residue located within a putative intrinsically disordered region. Our results indicate that post-translational modifications of lineage-instructive transcription factors may be sufficient to disrupt lineage commitment and suggest key roles for sirtuins in the physiology and malignancy of hematopoietic progenitors.

## Results

### SIRT7 is required for normal B cell development

To understand the role of sirtuins in hematopoiesis, we analyzed the expression levels of sirtuins *Sirt1*, *Sirt2*, *Sirt6* and *Sirt7* in Lin[−] (Ter119[−]CD11b[−]B220[−]Gr-1[−]) progenitors using publicly available single-cell RNA-sequencing (scRNA-seq) data from mouse bone marrow (BM) cells (Broad Institute, study SCP978). Among them, *Sirt7* displayed the highest expression, particularly in the cluster annotated as B cells (Fig. 1a and Extended Data Fig. 1a). In publicly available data from human BM cells (Broad Institute, study SCP101), *SIRT7* was highly expressed in *Cd79a*[+]*Ighm*[+] B lineage cells, *Nkg7*[+]*Klrf1*[+] natural killer (NK) cells and *Trbc2*[+]*Cd3d*[+] T cells (Extended Data Fig. 1b,c). SIRT7 protein was highly expressed in BM CD19[+] B cells purified by magnetic-activated cell separation compared to in the CD19[−] fraction (Fig. 1b). In mouse B cell progenitors, *Sirt7* expression peaked at the committed B220[+]CD19[+]IgM[−]CD43[−] pre-B cell stage (Extended Data Fig. 1d,e). *Sirt2* expression increased to a lesser extent in the same stage, whereas the expression of other sirtuin transcripts did not change during B cell differentiation (Extended Data Fig. 1d,e). Intracellular flow cytometry of mouse BM B220[+] B cell progenitor cells showed a gradual upregulation of SIRT7 expression from B220[+]CD19[−] pre-pro-B cells to B220[+]CD19[+]IgM[−]CD43[−] pre-B cells and was maintained in B220[+]CD19[+]IgM[+] immature and mature B220[hi]CD19[+] B cells (Fig. 1c,d and Extended Data Fig. 1f). These observations indicated that SIRT7 expression was upregulated during B lymphopoiesis.

To determine the role of SIRT7 in B cell development, we measured the number of B220[+]CD19[+] B cells in the BM of two reported models of *Sirt7*[−/−] mice, *Sirt7*[Δ4–10] 129Sv and *Sirt7*[Δ4–9] C57BL/6 mice (Methods)[24,25]. In both strains, *Sirt7*[−/−] mice displayed a significant reduction in the number of B220[+]CD19[+] B cells compared to wild-type littermates (used throughout unless otherwise specified; Fig. 1e and Extended Data Fig. 2a). We also found decreased numbers of B220[+]CD19[+]IgM[−]CD43[−] pre-B cells, B220[+]CD19[+]IgM[+] immature B cells and B220[hi]CD19[+] recirculating mature B cells (Fig. 1f,g and Extended Data Fig. 2b), indicative of an impaired pro-B-to-pre-B cell transition. We used the *Sirt7*[Δ4–10] 129Sv model, hereafter referred to as *Sirt7*[−/−], throughout for all experiments. The spleens of *Sirt7*[−/−] mice exhibited a contraction of the CD19[+] cell compartment (Fig. 1h,i) and compromised structural architecture, with fewer and less-organized follicles than spleens in the wild-type mice (Fig. 1j). There was a significant reduction in the number of B220[+]CD19[+] CD21[+]CD23[+]CD93[+] transitional, B220[+]CD19[+]CD21[hi]CD23[−] marginal zone and B220[+]CD19[+]CD21[+]CD23[+]CD93[−] follicular B cells in *Sirt7*[−/−] mice compared to in wild-type mice (Fig. 1k and Extended Data Fig. 2c). The number of B220[+]CD19[+]IgM[+]Gl7[+]Fas[+] germinal center B cells in *Sirt7*[−/−] mice was normal, whereas the number of CD19[+]CD38[+]CD138[−]Gl7[−] memory B cells, B220[lo]CD138[+] plasma cells and B220[+] class-switched IgG1[+] B cells was reduced compared to wild-type mice (Fig. 1k and Extended Data Fig. 2d–g), suggesting a defective germinal center response in *Sirt7*[−/−] mice. The levels of hen egg lysozyme (HEL)-specific IgM, IgG1 and IgG3 antibodies in serum at day 14 after immunization with HEL antigen coupled to NPP hapten (NP–HEL) were significantly reduced in *Sirt7*[−/−] mice compared to wild-type mice (Extended Data Fig. 2h), suggesting that SIRT7 deficiency leads to impaired B cell development and immunity.

To test the contribution of the BM stroma in *Sirt7*[−/−] mice, purified Lin[−]B220[+]CD19[+]IgM[−] pro-B cells from the BM of CD45.2 wild-type and *Sirt7*[−/−] mice were expanded in the presence of OP9 stromal cells, interleukin-7 (IL-7), stem cell factor (SCF) and FLT3-L for 4 days and injected into sublethally irradiated CD45.1/CD45.2 mice to generate separate wild-type and *Sirt7*[−/−] BM chimeras. Wild-type pro-B cells repopulated the splenic CD45.1[−]CD45.2[+]CD19[+] B cell compartment of recipient mice, whereas *Sirt7*[−/−] pro-B cells did not (Fig. 1l and Extended Data Fig. 2i), indicating that SIRT7 had a B cell-autonomous role. Retroviral expression of wild-type SIRT7 into *Sirt7*[−/−] pro-B cells injected into sublethally irradiated CD45.1/CD45.2 mice reversed the B cell differentiation block, whereas expression of a catalytically inactive SIRT7[H187Y] mutant did not (Fig. 1m). These findings indicate that SIRT7 promotes pro-B-to-pre-B cell transition through a mechanism dependent on its deacetylase activity.

### SIRT7 promotes B cell development independently of V(D)J

SIRT7 facilitates DNA damage repair by controlling nonhomologous end joining[24]. *Sirt7*[−/−] BM pre-B cells, but not *Sirt7*[−/−] pro-B cells, showed a twofold increase in 7AAD[+]Annexin-V[+] apoptotic cells (Extended Data Fig. 3a). *Sirt7*[−/−] B220[+]CD19[+]IgM[−]CD43[−]FSC[hi] large pre-B cells were partially arrested at G1 (Extended Data Fig. 3b) and displayed reduced phosphorylation of STAT5 at Y694, the major driver of pre-B

**Fig. 1 | SIRT7 is required for normal B cell development. a**, *t*-Distributed stochastic neighbor embedding plots displaying the single-cell expression profiles of nuclear sirtuins in purified mouse BM Lin[−] cells (left) and scRNA-seq feature plot (right) identifying B cells, macrophages, hematopoietic and progenitor stem cells (HPSCs), innate lymphoid cells (ILCs) and T cells. Data were obtained from singlecell.broadinstitute.org (study SCP978). **b**, Immunoblot of SIRT7 expression in mouse BM CD19[−] cells and CD19[+] B cells (*n* = 3). **c,d**, Representative histograms (**c**) and quantification of SIRT7 median fluorescence intensity (MFI; **d**) in B220[+]CD19[−] pre-pro-B cells, B220[+]CD19[+]IgM[−]CD43[+] pro-B cells, B220[+]CD19[+]IgM[−]CD43[−] pre-B cells, B220[+]CD19[+]IgM[+] immature B cells and B220[hi]CD19[+] mature B cells measured by intracellular flow cytometry (*n* = 8 mice). **e**, Total number of B220[+]CD19[+] B cells in the BM of wild-type and *Sirt7*[Δ4–10] 129Sv mice (*n* = 9). Data were pooled from four independent experiments. **f,g**, Representative histograms (**f**) and numbers (**g**) of pre-pro-B cell, pro-B cell, pre-B cell, immature B cell and mature B cell populations identified as in **c** in the BM of 129Sv wild-type and *Sirt7*[Δ4–10] mice (*n* = 9). Data were pooled from four independent experiments. **h,i**,

Representative histograms (**h**) and numbers (**i**) of splenic CD19[+] B cells from wild-type (*n* = 5) and *Sirt7*[−/−] (*n* = 4) mice. **j**, Hematoxylin and eosin staining of histological sections from the spleens of wild-type and *Sirt7*[−/−] mice (*n* = 5); scale bar, 500 μm. **k**, Number of splenic B220[+]CD19[+]CD21[+]CD23[+]CD93[+] transitional B cells, B220[+]CD19[+]CD21[hi]CD23[−] marginal zone (MZ) B cells, B220[+]CD19[+]CD21[+] CD23[+]CD93[−] follicular B cells, B220[+]CD19[+]IgM[+]Gl7[+]Fas[+] germinal center (GC) B cells, B220[+]IgG1[+] class-switched B cells and BM CD19[+]CD38[+]CD138[−]Gl7[−] memory B cells and B220[lo]CD138[+] plasma cells from wild-type and *Sirt7*[−/−] mice (*n* = 4). **l,m**, Numbers of donor-derived CD45.1[−]CD45.2[+]CD19[+] B cells in the spleens of recipient CD45.1/CD45.2 mice 4 weeks after congenic transplantation of either CD45.2 wild-type (*n* = 4) or *Sirt7*[−/−] (*n* = 5) Lin[−]B220[+]CD19[+]IgM[−] pro-B cells expanded ex vivo with OP9 cells and 10 ng ml[−1] IL-7, SCF and FLT3-L (**l**) or *Sirt7*[−/−] Lin[−]B220[+]CD19[+]IgM[−] pro-B cells retrovirally expressing empty vector (EV; *n* = 6), SIRT7[WT] (*n* = 3) or SIRT7[H187Y] (*n* = 4; **m**). Data are presented as mean ± s.d. (**d**, **i**, and **k–m**) or mean ± s.e.m. (**e**, **g**, **l** and **m**) and were analyzed by one-tailed *t*-test (**e**, **g**, **i**, **k** and **l**) or one-way analysis of variance (ANOVA) with Sidak multiple comparisons (**d** and **m**).

cell proliferation[26], without altering the expression of the upstream IL-7 receptor subunit CD127 (Extended Data Fig. 3c,d). SIRT7 recruits 53BP1, a key player in distal VH-to-DJH recombination, to double strand breaks[24,27]. Splenic IgM⁺ *Sirt7*⁻/⁻ B cells exhibited reduced usage of distal VH588 and VH7183 segments, whereas the proximal segments DHL-to-JH3 and VHQ52-DJH remained unaffected (Extended Data Fig. 3e,f). However, crossing wild-type and *Sirt7*⁻/⁻ mice with mice expressing a transgenic HEL-specific immunoglobulin (IgHEL)[28], which bypasses the V(D)J-related checkpoints, did not restore B cell

differentiation in *Sirt7*⁻/⁻ mice (Extended Data Fig. 3g). These observations indicate that SIRT7 promotes B cell development through a mechanism independent of V(D)J recombination.

## SIRT7 regulates Pax5 in B cell progenitors

To investigate the mechanisms through which SIRT7 regulates B cell development, we performed RNA-seq in sorted wild-type and *Sirt7*⁻/⁻ B220⁺CD19⁺IgM⁻CD43⁺ pro-B cells and B220⁺CD19⁺IgM⁻CD43⁻ pre-B cells. In a principal component analysis (PCA), wild-type and

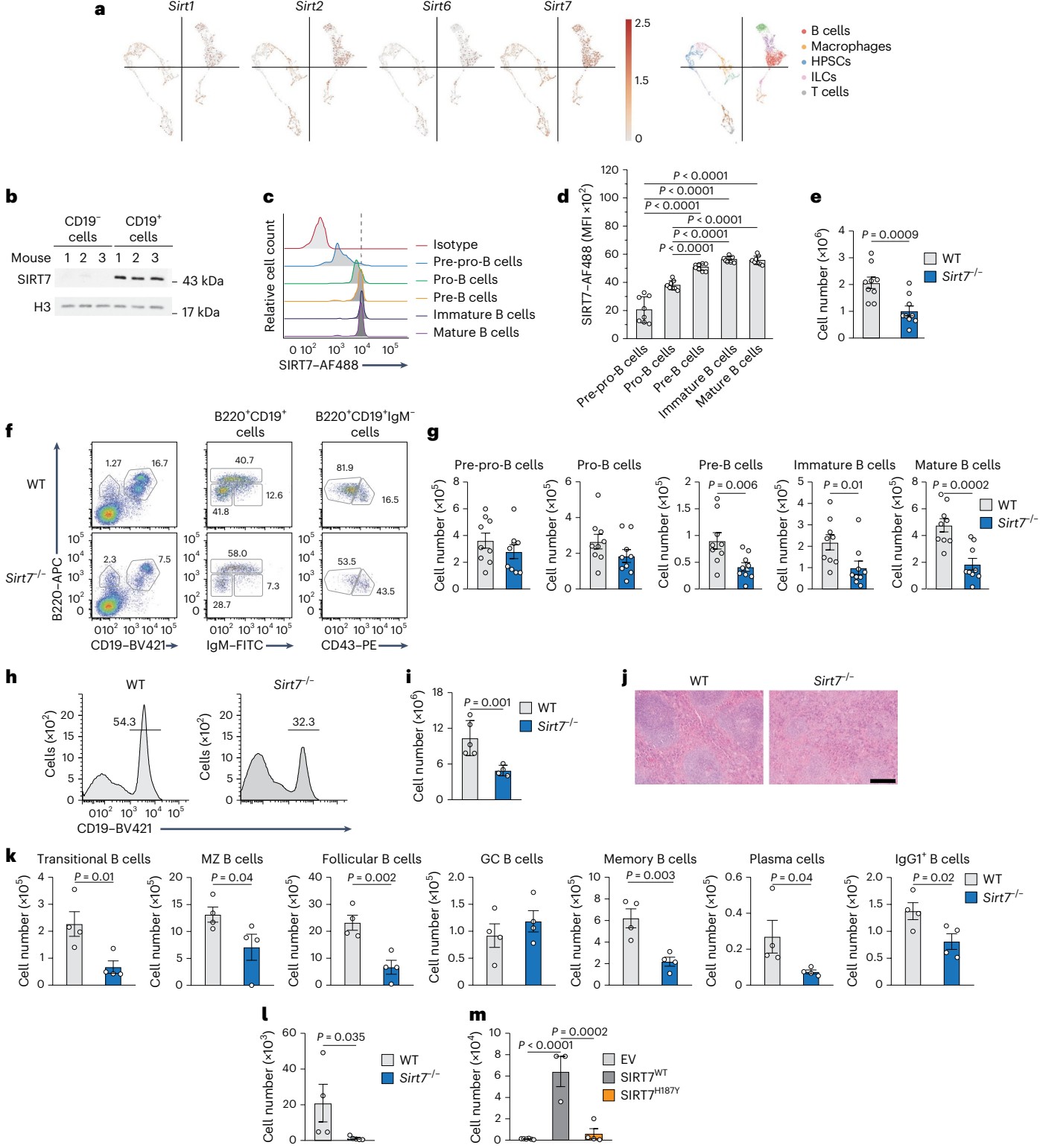

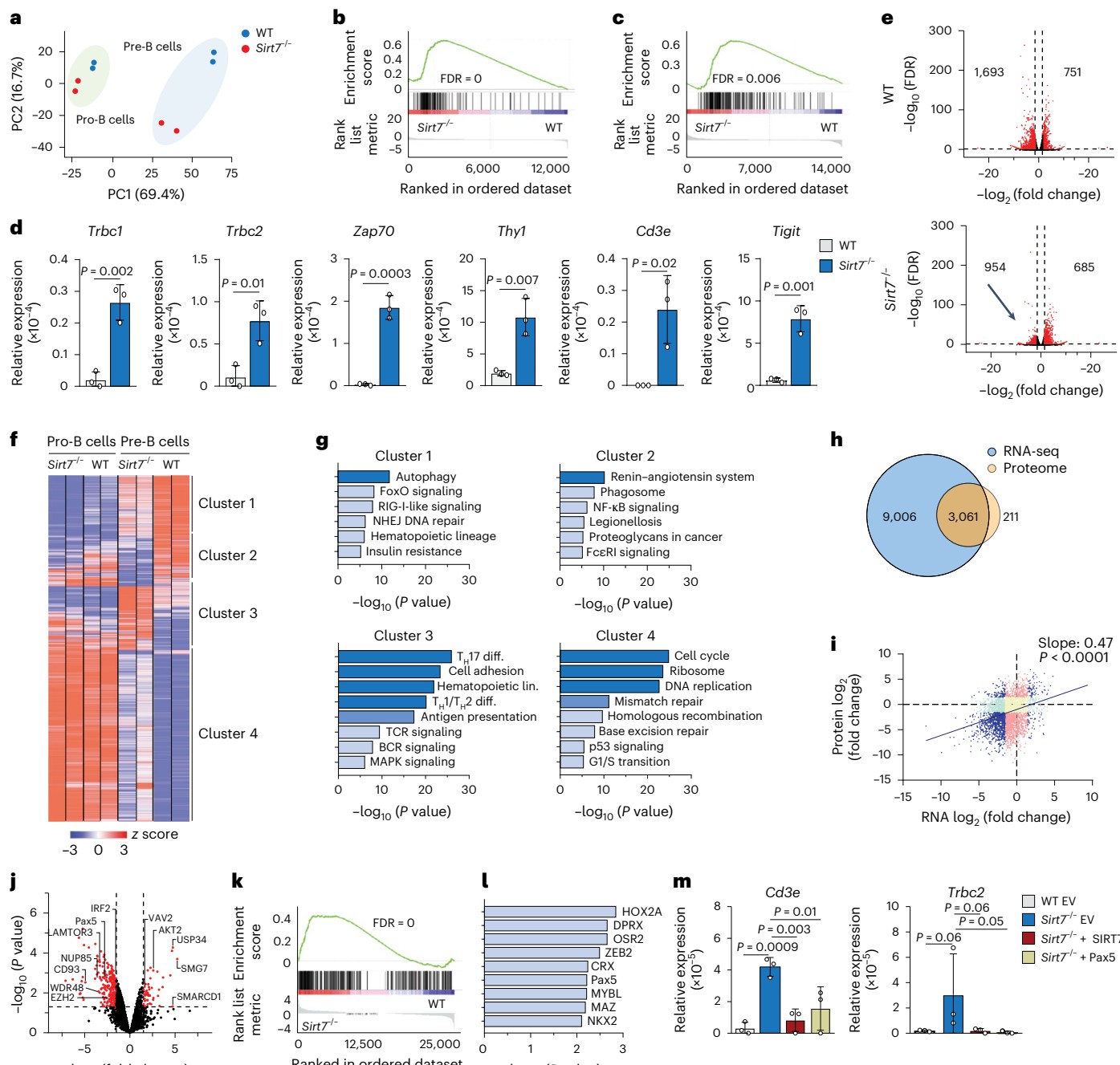

**Fig. 2 | SIRT7 regulates Pax5 in B cell progenitors. a**, PCA clustering of B220+CD19+IgM−CD43+ pro-B cells and B220+CD19+IgM−CD43− pre-B cells sorted from the BM of wild-type and *Sirt7*−/− 129Sv mice based on the top 2,000 differentially expressed genes (*n* = 2); PC, principal component. **b,c**, GSEA of the 'Travaglini lung proliferating NK T cell' (**b**) and 'Hallmark E2F targets' (**c**) gene sets in *Sirt7*−/− versus in wild-type B220+CD19+IgM−CD43− pre-B cells. **d**, RT–qPCR of *Trbc1*, *Trbc2*, *Zap70*, *Thy1*, *Cd3e* and *Tigit* expression relative to *Hprt* (*n* = 3). **e**, Volcano plot of differentially expressed genes in pre-B cells versus in pro-B cells (defined as in **a**) from wild-type (top) and *Sirt7*−/− (bottom) mice. Numbers on the right and left represent significantly induced or repressed genes, respectively. The black arrow indicates the lack of downregulated genes. Red dots represent significantly regulated genes (| log2 (fold change) | > 1.5, FDR < 0.05). **f**, Unsupervised clustering of the top 2,000 differentially expressed genes (*P* < 0.03) in wild-type and *Sirt7*−/− pro-B and pre-B cells defined as in **a**. **g**, Gene Ontology analyses of genes in clusters 1–4. Only terms with a −log10 (*P* value) of >5 are reported; NHEJ, nonhomologous end joining; T$_H$17 diff., differentiation of IL-17-producing helper T cells; lin., lineage; T$_H$1/T$_H$2 diff., differentiation of type 1 helper T/type 2 helper T cells; TCR, T cell antigen receptor; BCR, B cell receptor. **h,i**, Venn diagram displaying the number of detected proteins and transcripts (**h**)

and scatter plot (**i**) showing the linear correlation between the mean log2 (fold change) in protein expression and their corresponding transcript levels in wild-type B220+CD19+IgM−CD43− pre-B cells, as determined by proteomics and RNA-seq, respectively. **i**, Changes in protein and RNA levels with a | log2 (fold change) | of ≥1.5 (blue), changes in protein levels only (light red), changes in RNA levels only (light green) and changes of | log2 (fold change) | ≤1.5 (yellow) are shown. Fold change values were calculated relative to Nup50 expression. The Spearman correlation slope = 0.47; *P* < 0.0001 between protein and transcript pairs. **j**, Volcano plot of differentially expressed proteins in sorted *Sirt7*−/− versus wild-type B220+CD19+IgM−CD43− pre-B cells. Red dots indicate | log2 (fold change) | of ≥1.5 and a *P* value of <0.05. **k**, GSEA of *Sirt7*−/− versus wild-type B220+CD19+IgM−CD43− pre-B cells compared to Pax5 target genes[20]. **l**, HOMER de novo motif enrichment analysis showing nine of the top transcription factors enriched in cluster 3 genes. Motif enrichment was performed on promoter regions (transcription start site to 2,000 bp upstream). **m**, RT–qPCR of *Cd3e* and *Trcb2* relative to *Hprt* in wild-type or *Sirt7*−/− B220+CD19+IgM− pro-B cells retrovirally expressing empty vector, SIRT7 or Pax5 (*n* = 3). Data are shown as mean ± s.d. (**d** and **m**) and were analyzed by one-tailed *t*-test (**i**), two-tailed *t*-test (**d**) or one-way ANOVA with Fisher's least significant different (LSD) test (**m**).

$Sirt7^{-/-}$ pre-B cells clustered at a greater distance than wild-type and $Sirt7^{-/-}$ pro-B cells (Fig. 2a). SIRT7 regulated the expression of 220 genes in pro-B cells and 429 genes in pre-B cells (false discovery rate (FDR) < 0.05), including genes important for B cell development (pro-B: *Vpreb2*, *Ikzf3* and *Irf4*; pre-B: *Vpreb1*, *Runx2*, *Erg*, *Mef2c*, *Rag1* and *Bcl2*; Supplementary Table 1). Gene set enrichment analysis (GSEA) indicated that SIRT7 mainly repressed lineage-inappropriate (*Thy1* and *Il2ra*) and cell cycle-related genes (*Myc* and *Grb7*), such as 'lung NK and T cell' and 'E2F targets' gene sets (Fig. 2b,c and Supplementary Table 1). Accordingly, T cell-related genes, such as *Trbc1*, *Trbc2*, *Zap70*, *Thy1*, *Cd3e* and *Tigit*, were derepressed in ex vivo-expanded $Sirt7^{-/-}$ B cell progenitors (Fig. 2d).

Volcano plots indicated that $Sirt7^{-/-}$ pre-B cells induced gene expression normally but repressed fewer genes than wild-type pre-B cells (Fig. 2e and Extended Data Fig. 4a). We identified four gene clusters regulated by SIRT7 (Fig. 2f). Genes within clusters 1 and 2 showed lower expression in $Sirt7^{-/-}$ pro-B and pre-B cells than in their wild-type counterparts and were involved in autophagy, nonhomologous end joining DNA repair and cellular signaling. Cluster 3 genes were increased in $Sirt7^{-/-}$ pre-B cells relative to wild-type pre-B cells and were enriched for genes involved in T cell differentiation, suggesting compromised lineage commitment in $Sirt7^{-/-}$ pre-B cells. Cluster 4 comprised genes similarly expressed in wild-type and $Sirt7^{-/-}$ pro-B cells that were downregulated in wild-type pro-B cells but were only partially reduced in $Sirt7^{-/-}$ pre-B cells, including genes implicated in DNA repair and the cell cycle (Fig. 2g).

Quantitative mass spectrometry (MS) in sorted wild-type and $Sirt7^{-/-}$ B220$^+$CD19$^+$IgM$^-$CD43$^-$ pre-B cells identified 2,917 proteins (Supplementary Table 2), most of which were also identified by bulk RNA-seq analysis of the same cells (Fig. 2h). mRNA and protein expression levels positively correlated for most transcript–protein pairs (Fig. 2i), as previously reported for B-ALL[29]. Although the RNA-seq analysis identified differential expression for 1.6% of all detected transcripts between wild-type and $Sirt7^{-/-}$ pre-B cells, 5.2% of the proteome (Extended Data Fig. 4b), including proteins involved in DNA repair, cell cycle and intracellular signaling (Extended Data Fig. 4c), was differentially expressed.

Among the top differentially expressed proteins between wild-type and $Sirt7^{-/-}$ pre-B cells, we found several transcriptional and chromatin regulators (Pax5, IRF2, EZH2 and SMARCD1; Fig. 2j). Pax5 was downregulated 2.8-fold in $Sirt7^{-/-}$ pre-B cells compared to wild-type

cells (Fig. 2j and Supplementary Table 2). GSEA of the $Sirt7^{-/-}$ pre-B cell bulk RNA-seq dataset indicated that SIRT7 and Pax5 controlled a similar group of genes, including signaling genes, transcriptional regulators and surface receptors, such as *Vpreb1*, *Runx2*, *Bach2*, *Il2ra* and *Thy1* (Fig. 2k), whereas unsupervised de novo motif discovery analysis identified enrichment of the Pax5 motif in cluster 3 genes (Fig. 2l). Analysis of publicly available chromatin immunoprecipitation with sequencing (ChIP–seq) datasets[17] revealed that Pax5 bound to 62% of cluster 3 genes and to 43% of their promoters (Extended Data Fig. 4d). Retroviral expression of either SIRT7 or Pax5 in wild-type or $Sirt7^{-/-}$ BM CD19$^+$ cells restored *Cd3e* and *Trcb2* repression in $Sirt7^{-/-}$ CD19$^+$ cells (Fig. 2m), indicating that Pax5 expression bypassed SIRT7 deficiency to repress lineage-inappropriate genes. According to bulk RNA-seq datasets of wild-type and $Sirt7^{-/-}$ pro-B and pre-B cells, SIRT7 also regulated transcriptional programs not associated with Pax5, such as DNA replication and repair and ribosome biogenesis (Fig. 2g). These observations suggest that SIRT7 collaborates with Pax5 to repress transcription of lineage-inappropriate genes during early B cell development.

## SIRT7 regulates Pax5 stability by deacetylating K198

Intracellular flow cytometry of B220$^+$CD19$^+$IgM$^-$CD43$^+$ pro-B cells and B220$^+$CD19$^+$IgM$^-$CD43$^-$ pre-B cells and immunoblotting of B220$^+$CD19$^+$IgM$^-$ pro-B cells expanded in vitro indicated reduced amounts of Pax5 protein in $Sirt7^{-/-}$ pro-B and pre-B cells compared to their wild-type counterparts (Fig. 3a,b and Extended Data Fig. 5a). By contrast, RNA-seq and quantitative PCR with reverse transcription (RT–qPCR) on the same cells showed normal *Pax5* mRNA expression (Fig. 3c and Extended Data Fig. 5b), indicating that SIRT7 directly regulates Pax5 protein levels. SIRT7 retroviral expression in $Sirt7^{-/-}$ pro-B cells rescued Pax5 protein expression (Fig. 3d and Extended Data Fig. 5c). To test whether SIRT7 controlled Pax5 turnover, we used CRISPR–Cas9 to knock out *Sirt7* in the mouse pre-B cell line HAFTL (HAFTL$^{SIRT7KO}$ cells). Pax5 was downregulated in HAFTL$^{SIRT7KO}$ cells compared to in HAFTL cells, which was fully reversed by treatment with the proteasome inhibitor lactacystin (Fig. 3e and Extended Data Fig. 5d). Cycloheximide protein stability assays in HAFTL$^{SIRT7KO}$ compared to in HAFTL cells indicated that Pax5 half-life was reduced by twofold in the former (Fig. 3f,g), indicating that SIRT7 prevents Pax5 proteasomal degradation. SIRT7 interacted with endogenous Pax5 in HAFTL cells and with transiently expressed PAX5 in HEK293F cells (Fig. 3h and Extended Data Fig. 5e). In gel filtration chromatography of HAFTL nuclear extracts, a

---

**Fig. 3 | SIRT7 regulates Pax5 stability by deacetylating K198. a**, Pax5 MFI measured by intracellular flow cytometry of B220$^+$CD19$^+$IgM$^-$CD43$^+$ pro-B and B220$^+$CD19$^+$IgM$^-$CD43$^-$ pre-B cells from the BM of wild-type ($n = 6$) and $Sirt7^{-/-}$ ($n = 8$) mice. Data were pooled from three independent experiments. **b**, Immunoblot of Pax5 protein in wild-type and $Sirt7^{-/-}$ B220$^+$CD19$^+$IgM$^-$ pro-B cells expanded ex vivo for 4 days with OP9 cells and 10 ng ml$^{-1}$ IL-7, SCF and FLT3-L. **c**, RT–qPCR analysis of *Pax5* gene expression in wild-type and $Sirt7^{-/-}$ pro-B cells expanded as in **b** ($n = 3$). **d**, Immunoblot of Pax5 and SIRT7 protein in wild-type and $Sirt7^{-/-}$ B220$^+$CD19$^+$IgM$^-$ pro-B cells retrovirally transduced with empty vector and SIRT7. **e**, Quantification of Pax5 protein expression in wild-type and CRISPR–Cas9-generated HAFTL$^{SIRT7KO}$ cells treated with vehicle (Ct) or 2 µM lactacystin for 8 h ($n = 3$ biological replicates). **f,g**, Time course immunoblot (**f**) and Pax5/H3 ratio (**g**) in HAFTL ($n = 6$ independent time courses) and HAFTL$^{SIRT7KO}$ ($n = 8$ independent time courses) cells treated with 100 µg ml$^{-1}$ cycloheximide for 3, 6, 9 and 24 h. Nonlinear fits with variable slope (four parameters) are depicted. **h**, Flag-specific coimmunoprecipitation from HAFTL cells retrovirally transduced with empty vector or SIRT7–Flag, followed by Pax5 and SIRT7 immunoblotting; IP, immunoprecipitate. **i**, Immunoblot of Pax5 and SIRT7 in the input and gel filtration chromatography fractions in HAFTL cells. Approximate molecular weights are shown. Red brackets indicate fractions in which Pax5 and SIRT7 coelute. **j**, Immunoblot of Pax5 protein in HAFTL and KOPN-8 cells treated with vehicle (−) or 5 mM nicotinamide (NAM) for 48 h. **k**, Pan-acetyl-lysine (AcK) immunoblotting of in vitro deacetylation assays with purified Pax5 alone, with SIRT7 or

SIRT7 and NAD$^+$. **l**, Fragmentation MS/MS spectra of the KacRDEGIQ(+.98) ESPVPNGHSLPGR peptide, as determined by proteomic analysis of Pax5 protein immunoprecipitated from $SIRT7^{-/-}$ HEK293F cells transiently expressing Pax5 and empty vector (left) or Pax5 and SIRT7 (right) by polyethylenimine transfection. **m**, Schematic of mouse Pax5 functional domains; PRD, paired box domain; OP, octapeptide domain; HD, partial homeodomain; TAD, transactivation domain; ID, inhibitory domain. The start and end of each domain and the detected acetyl-lysine residues are indicated. **n**, Conservation of Pax5 186–229 peptide in *Mus musculus*, *Homo sapiens*, *Gallus gallus*, *Danio rerio* and *Xenopus laevis*. K198 is shown in red. **o**, Expression of Pax5$^{WT}$ and Pax5$^{K198Q}$ and Pax5$^{K198R}$ mutants expressed in HEK293F cells as in **l**. **p**, Ubiquitination of Flag-purified Pax5$^{WT}$ and Pax5$^{K198Q}$ and Pax5$^{K198R}$ mutants in HEK293F cells coexpressing HA–ubiquitin (HA–Ub) as in **l**. **q**, Expression of Pax5$^{WT}$, Pax5$^{K198Q}$ and Pax5$^{K198R}$ in HEK293F cells treated as in **g** ($n = 4$). **r**, Scatter plot of the correlation between Pax5 and PCAF protein in individuals with B-ALL[29], as determined by proteomics. Linear regression and 95% confidence intervals (dashed lines) are shown ($n = 27$, $P = 0.02$). **s**, Pax5 expression (left; pooled from three independent experiments) and Pax5 acetylation (right; Ack/Pax5, pooled from four (Pax5$^{WT}$) and three (Pax5$^{K198R}$) independent experiments) in HEK293F cells expressing Pax5$^{WT}$ or Pax5$^{K198R}$ together with empty vector, p300, PCAF, NCOA3 or GTF3C4. Data are shown as mean ± s.d. (**a**, **c**, **e**, **g**, **q** and **s**) and were analyzed by two-tailed *t*-tests with a Holm–Sidak comparison (**a**), one-tailed *t*-test (**e** and **r**), two-way ANOVA with Sidak comparisons (**g**) or one-way ANOVA with Fisher's LSD test (**q** and **s**).

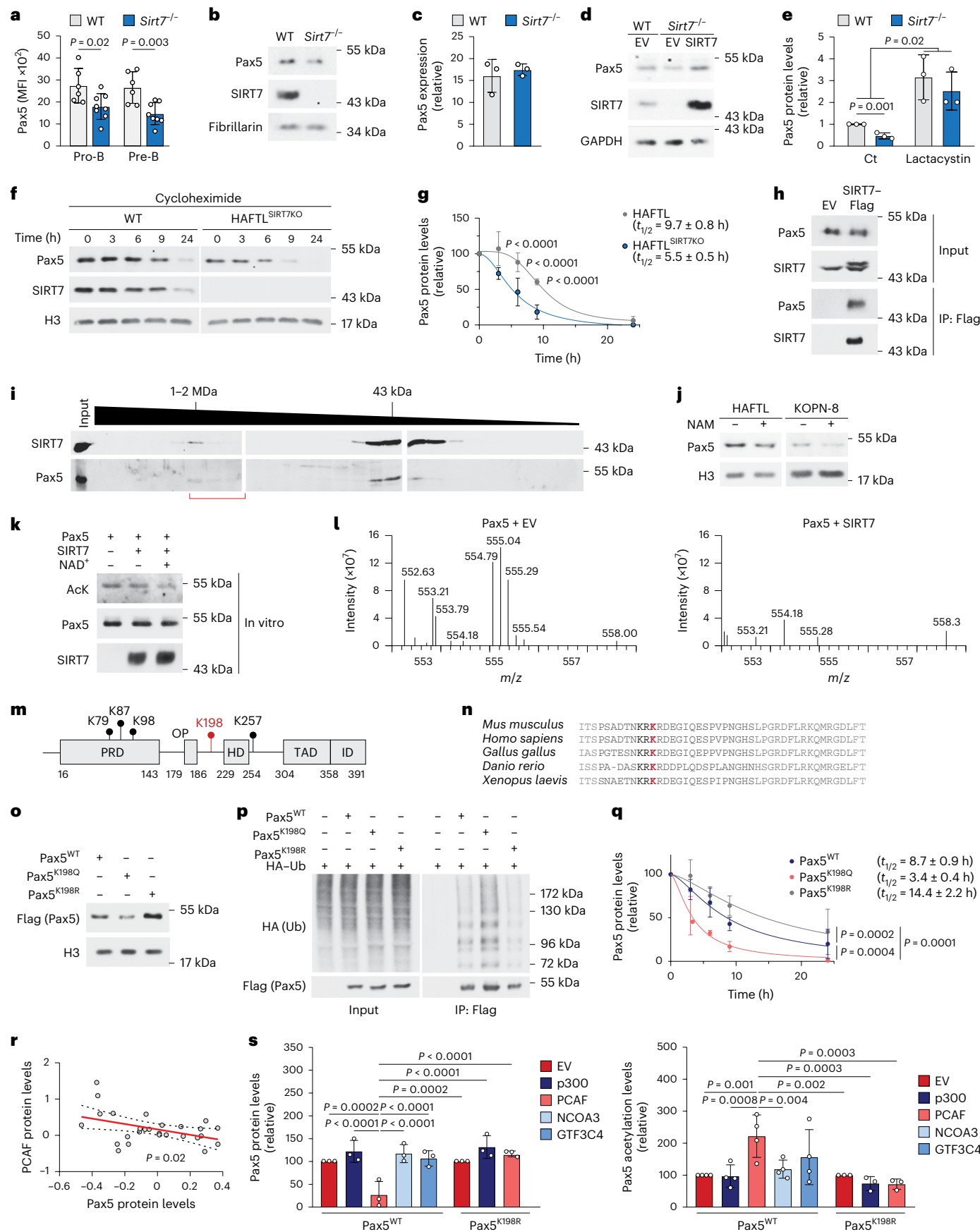

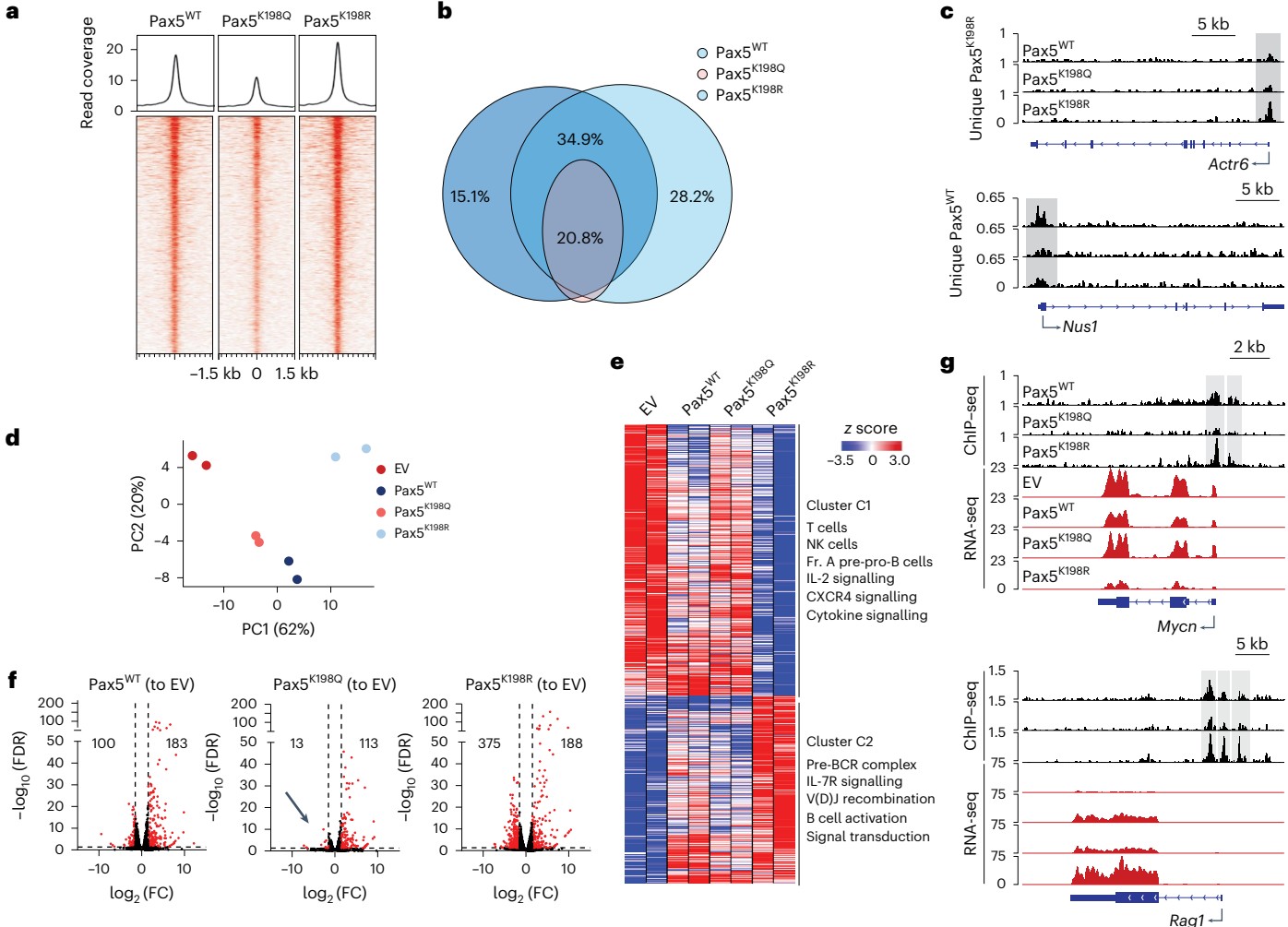

**Fig. 4 | Pax5 K198 acetylation regulates gene expression. a**, ChIP–seq analysis showing the genomic occupancy of Pax5-WT, Pax5-K198Q and Pax5-K198R in *Pax5*⁻/⁻ Lin⁻B220⁺IgM⁻ pro-B cells retrovirally transduced with Pax5^WT, Pax5^K198Q and Pax5^K198R and expanded ex vivo for 7 days in OP9 cells and in the presence of 10 ng ml⁻¹ IL-7, SCF and FLT3-L. One thousand random significant peaks are displayed. Top, read coverage profiles. **b**, Venn diagram of the overlap between the significant peaks (*q* < 0.05) detected by ChIP–seq of Pax5^WT, Pax5^K198Q and Pax5^K198R forms. **c**, Binding of Pax5^WT, Pax5^K198Q and Pax5^K198R in the *Actr6* (top) and *Nus1* regions (bottom). The *y* axis represents read coverage. **d**, PCA clustering of RNA-seq data from *Pax5*⁻/⁻ Lin⁻B220⁺IgM⁻ pro-B cells retrovirally transduced with empty vector, Pax5^WT, Pax5^K198Q and Pax5^K198R. **e**, Unsupervised clustering of differentially expressed genes (FDR < 0.05) in *Pax5*⁻/⁻ Lin⁻B220⁺IgM⁻ pro-B cells

retrovirally transduced with empty vector, Pax5^WT, Pax5^K198Q and Pax5^K198R as in **d**. Significant Gene Ontology terms for clusters C1 and C2 are shown. **f**, Volcano plots of differentially expressed genes in *Pax5*⁻/⁻ Lin⁻B220⁺IgM⁻ pro-B cells retrovirally transduced with Pax5^WT, Pax5^K198Q and Pax5^K198R versus empty vector. The black arrow indicates the absence of downregulated genes; significantly regulated genes (| log₂ (fold change) | ≥ 1.5, FDR < 0.05) are shown in red. Numbers on the right and left show the number of genes significantly induced and repressed, respectively; FC, fold change; Fr. A, fraction A. **g**, ChIP–seq of Pax5^WT, Pax5^K198Q and Pax5^K198R forms and RNA-seq of *Pax5*⁻/⁻ Lin⁻B220⁺IgM⁻ pro-B cells retrovirally expressing empty vector, Pax5^WT, Pax5^K198Q or Pax5^K198R in *Mycn* (top) and *Rag1* (bottom) genes. The *y* axis represents read coverage.

subpopulation of SIRT7 and Pax5 coeluted in high-molecular-weight (≈1 MDa) fractions (Fig. 3i), in line with previous observations that Pax5 works in concert with other transcription factors and epigenetic regulators[18].

We next investigated whether the catalytic activity of SIRT7 was required for Pax5 protein stabilization. Treatment with the pan-sirtuin inhibitor nicotinamide induced Pax5 degradation in HAFTL cells and B-ALL KOPN-8 cells, whereas the expression of *Pax5* mRNA was unaffected (Fig. 3j and Extended Data Fig. 5f,g). To test whether SIRT7 deacetylated Pax5 directly, we expressed and purified them in HEK293F cells and performed in vitro deacetylation assays. Incubation of Pax5 with SIRT7 in the presence of the cofactor NAD⁺ partially reduced Pax5 global acetylation (Fig. 3k). Ectopic expression of Pax5 alone or together with SIRT7 in *Sirt7*⁻/⁻ HEK293F cells followed by Pax5 purification and MS detected several acetylated residues, including K79, K87, K98, K198

and K257 (Fig. 3l,m). Of them, only K198 was lost in Pax5 isolated from *SIRT7*⁻/⁻ HEK293F cells overexpressing SIRT7 (as evidenced by the loss of the cluster of acetylated peptide peaks spanning from 554.79 to 555.54 *m/z*; Fig. 3l), indicating that SIRT7 specifically deacetylated Pax5 at K198. K198 is located within a putative Pax5 intrinsically disordered region of unknown function, located between the conserved octapeptide and the partial homeodomain[30] and conserved across Pax5 orthologs in chordates (Fig. 3n). A lysine-to-glutamine Pax5 mutant at K198 (Pax5^K198Q) that mimicked acetylated Pax5 was expressed at lower levels than wild-type Pax5 protein (Pax5^WT) in HEK293F cells, whereas a lysine-to-arginine mutant (Pax5^K198R) that mimicked the deacetylated form of Pax5 was expressed at higher levels than Pax5^WT (Fig. 3o and Extended Data Fig. 5h). When Pax5^WT, Pax5^K198Q and Pax5^K198R were expressed together with hemagglutinin–ubiquitin (HA–ubiquitin) in HEK293F cells, ubiquitin was preferentially loaded into Pax5^K198Q (Fig. 3p

and Extended Data Fig. 5i), suggesting increased proteasomal degradation. The half-life of Pax5[K198Q] was diminished 2.5-fold compared to Pax5[WT], whereas the half-life of Pax5[K198R] increased 1.7-fold (Fig. 3q). A K198 lysine-to-alanine mutant (Pax5[K198A]) showed similar protein levels (Extended Data Fig. 5j) and half-life as Pax5[K198R] in untreated and cycloheximide-treated HEK293F cells, respectively (Extended Data Fig. 5k,l), ruling out the possibility that the increased stability of Pax5[K198R] was due to potential modifications on arginine. Finally, cellular fractionation experiments showed that Pax5[WT], Pax5[K198Q] and Pax5[K198R] were similarly distributed in the nucleoplasm and chromatin and were excluded from the cytoplasm in HEK293F cells (Extended Data Fig. 5m). Thus, SIRT7-mediated deacetylation of Pax5 at K198 prevented the proteasomal degradation of Pax5 and enhanced its stability.

To identify the acetyltransferase(s) that catalyze Pax5 K198 acetylation, we searched for proteins with acetyltransferase activity[31] among reported interactors of Pax5 (Methods) and identified four potential candidates (p300, PCAF, GTF3C4 and NCOA3; Extended Data Fig. 5n and Supplementary Table 3). Analysis of the expression of Pax5 and the four candidates in publicly available proteomics data from 27 individuals with B-ALL[29] showed that only PCAF levels correlated negatively with Pax5 protein expression (Fig. 3r and Extended Data Fig. 5o). Following coexpression of Pax5 with all four enzymes in HEK293F cells, Pax5-WT expression was strongly reduced when coexpressed with PCAF, but not with the other acetyltransferases, compared to the levels of Pax5 expressed alone (Fig. 3s and Extended Data Fig. 5p). Pax5-K198R expression was similar whether coexpressed with PCAF or not (Fig. 3s and Extended Data Fig. 5q), suggesting that PCAF modulates Pax5 expression through K198 acetylation. To measure the ability of all four enzymes to acetylate Pax5, we coexpressed them with Pax5[WT] or Pax5[K198R], purified Pax5 by immunoprecipitation and measured global Pax5 acetylation by pan-acetyl-lysine immunoblotting. Only PCAF significantly increased Pax5[WT], but not Pax5[K198R] acetylation (Fig. 3s and Extended Data Fig. 5r), indicating that PCAF specifically acetylates Pax5 at K198. Thus, SIRT7 and PCAF regulate Pax5 protein stability by controlling Pax5 K198 acetylation.

## Pax5 K198 acetylation regulates gene expression

To investigate the impact of Pax5 K198 acetylation on Pax5 genome distribution and transcriptional activity, we performed ChIP–seq and RNA-seq in sorted Lin[−]B220[+]IgM[−] pro-B cells from Pax5[−/−] mice retrovirally infected with Pax5[WT], Pax5[K198Q] and Pax5[K198R]. ChIP–seq indicated that Pax5[K198Q] had markedly reduced binding to chromatin, whereas Pax5[K198R] occupancy at Pax5 target loci was strongly increased compared to Pax5[WT] (Fig. 4a). Motif enrichment analysis indicated that Pax5[K198Q] and Pax5[K198R] bound to the same Pax5 binding motif (Extended Data Fig. 6a). Differences in the binding to other transcription factor motifs were observed, including a reduced binding to the PU.1 motif and increased binding to the FLI1 motif by Pax5[K198R] compared to Pax5[WT] (Extended Data Fig. 6b).

Comparison of the Pax5[K198Q], Pax5[K198R] and Pax5[WT] peaks revealed that Pax5[K198Q] lost occupancy in nearly 80% of the peaks occupied by Pax5[WT] and Pax5[K198R] (for example, peaks in Klf6, Myc and Bcl6 genes; Fig. 4b and Supplementary Table 4). Notably, Pax5[K198R] bound 3,093 new regions (28.2% of all detected peaks), such as Csf1 and Actr6, and lost 1,659 binding sites (15.1%; for example, Nus1 and Fundc1), compared to Pax5[WT], whereas 6,616 peaks (55.8%; for example Cd19 and Vpreb1) were shared with Pax5[WT] (Fig. 4b). More than 90% of the regions bound by Pax5[K198R] were previously reported[17], such as those in the Actr6 and Foxo1 genes (Extended Data Fig. 6c), indicating that K198 deacetylation stabilized Pax5 binding at regions where it binds weakly rather than causing Pax5 redistribution. Consistently, Pax5[WT] and Pax5[K198Q] exhibited mild but clear binding to regions that were only bound significantly by Pax5[K198R] (Fig. 4c and Extended Data Fig. 6d). Similarly, we observed mild binding of Pax5[K198R] to unique Pax5[WT] sites, such as the Nus1 gene (Fig. 4c). Gene ontology analysis of the peaks uniquely bound

by Pax5[K198R] indicated enrichment in lineage-inappropriate genes (Csf1, Lif and Itga6) as well as in genes involved in B cell differentiation (Ikzf1 and Cxcr4), the cell cycle (Rb1 and Cdk4) and metabolism (Pten and Cox20; Extended Data Fig. 6e). These data suggest that K198 deacetylation promotes Pax5 binding to regions linked to B cell development and commitment.

In a PCA of RNA-seq data, Pax5[−/−] pro-B cells transduced with empty vector clustered closer to Pax5[−/−] pro-B cells expressing Pax5[K198Q] than to those expressing Pax5[WT] (Fig. 4d), whereas Pax5[−/−] pro-B cells expressing Pax5[K198R] clustered away from the rest (Fig. 4d). Unsupervised clustering unveiled two clusters: cluster C1, which comprised genes repressed by Pax5, such as lineage-inappropriate genes (Zap70 and Nkg7) and cytokine signaling genes (Stat3 and Il2rg; Fig. 4e), and cluster C2, which comprised genes induced by Pax5, such as genes related to pre-B cell receptor (Vpreb1a/b and Cd79a/b), IL-7 receptor-mediated signaling (Il7r and Stat5b) and V(D)J recombination (Foxo1 and Rag1/Rag2; Fig. 4e). Pax5[K198R] induced genes related to V(D)J recombination (Blnk, Irf4 and Rag1) more strongly than Pax5[WT] and Pax5[K198Q], whereas genes linked to proliferation (Il7r, Dusp4 and Mycn) were similarly regulated by Pax5[WT] and Pax5[K198Q], but not by Pax5[K198R] (Extended Data Fig. 6f,g). All three Pax5 proteins significantly increased the expression of a similar number of genes compared to Pax5[−/−] pro-B cells (for example, Cd19 and Vpreb3), with Pax5[K198R] inducing the expression of some targets (Cd55 and Bcar3) stronger than Pax5[WT] (Fig. 4f). By contrast, introduction of Pax5[K198Q] significantly reduced the expression of only 13 genes (for example, Cyp2d13 and Bcl2a1a), whereas Pax5[K198R] stringently repressed 375 genes (including Gata3 and Il2rb; Fig. 4f and Supplementary Table 5). These observations indicate that Pax5 K198 acetylation affects the extent to which Pax5 regulates its canonical targets rather than controlling different subsets of genes. Binding of Pax5[K198Q] and Pax5[K198R] to the promoter or to putative proximal enhancer regions correlated with the induction or silencing of the bound genes (Fig. 4g). Therefore, modulation of acetylation at Pax5 K198 creates a regulatory switch whereby K198-deacetylated Pax5 potently represses gene expression of lineage-inappropriate and cytokine signaling genes in pro-B cells, whereas K198-acetylated Pax5 fails to repress lineage-inappropriate genes but efficiently regulates genes linked to V(D)J and proliferation.

## Pax5 K198 deacetylation regulates B cell identity

To confirm that Pax5 deacetylation is required for B cell development and lineage restriction, we injected CD45.2 Pax5[−/−] Lin[−]B220[+]IgM[−] pro-B cells retrovirally expressing either empty vector or vectors encoding Pax5[WT], Pax5[K198Q] or Pax5[K198R] into sublethally irradiated CD45.1 mice. Four weeks after transplantation, CD45.2[+]B220[+]CD19[+] B cells were detected in the BM of mice injected with Pax5[−/−] pro-B cells expressing Pax5[WT], but not in mice transferred with Pax5[−/−] pro-B cells expressing Pax5[K198Q] or Pax5[K198R] (Fig. 5a and Extended Data Fig. 7a), indicating that Pax5 K198 dynamic acetylation/deacetylation is required for B cell development. Pax5-deficient pro-B cells differentiate into alternative lineages, including T cells[7,32]. Donor-derived CD45.2[+] CD4[+] and CD8[+] T cells were detected in the thymus of mice injected with Pax5[−/−] pro-B cells expressing empty vector or Pax5[K198Q], but not in those injected with Pax5[−/−] pro-B cells expressing Pax5[WT] or Pax5[K198R] (Fig. 5b and Extended Data Fig. 7b), suggesting that SIRT7-mediated K198 deacetylation is required for pro-B cell commitment. Transfer of sorted CD45.2 wild-type or Sirt7[−/−] Lin[−]B220[+]CD19[+]IgM[−] pro-B cells into sublethally irradiated CD45.1/CD45.2 mice revealed that 4 weeks after transplantation, donor-derived CD45.1[−]CD45.2[+]TCRβ[+] T cells and CD45.1[−]CD45.2[+]NKP46[+] NK cells were present in the spleens of mice injected with Sirt7[−/−] pro-B cells (Fig. 5c–e), indicating that Sirt7[−/−] pro-B cells differentiated into T cells and NK cells. Lin[−]IgM[+]IgD[+] mature B cells sorted from the spleens of CD45.2 wild-type or Sirt7[−/−] mice and transferred into sublethally irradiated CD45.1/CD45.2 mice did not lose CD19 expression, and no CD45.1[−]CD45.2[+]TCRβ[+] T cells were detectable in the spleens of mice injected with wild-type or Sirt7[−/−] mature B cells

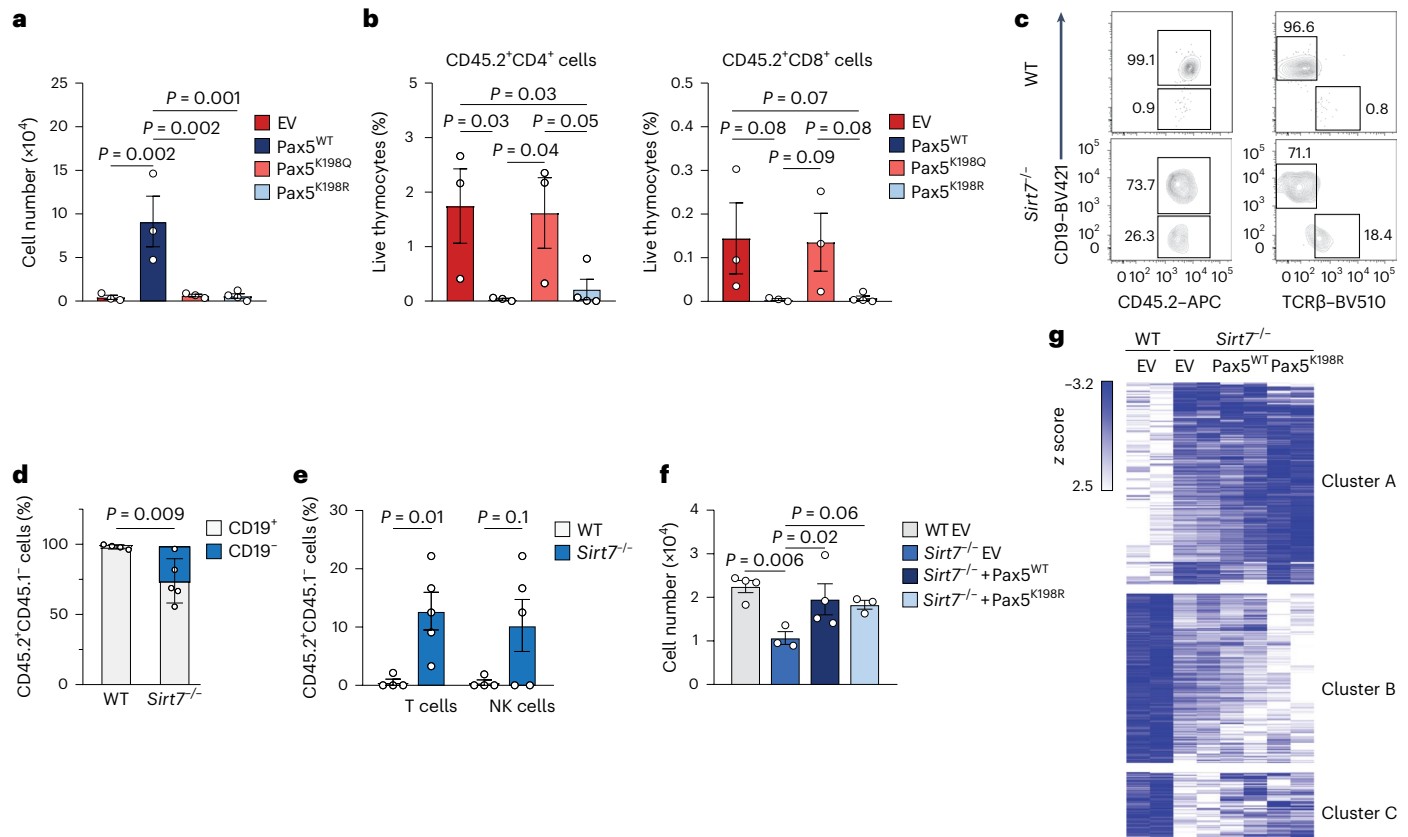

**Fig. 5 | Pax5 K198 deacetylation regulates B cell identity. a,b,** Number of donor-derived CD45.2⁺CD19⁺B220⁺ cells in the BM (**a**) and percentages of donor-derived CD45.2⁺CD4⁺ cells (left) and CD45.2⁺CD8⁺ (right) cells in the thymus (**b**) of CD45.1 wild-type recipient mice 4 weeks after transplantation of Lin⁻B220⁺IgM⁻ *Pax5⁻/⁻* pro-B cells retrovirally transduced with empty vector, Pax5ᵂᵀ, Pax5^K198Q or Pax5^K198R (empty vector, *n* = 3; Pax5ᵂᵀ, *n* = 3; Pax5^K198Q, *n* = 3; Pax5^K198R, *n* = 4). Data are presented as mean ± s.e.m. and were analyzed by one-way ANOVA with Fisher's LSD test. **c−e,** Representative flow cytometry plots (**c**) and percentages of donor-derived CD45.1⁻CD45.2⁺CD19⁺ and CD45.1⁻CD45.2⁺CD19⁻ cells (**d**) and donor-derived CD45.1⁻CD45.2⁺TCRβ⁺ and CD45.1⁻CD45.2⁺NKP46⁺ cells (**e**) in the spleens of CD45.1/CD45.2 recipient mice injected with CD45.2 wild-type (*n* = 4) or

*Sirt7⁻/⁻* (*n* = 5) Lin⁻B220⁺CD19⁺IgM⁻ pro-B cells. Data are presented as mean ± s.d. (**d**) or mean ± s.e.m. (**e**) and were analyzed by two-way ANOVA with Fisher's LSD test (**d**) or multiple *t*-tests with Holm−Sidak comparisons (**e**). **f,** Number of donor-derived B220⁺CD19⁺GFP⁺ B cells in the spleens of CD45.1/CD45.2 recipient mice injected with wild-type and *Sirt7⁻/⁻* Lin⁻B220⁺CD19⁺IgM⁻ pro-B cells expressing empty vector, Pax5ᵂᵀ and Pax5^K198R retroviruses (wild-type + empty vector, *n* = 4; *Sirt7⁻/⁻* + empty vector, *n* = 3; *Sirt7⁻/⁻* + Pax5ᵂᵀ, *n* = 3; *Sirt7⁻/⁻* + Pax5^K198R, *n* = 3). Data are presented as mean ± s.d. and were analyzed by one-way ANOVA with Fisher's LSD test. **g,** Unsupervised clustering of the differentially expressed genes (*P* < 0.05) between wild-type and *Sirt7⁻/⁻* Lin⁻B220⁺CD19⁺IgM⁻ pro-B cells retrovirally transduced with empty vector, Pax5ᵂᵀ or Pax5^K198R.

6 weeks after transplantation (Extended Data Fig. 7c), suggesting that SIRT7 is required for the establishment, but not maintenance, of B cell commitment.

Conditional deletion of *PCAF* and its homolog *Gcn5* in B cell progenitors impairs B cell development at the pro-B cell stage[33]. To test whether acetylation of Pax5 K198 is required for B lymphopoiesis, we retrovirally expressed Pax5ᵂᵀ or Pax5^K198R (together with green fluorescent protein (GFP)) in CD45.2 *Sirt7⁻/⁻* pro-B cells. After 4 weeks of transplantation into sublethally irradiated CD45.1/CD45.2 mice, we detected reduced numbers of B220⁺CD19⁺GFP⁺ B cells in the spleens of mice injected with *Sirt7⁻/⁻* pro-B cells compared to mice injected with wild-type cells. This effect was reversed in *Sirt7⁻/⁻* pro-B cells expressing Pax5ᵂᵀ or Pax5^K189R (Fig. 5f and Extended Data Fig. 7d), indicating that PAX5 K198 deacetylation rescues defective B cell development in *Sirt7⁻/⁻* pro-B cells. To understand how Pax5 rescues B cell development in *Sirt7⁻/⁻* progenitors at the gene expression level, we performed RNA-seq in these pro-B cells. Unsupervised clustering analysis revealed that expression of Pax5^K198R and, to a lesser extent, Pax5ᵂᵀ restored the repression of 93 of the 297 genes (31.3%) upregulated in *Sirt7⁻/⁻* pro-B cells, including *Cxcr5* and *Tet1* (cluster C), and did not affect the expression of genes downregulated in *Sirt7⁻/⁻* pro-B cells (cluster A; Fig. 5g and Supplementary Table 6). Pax5ᵂᵀ expression in *Sirt7⁻/⁻* pro-B cells mostly

repressed lineage-inappropriate genes (for example, *Nfatc2* and *Cyyr1*; Extended Data Fig. 7e), consistent with its role in lineage commitment. These results indicated that B cell development required the dynamic acetylation and deacetylation of Pax5, with Pax5 deacetylation being necessary to restrict lineage plasticity.

**The Pax5−SIRT7 interplay is conserved in human B-ALL**
Because Pax5 is a haploinsufficient tumor suppressor in human B-ALL[8,10], we investigated whether the SIRT7−PAX5 interplay was functionally relevant in this disease. In a panel of B-ALL cell lines, Pax5 and SIRT7 protein, but not mRNA, levels strongly correlated, despite their diverse genetic backgrounds (Fig. 6a−c). Retroviral overexpression of SIRT7 in NALM-20 and TANOUE cells (two B-ALL cell lines with reduced Pax5 levels) significantly increased Pax5 protein expression (Fig. 6d,e). Analysis of publicly available proteomics data from a cohort of ETV6-RUNX1⁺ and high hyperploidy (HeH⁺) B-ALL cases[29] found a highly significant correlation between Pax5 and SIRT7 protein levels in these individuals (*P* = 0.005; Fig. 6f). *PAX5* and *SIRT7* mRNA correlated weakly in HeH⁺ B-ALL (Extended Data Fig. 8a), although the correlation was stronger for the protein (protein, *P* = 0.019; RNA, *P* = 0.026). A comparable link at the protein level was found in a panel of B cell chronic lymphoblastic leukemia (B-CLL) cell lines[34] (Extended Data

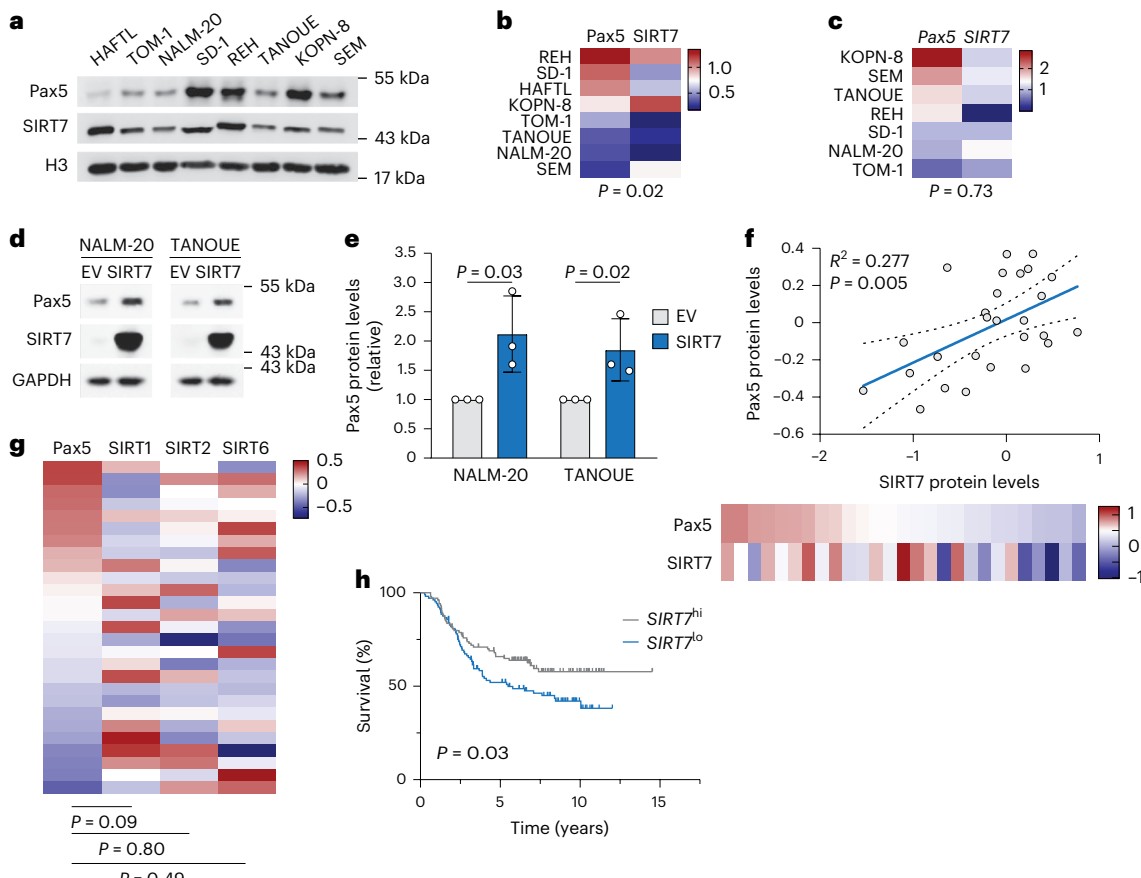

**Fig. 6 | The Pax5–SIRT7 interplay is conserved in human B-ALL. a**, Immunoblot of Pax5, SIRT7 and H3 in HAFTL, TOM-1, NALM-20, SD-1, REH, TANOUE, KOPN-8 and SEM cell lines. **b,c**, Heat map of the correlation between Pax5 and SIRT7 protein expression normalized to H3 expression (**b**) and *Pax5* and *Sirt7* RNA expression normalized to *Hprt* RNA expression (**c**) in B-ALL cell lines as in **a**. Mean *z* scores for the normalized protein (**b**) and mRNA (**c**) are shown. The *P* values were determined by Spearman's rank correlation. Data were pooled from three (**b**) or two (**c**) experiments. **d,e**, Immunoblots of the levels of Pax5 and SIRT7 in NALM-20 and TANOUE B-ALL cells expressing empty vector and SIRT7 retroviruses (**d**) and quantification of the Pax5/GAPDH ratio (**e**). Data are shown as mean ± s.d. and were analyzed by one-tailed *t*-tests (*n* = 3 biological replicates). **f**, Scatter plot (top) and heat map (bottom, *z* score) of Pax5 and SIRT7 protein determined by proteomics in individuals with B-ALL[29] (*n* = 27). Each point corresponds to one sample. Linear regression, 95% confidence intervals (dashed lines) and Spearman's rank correlation coefficient (*R*²) are shown (*P* = 0.005, one-tailed *t*-test). **g**, Heat map of the correlation between the expression of Pax5, SIRT1, SIRT2 and SIRT6 in the same human B-ALL samples as in **f**. **h**, Kaplan–Meier survival curves for children with high-risk B-ALL from the COG-P9906 study[35,36] (*n* = 209) stratified as higher (*SIRT7*ʰⁱ) or lower (*SIRT7*ˡᵒ) than median *SIRT7* RNA expression. Statistical significance was determined by a long-rank test (*P* value < 0.03).

Fig. 8b), indicating that the SIRT7–Pax5 expression correlation was independent of genetic background and differentiation stage. Among the tested sirtuins, only SIRT7 associated with Pax5 protein levels in individuals with B-ALL[29] (Fig. 6g and Extended Data Fig. 8c). Stratification of pediatric individuals with B-ALL from the COG9906 study[35,36] according to their *SIRT7* mRNA expression indicated that those with higher than median *SIRT7* mRNA levels had a better prognosis than those expressing lower than median *SIRT7* mRNA (Fig. 6h), indicating that SIRT7 could be an independent prognostic factor in human B-ALL. *PAX5* inactivating mutations in B-ALL are usually monoallelic[8]. Classification of individuals with B-ALL[35–37] into those with or without *PAX5* deletions and further into *SIRT7*ʰⁱ and *SIRT7*ˡᵒ subgroups indicated that higher than median SIRT7 expression tended to be protective only in individuals with B-ALL with *PAX5* deletions (Extended Data Fig. 8d), suggesting a tumor-suppressive function of SIRT7 by increasing PAX5 dosage in individuals with B-ALL with haploinsufficient *PAX5*.

## Discussion

Here, we present evidence implicating SIRT7 in B cell lymphopoiesis by establishing an acetylation switch on Pax5 that regulates its essential role in B cell development and identity. SIRT7-dependent deacetylation of Pax5 K198 promoted its protein stability, resulting in increased occupancy in a wide range of target genes, and enhanced its repressive activity. Conversely, PCAF-mediated Pax5 K198 acetylation reduced Pax5 protein levels and was associated with a global reduction of Pax5 occupancy and impaired repression of lineage-inappropriate genes. Transplantation experiments of Pax5 K198-acetylated and K198-deacetylated mimics demonstrated that, although a deacetylated mimic Pax5-K198R was enough to drive lineage commitment, both forms were required for B cell differentiation.

SIRT7 mRNA and protein levels were upregulated during early B cell development, coinciding with the activation of Pax5 expression in pro-B cells[38]. SIRT7 participated in repression of lineage-potential genes and balanced the expression of DNA repair and cell cycle factors to preserve pre-B cell proliferation and survival. SIRT7 also enhanced distal *Igh* recombination in pro-B cells and behaved as a tumor suppressor in B-ALL, suggesting that SIRT7 may contribute to the prevention of malignant transformation in hematopoietic progenitors. The identification of PCAF as the major Pax5 K198 acetyltransferase provided a mechanistic explanation for the impairment of pro-B cell development described following conditional deletion of *Pcaf* and its homolog *Gcn5* in mice[33]. Furthermore, the observed deacetylation of PCAF by SIRT7 in colon cancer cells[39] suggests that both enzymes may have an interdependent relationship during B cell lymphopoiesis.

Expression of the constitutively deacetylated Pax5[K198R] mutant in *Pax5*[-/-] pro-B cells abrogated their T cell potential but failed to restore B cell development, suggesting that PAX5 promotes these processes through independent mechanisms. Because Pax5-acetylated or Pax5-deacetylated mimics could not rescue B cell differentiation, both forms seem to be required for lymphopoiesis, raising the question of how Pax5 acetylation is regulated during B development. One possibility is that pools of acetylated and deacetylated Pax5 coexist in B cell progenitors and that their balanced actions lead to optimal B cell differentiation. Despite its ability to efficiently repress lineage plasticity in pro-B cells, the deacetylated mimic Pax5[K198R] overactivated *Igh* recombination genes and failed to induce normal expression of clonal expansion programs, consistent with its roles in V(D)J recombination[15–17] and as a negative regulator of pro-B cell proliferation[13]. Thus, it is also possible that a pool of constitutively deacetylated Pax5 ensures lineage commitment throughout B cell development, whereas a pool of Pax5 that undergoes dynamic acetylation and deacetylation controls the B cell developmental programs.

K198 is located within a putative intrinsically disordered region of Pax5, between the octapeptide and the partial homeodomain[30]. Although K198 acetylation strongly decreased Pax5 protein stability, it may also regulate Pax5 in other ways. Pax5 shapes chromatin organization and transcription by recruiting chromatin remodelers and histone modifiers to its target loci[18], so the observed cofractionation of Pax5 and SIRT7 in high-molecular-weight complexes suggests that K198 deacetylation may also influence the ability of Pax5 to mediate chromatin modification. Additionally, acetylation of Pax5 K198 might also affect global nuclear organization[12], as a substantial number of Pax5 binding sites lie in distal intergenic regions[18,20], and Pax5 binding to DNA plays a key organizational role in genome architecture, independent of transcription[12].

Finally, our findings provide a potential avenue to target leukemia pharmacologically. Inactivating mutations of the *PAX5* gene are present in 30% of B-ALL cases[8], and restoration of Pax5 levels in human and mouse B-ALL cells leads to leukemia regression[40,41]. Because *PAX5* mutations in B-ALL are usually monoallelic, enhancing wild-type *PAX5* allele function to overcome its haploinsufficiency represents a promising strategy for B-ALL treatment. Because SIRT7 activity strongly increased Pax5 levels in B cell progenitors, human B-ALL samples and cell lines, we propose that developing SIRT7-activating compounds to stimulate Pax5 functions may provide a basis for leukemia therapies. Further research should establish the relevance of the SIRT7–Pax5 regulatory axis in B cell malignancies.

## Online content

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

[1]Chromatin Biology Laboratory, Josep Carreras Leukemia Research Institute, Badalona, Spain. [2]Cancer Epigenetics Laboratory, Josep Carreras Leukemia Research Institute, Badalona, Spain. [3]Pompeu Fabra University, Barcelona, Spain. [4]Department of Genetics, Human Genetics Institute of New Jersey, Rutgers University, Piscataway, NJ, USA. [5]Epigenetic Control of Hematopoiesis Laboratory, Josep Carreras Leukemia Research Institute, Badalona, Spain. [6]Department of Cardiac Development and Remodeling, Max Planck Institute for Heart and Lung Research, Bad Nauheim, Germany. [7]Proteomics Unit, Josep Carreras Leukemia Research Institute, Badalona, Spain. [8]Centro de Investigacion Biomedica en Red Cancer (CIBERONC), Madrid, Spain. [9]Institucio Catalana de Recerca i Estudis Avançats (ICREA), Barcelona, Spain. [10]Physiological Sciences Department, School of Medicine and Health Sciences, University of Barcelona, Barcelona, Spain. [11]Division of Molecular Hematology, Department of Laboratory Medicine, Lund Stem Cell Center, Faculty of Medicine, Lund University, Lund, Sweden. [12]Departament de Biologia Cel.lular, Fisiologia i Immunologia, Universitat Autònoma de Barcelona, Bellaterra, Spain. [13]These authors contributed equally: Maria Espinosa-Alcantud, Alberto Bueno-Costa. ✉e-mail: bvazquez@carrerasresearch.org; avaquero@carrerasresearch.org

## Methods

### Mice

Two previously reported germline CD45.2 $Sirt7^{-/-}$ mouse models that contained exon 4 to 10 (ref. 24; $Sirt7^{\Delta4-10}$) or exon 4 to 9 deletions[25] ($Sirt7^{\Delta4-9}$) and that were maintained on the 129Sv or C57BL/6 genetic background, respectively, were used. 129Sv $Sirt7^{-/-}$ mice were used for most experiments. IgHEL[28] mice (CD45.2) were on the 129Sv background, whereas $Pax5^{-/-}$[5] and CD45.1 mice were on the C57BL/6 background. Heterozygous CD45.1/CD45.2 mice were generated by crossing CD45.1 C57BL/6 and wild-type CD45.2 129Sv mice for one generation to avoid rejection following transplantation of 129Sv cells. All mice were bred at the Comparative Medicine and Bioimage Centre of Catalonia animal facility of the Germans Trias i Pujol Research Institute. Animal studies were conducted at Josep Carreras Leukemia Research Institute (IJC) (Spain) according to national authorities and institutional ethics committees (Germans Trias i Pujol Reserach Institute Ethics Committee). The collection of BM samples from C57BL/6 wild-type and $Sirt7^{-/-}$ mice and the generation of $Pax5^{-/-}$ mouse B cell progenitors were conducted according to national authorities and institutional ethics committees at Max Plank Institute for Heart and Lung Research (MPI-HLR) (Germany) and Lund University (Sweden), respectively. Mouse housing conditions included a temperature of 21–25 °C, 40–70% humidity and a light cycle from 0800 to 2000 h with a 15-min intensity ramp simulating sunrise and sunset.

### Determination of anti-HEL isotypes

To analyze antigen-specific responses, mice were intraperitoneally injected with 100 µl of an emulsion containing 50 µg of NP–HEL along with complete Freund adjuvant. After 14 days, mice were bled, and serum samples were collected. For antibody isotype detection, enzyme-linked immunosorbent assay plates were coated overnight at 4 °C with HEL peptide (6 µg ml$^{-1}$). Following removal of excess peptide, serum dilutions from control or immunized mice were added and incubated for 1 h at room temperature. After three washes with a 0.05% Tween–PBS solution, rat anti-mouse immunoglobulin subclasses (IgM, IgG1 and IgG3; 1:250) were added for 1 h at room temperature. After 3 washes, an anti-rat immunoglobulin (1:5,000) was added for 1 h at room temperature, followed by three additional washes. Finally, TMB developing solution was added for 15 min and stopped with 1 N H$_2$SO$_4$. Color production was measured at 450 nm and 570 nm (for background subtraction) using a Multiskan Sky (Thermo Fisher) plate reader.

### Transplantation experiments

Two million short-term expanded pro-B cells or 1.5 million splenic Lin$^-$IgM$^+$IgD$^+$ B cells were washed and resuspended in 200 µl of PBS supplemented with 1% heat-inactivated fetal calf serum. Cells were injected via the tail vein into sublethally irradiated (5 Gy) CD45.1/CD45.2 or CD45.2 6- to 10-week-old randomized recipient mice. Four weeks after transplantation, the spleen (CD45.1/CD45.2 mice) or BM and thymus (CD45.2 mice) were collected and analyzed by fluorescence-activated cell sorting (FACS).

### Flow cytometry and cell sorting

BM, spleen and thymus samples were collected from wild-type, $Sirt7^{-/-}$, IgHEL$^{+/-}$, $Sirt7^{-/-}$IgHEL$^{+/-}$, CD45.1/CD45.2 and CD45.2 mice. BM samples were crushed in staining buffer (3% fetal bovine serum (FBS) and 2 mM EDTA in PBS) to obtain single-cell suspensions. Spleen and thymus samples were similarly processed. Red blood cells were lysed in ACK buffer (Gibco), and the reaction was stopped by adding 5 volumes of staining buffer. Cells were filtered through 40-µm sterile strainers and incubated with Fc-block (eBioscience) before staining for 30 min (4 °C) with anti-B220 (RA3-6B2, eBioscience, 1:400), anti-CD19 (HIB19, eBioscience, 1:400), anti-CD43 (eBioR2/60 or 1G10, BD Biosciences, 1:800), anti-IgM (II/41, eBioscience, 1:800), anti-IgD (11-26c, eBioscience, 1:400), anti-CD21 (7G6, BD Biosciences, 1:800), anti-CD23

(B3B4, BD Biosciences, 1:200), anti-CD93 (AA4.1, eBioscience, 1:400), anti-Gl7 (Gl7, eBioscience, 1:200), anti-CD38 (90, eBioscience, 1:400), anti-CD138 (300506, Invitrogen, 1:400), anti-Fas (SA367H8, Biolegend, 1:600), anti-IgG1 (A85-1, BD Pharmigen, 1:600), anti-CD127 (eBioSD/199, eBioscience, 1:200), anti-CD45.1 (A20, eBioscience, 1:400), anti-CD45.2 (104, eBioscience, 1:200), anti-TCRβ (H57-597, BD Biosciences, 1:200), anti-NKP46 (29A1.4, Biolegend, 1:100), anti-CD4 (GSK1.5, eBioscience, 1:800), anti-CD8 (53-6.7, eBioscience, 1:800), anti-hCD4 (RPA-T4, Biolegend, 1:400), anti-CD3e (145-2C11, BD Biosciences, 1:400), anti-Ly76 (TER-119, BD Biosciences, 1:400), anti-CD11b (M1/70, BD Biosciences, 1:400) or anti-GR1 (RB6-8C5, BD Biosciences, 1:400). After staining, cells were washed twice and resuspended in staining buffer before analysis with a FACS Canto II (BD Biosciences) or sorting with a FACSAria II (BD Biosciences). B cell subsets are defined as pre-pro-B (B220$^+$CD19$^-$), pro-B (B220$^+$CD19$^+$IgM$^-$CD43$^+$), pre-B (B220$^+$CD19$^+$IgM$^-$CD43$^-$), large pre-B (B220$^+$CD19$^+$IgM$^-$CD43$^-$FSC$^{hi}$), small pre-B (B220$^+$CD19$^+$IgM$^-$CD43$^-$FSC$^{lo}$), immature B (B220$^+$CD19$^+$IgM$^+$), BM mature B (B220$^{hi}$CD19$^+$), marginal zone B (B220$^+$CD19$^+$CD21$^{hi}$CD23$^-$), transitional B (B220$^+$CD19$^+$CD21$^+$CD23$^+$CD93$^+$), follicular B (B220$^+$CD19$^+$CD21$^+$CD23$^+$CD93$^-$), germinal center B (B220$^+$CD19$^+$IgM$^+$Gl7$^+$Fas$^+$), memory B (CD19$^+$CD38$^+$CD138$^-$Gl7$^-$), class-switched IgG1$^+$ B cells (B220$^+$IgG1$^+$) and plasma cells (B220$^{lo}$CD138$^+$).

Apoptosis was measured in BM pro-B and pre-B cells stained with 7AAD (BD Biosciences) and Annexin V-FITC (Abcam), following the manufacturer's instructions. To measure cell cycle distribution, BM pre-B cells were stained with surface markers and fixed and incubated with a solution containing 1 µg ml$^{-1}$ DAPI in permeabilization buffer for at least 1 h before FACS analysis. All flow cytometry experiments were analyzed with FlowJo.

For SIRT7 and Pax5 intracellular staining, BM single-cell suspensions were stained and washed before fixation for 30 min and permeabilization with Foxp3/Transcription Factor Fixation/Permeabilization buffers (eBioscience) according to manufacturer's instructions. Permeabilized cells were blocked with 2% FBS for 10 min, stained with anti-SIRT7 (D3K5A, Cell Signaling, 0.25 µg per sample) or anti-Pax5 (1H9, eBioscience, 0.2 µg per sample) for 1 h at room temperature and washed twice with permeabilization buffer containing 2% FBS. For SIRT7 staining, cells were subsequently incubated with a polyclonal anti-IgG (H + L) secondary antibody (Invitrogen, 0.25 µg per sample) for 1 h at room temperature before FACS analysis.

For phospho-STAT5-Y694 staining (47/Stat5(pY694), BD Biosciences, 20 µl per sample) in pre-B cells, BM cells were incubated in RPMI (Gibco) for 30 min at 37 °C and stimulated with IL-7 (5 ng ml$^{-1}$) in RPMI for an additional 30 min at 37 °C. Cells were fixed with Foxp3/Transcription Factor Fixation buffer, followed by washing and incubation with ice-cold 100% methanol for 1 h. After Fc blocking, cells were stained and analyzed with a FACS Canto II.

### Isolation of primary B cell progenitors

BM wild-type and $Sirt7^{-/-}$ pro-B cells were isolated by magnetic-activated cell separation enrichment of CD19$^+$ cells, followed by cell sorting. Briefly, BM single-cell suspensions were prepared in staining buffer. Cells were stained with Fc-block for 20 min and an additional 30 min with a biotinylated antibody to CD19 (1D3, BD Biosciences, 0.1 µg per 10 million cells). After washing, stained cells were incubated with Streptavidin MicroBeads (Miltenyi) and separated magnetically. CD19$^+$ cells were cultured overnight in Opti-MEM supplemented with 10% heat-inactivated FBS, 25 mM HEPES, 50 µg ml$^{-1}$ gentamicin and 50 µM β-mercaptoethanol in the presence of 10 ng ml$^{-1}$ IL-7, 10 ng ml$^{-1}$ SCF and 10 ng ml$^{-1}$ FTL3-L before Lin$^-$CD19$^+$B220$^+$IgM$^-$ cell sorting. Fetal liver $Pax5^{-/-}$ B cell progenitors were obtained as described previously[13].

### Cells and reagents

HAFTL, TANOUE, NALM-20, REH, KOPN-8, SD-1, SEM and TOM-1 cells were cultured in RPMI supplemented with 10% heat-inactivated FBS and

100 U ml$^{-1}$ penicillin/streptomycin (Gibco), whereas HEK293F, *SIRT7*$^{-/-}$ HEK293F (described in Simonet et al. [42]) and Platinum E cells were grown in DMEM (Gibco) supplemented with 10% FBS and 100 U ml$^{-1}$ penicillin/streptomycin. OP9 cells were maintained in MEM-α (Gibco) supplemented with 20% FBS and 100 U ml$^{-1}$ penicillin/streptomycin. KOPN-8, NALM-20, REH, TANOUE, SD-1, TOM-1 and SEM cells were kindly provided by M. Parra (IJC; purchased from the DSMZ German Collection of Microorganisms and Cell Cultures). OP9 and HEK293F cells were purchased from ATCC, and Platinum E cells were purchased from Cell Biolabs. None of the cell lines used were found in the Commonly misidentified lines database. Primary pro-B cells from wild-type, *Sirt7*$^{-/-}$ or *Pax5*$^{-/-}$ mice were plated onto a layer of mitomycin C-inactivated OP9 feeder cells and grown on Opti-MEM supplemented with 10% heat-inactivated FBS, 25 mM HEPES, 50 μg ml$^{-1}$ gentamicin and 50 μM β-mercaptoethanol in the presence of 10 ng ml$^{-1}$ IL-7, 10 ng ml$^{-1}$ SCF and 10 ng ml$^{-1}$ FTL3-L. All cells were cultured at 37 °C in a humidified atmosphere containing 5% $CO_2$. For transient transfections, HEK293F or *SIRT7*$^{-/-}$ HEK293F cells were transfected using polyethylenimine and the corresponding plasmids. For retroviral transduction of pro-B cells and B-ALL cell lines, Platinum E cells were transiently transfected with polyethylenimine, a pVSV-G vector encoding the viral envelop and pMIG bicistronic vectors encoding either the selection marker (hCD4 or GFP) alone or together with the SIRT7$^{WT}$, SIRT7$^{H187Y}$, Pax5$^{WT}$, Pax5$^{K198Q}$ or Pax5$^{K198R}$ coding sequences. Pro-B cells were resuspended in retroviral supernatants, centrifuged for 1.5 h at 1,000*g* (32 °C) and selected by hCD4$^+$ cell sorting 96 h after infection. Treatments were performed with 5 mM nicotinamide (Sigma) for 48 h, 100 μg ml$^{-1}$ cycloheximide (Sigma-Aldrich) for the indicated times, 2 μM lactacystin (Santa Cruz Biotechnology) for 8 h and 1 μM trichostatin A (Sigma-Aldrich) for 3 h.

### Histology
Spleens from wild-type and *Sirt7*$^{-/-}$ mice were collected, fixed in 10% formalin for 24 h, embedded in paraffin and sectioned at 4 μm before staining with hematoxylin and eosin. Histological sections were visualized with an Olympus BX53 microscope.

### Immunoprecipitation, gel filtration high-performance liquid chromatography and immunoblotting
For immunoprecipitation, cell pellets were lysed in RIPA buffer (50 mM Tris-HCl (pH 8.0), 150 mM NaCl, 0.5% sodium deoxycholate, 0.1% SDS, 1% NP-40 and 2 mM $MgCl_2$) containing cOmplete Protease Inhibitor (Roche) and incubated for 8 h with benzonase nuclease (Millipore) at 4 °C. Cell lysates were clarified by centrifugation (17,000*g* for 10 min at 4 °C) and incubated overnight with anti-Flag beads (Millipore) at 4 °C with gentle rotation. The immunoprecipitated protein complexes were washed five times with lysis buffer (20 mM Tris-HCl (pH 8.0), 500 mM NaCl, 10% glycerol and 1 mM EDTA) and eluted with Laemmli buffer supplemented with 10% β-mercaptoethanol. Samples were then boiled at 95 °C for 5 min and analyzed by immunoblotting. Densitometric quantification of immunoblotting experiments was performed with ImageJ software.

Cellular fractionation experiments were performed using the Dignam method, as described in Simonet et al.[42]. For size-exclusion chromatography analysis, nuclei from HAFTL cells were purified and lysed under native conditions according to the Dignam method, as previously described[42]. Nuclear lysates were incubated overnight with benzonase nuclease before clarification and concentration with Amicon Ultra centrifugal filters (Millipore). Concentrated nuclear lysates were then fractionated by molecular weight on the gel filtration column Superose 6 (Cytiva) with a fractionation range of $5 \times 10^3$–$5 \times 10^6$ Da. The eluted fractions containing size-excluded proteins and protein complexes were denatured in Laemmli buffer supplemented with 10% β-mercaptoethanol and analyzed by immunoblotting. The following antibodies were used for immunoblotting: anti-SIRT7 (D3K5A, Cell Signaling, 1:1,000), anti-Pax5 (D19F8, Cell Signaling, 1:1,000), anti-H3 (ab1791, Abcam, 1:1,000), anti-fibrillarin (B1, Santa Cruz Biotechnology,

1:1,000), anti-acetyl-lysine (9814, Cell Signaling, 1:200), anti-Flag (M2, Sigma-Aldrich, 1:10,000), anti-HA (6908, Sigma), anti-Myc (9B11, Cell Signaling, 1:1,000), anti-V5 (ab9116, Abcam, 1:1,000) and anti-actin (A1978, Sigma, 1:5,000).

### SIRT7 and Pax5 purification and in vitro deacetylation assay
*SIRT7*$^{-/-}$ HEK293F cells were transiently transfected with vectors encoding Pax5-Myc–Flag or SIRT7–Flag for 48 h. Before collection, cells expressing these constructs were treated overnight with 5 mM nicotinamide and for 3 h with 1 μM trichostatin A to hyperacetylate Pax5. Cell pellets were lysed, incubated with benzonase nuclease for 8 h, clarified and incubated overnight with anti-Flag beads (Millipore). The immunoprecipitated protein complexes were washed five times with BC500 buffer (20 mM Tris-HCl (pH 8.0), 500 mM NaCl, 10% glycerol and 1 mM EDTA), eluted with synthetic Flag peptide (0.6 μg ml$^{-1}$; GenScript) and dialyzed in BC100 buffer (20 mM Tris-HCl (pH 8.0), 100 mM NaCl, 10% glycerol and 1 mM EDTA). Purified PAX5 and SIRT7 proteins were incubated for 1 h at 37 °C, with or without 1.25 mM NAD$^+$, in deacetylation buffer (10 mM Tris-HCl (pH 8.0), 150 mM NaCl, 1 mM DTT and 10% glycerol), and the reaction was stopped with 5× Laemmli buffer containing 10% β-mercaptoethanol. PAX5 acetylation was determined by immunoblotting using anti-pan-acetyl-lysine.

### Pre-B cell proteome and Pax5 acetylation analysis
For determination of the pre-B cell proteome, pre-B cells were sorted from the BM of wild-type and *Sirt7*$^{-/-}$ mice. Proteins were extracted with 6 M urea, 100 mM Tris (pH 8.0) and the help of a bioruptor, quantified using a NanoDrop at 280 nm and precipitated with trichloroacetic acid/acetone. Samples were then reduced and alkylated with 10 mM DTT and 55 mM chloroacetamide, respectively. Proteins were then resuspended in 6 M urea and 100 mM Tris (pH 8.0) and digested with LysC/trypsin. LysC digestion was performed for 16 h, while trypsin was added for 8 h, and both reactions were performed at 30 °C. The reactions were stopped with 10% formic acid. The peptides were then desalted with a C18 reverse-phase ultramicrospin column and desiccated in a speedvac. Total proteome samples were separated using a C18 analytical column (nanoEaseTM M/Z HSS C18 T3; 75 μm × 25 cm, 100 Å; Waters) with a 180-min run comprising three consecutive steps with linear gradients from 3% to 35% B in 150 min, from 35% to 50% B in 5 min and from 50% to 85% B in 2 min, followed by isocratic elution at 85% B in 5 min and stabilization to initial. The mass spectrometer was operated in data-dependent acquisition mode, and the data were acquired with Xcalibur software 4.0.27.10 (Thermo Scientific).

For identification of acetylated Pax5 residues, PAX5 was purified from *SIRT7*$^{-/-}$ HEK293F cells transiently expressing Pax5-Myc–Flag together with an empty vector or a vector encoding SIRT7–Flag. PAX5-containing beads were washed three times with 100 mM Tris (pH 8.0) and then resuspended in 6 M urea and 100 mM Tris (pH 8.0). Reduction and alkylation were then performed by using 10 mM and 55 mM chloroacetic acid. The digestion was performed by adding 1 μg of trypsin for 16 h at 30 °C. Finally, the digestion was stopped with 10% formic acid, and the peptides were desalted with a polyLC C18 pipette tip and dried in a speedvac. The acetylomes were separated using an Evosep EV1000 column (150 μm × 150 mm, 1.9 μm; Evosep) with an 88-min run. The spectrometer was working in positive polarity mode, and single-charge state precursors were rejected for fragmentation. The data were acquired with Xcalibur software 4.2.28.14 (Thermo Scientific).

For both total proteome and acetylome analyses, the peptides were reconstituted with 3% acetonitrile and 0.1% formic acid aqueous solution at 100 ng μl$^{-1}$, and 800 ng was injected into the mass spectrometer.

### Semiquantitative PCR, RT−qPCR and RNA-seq
Semiquantitative PCR of *Igh* segments was performed as described in Ng et al.[43] using genomic DNA extracted from sorted splenic IgM$^+$

cells and degenerate primers. RNA for RNA-seq and RT–qPCR was extracted from frozen pellets using a Maxwell RSC simplyRNA Tissue kit (Promega). For RT–qPCR, cDNA was synthesized using a Transcriptor First Strand cDNA Synthesis kit (Roche) according to manufacturer's instructions. RT–qPCR reactions were performed in a QuantStudio 5 Real-Time PCR System. Primer sequences (Integrated DNA Technologies) are shown in Extended Data Table 1.

RNA for RNA-seq was extracted from sorted wild-type and *Sirt7*[−/−] BM pro-B cells and pre-B cells or *Pax5*[−/−] ex vivo-expanded B cell progenitors expressing an empty vector, PAX5-WT, PAX5-K198Q or PAX5-K198R. After library construction, 150-bp paired-end sequencing was performed on a DNBSEQ-G400 (MGI Tech).

### ChIP–seq

Chromatin was prepared using a truChIP Chromatin Shearing kit (Covaris) with modifications. Ten million pro-B cells were collected and washed once in PBS before fixation in 1 mg ml$^{-1}$ DSG (Thermo Fisher) for 30 min, followed by an additional 10-min incubation with 1% formaldehyde. The cross-linking reaction was stopped by adding 1/20 quenching buffer, and fixed cells were washed twice with ice-cold 0.5% bovine serum albumin (wt/vol) in PBS. Nuclei were purified and sonicated following the manufacturer's instructions. One volume of 2× dilution buffer supplemented with 0.1% SDS and protease inhibitors was added before clarifying the lysate (10,000*g*, 5 min, 4 °C). ChIP was performed overnight at 4 °C with 10 µg of anti-Pax5 (ab183575, Abcam) previously conjugated with 20 µl of Pierce ChIP-grade Protein A/G Magnetic Beads (Thermo Fisher) for 4 h at 4 °C. After immunoprecipitation, samples were serially washed once with low-salt wash buffer (0.1% SDS, 1% Triton, 2 mM EDTA, 20 mM Tris-HCl (pH 8.1) and 150 mM NaCl), high-salt wash buffer (0.1% SDS, 1% Triton, 2 mM EDTA, 20 mM Tris-HCl (pH 8.1) and 500 mM NaCl) and LiCl immune complex wash buffer (10 mM Tris-HCl (pH 8), 250 mM LiCl, 1% NP-40, 1% sodium deoxycholate and 1 mM EDTA) supplemented with protease inhibitors and twice with modified TE buffer (0.1 mM EDTA and 10 mM Tris-HCl (pH 8)). Cross-linking was reversed in elution buffer (0.1 M NaHCO$_3$ and 1% SDS) with 10 µg of RNase A (Thermo Fisher) for 30 min at 37 °C, followed by an additional 6-h incubation at 65 °C with 50 µg of Proteinase K (Apollo Scientific). DNA was subsequently cleaned up using NucleoSpin Gel and PCR clean-up columns (Macherey-Nagel) before library construction and 100-bp paired-end sequencing on a DNBSEQ-G400.

### Public data analysis

Mouse and human BM 10x scRNA-seq data were derived from the Broad Institute Single Cell Portal (https://singlecell.broadinstitute.org; projects 'Bone Marrow from B6 Mice, 10x' and 'A Census of Immune Cells', respectively). For human scRNA-seq analysis, the associated Loom file was downloaded and reanalyzed using Scanpy[44]. Cell-type annotation was performed using DecoupleR (v1.34)[45] and PanglaoDB[46]. Mouse scRNA-seq feature and *t*-distributed stochastic neighbor embedding plots were directly downloaded from the Single Cell Portal. Microarray normalized data from mouse B cell progenitors were from the Immgen Consortium datasets[47] (accession number GSE15907). Pax5-regulated genes were retrieved from accession number GSE38046 (ref. 20). Pax5 peaks were obtained from a publicly available PAX5 ChIP–seq dataset[17]. The list of mammalian proteins with acetyltransferase activity was from a previous report[31], whereas Pax5 protein-protein interactions (PPIs) combined the data from another study[19] with the curated interactions compiled in the BioGRID repository. Normalized proteomics and RNA-seq data for individuals with B-ALL were from Yang et al.[29]. Proteomics and RNA-seq raw data from this project are available at the ProteomeXchange Consortium (identifier PXD010175) and at the European Genome–Phenome Archive (accession EGAS00001003079), respectively. Relative Pax5 and SIRT7 protein levels in B-CLL samples were obtained from

Johnston et al.[34], and raw data are deposited at the PRIDE archive (PXD002004). Children's Oncology Group clinical trial P9906 RNA expression and outcome data are available at the Genomic Data Commons (https://portal.gdc.cancer.gov) and were generated by the Therapeutically Applicable Research to Generate Effective Treatments (https://www.cancer.gov/ccg/research/genome-sequencing/target; GSE11877) initiative, whereas *PAX5* deletion data in these individuals were retrieved from Roberts et al.[37] (EGAS00001000654).

### Proteomics data analysis

The raw thermo files were processed with MaxQuant 1.6.7.0 using a mouse database downloaded from https://www.uniprot.org/ (December 2018) or with PEAKS X+ software using a mouse database downloaded from https://www.uniprot.org/ (November 2019) for the total proteome and acetylome, respectively. In both cases, only reviewed entries were included (16,997 or 17,024 entries, respectively). The search was performed using the following parameters: trypsin was selected as the enzyme, and a maximum of two or three missed cleavages was allowed for the total proteome and acetylome. For the total proteome, the modifications were carbamidomethylation as a fixed modification, whereas oxidation in methionines and acetylation of protein N termini were used as variable modifications. The iBAQ intensity was used to quantify the proteins. Alternatively, for the acetylome, carbamidomethylation was set as a fixed modification, whereas oxidation in methionines; acetylation at both lysines and the protein N terminus; deamidation at asparagine and glutamine; phosphorylation at serines, threonines and tyrosines and dehydration at aspartic acid, tyrosines, threonines, serines, glutamine and asparagine were used as variable modifications. The mass tolerance for the parental ion and MS/MS fragments were set to 10 ppm and 0.5 Da. In both analyses, the results were filtered at 1% FDR at the peptide and protein levels.

Data processing and statistical analyses were performed using R (https://cran.r-project.org/) and RStudio (https://www.rstudio.com/) software. Total proteome statistical analysis was performed with the 'limma' package. For the PAX5 acetylome, the peptide spectrum matches were calculated by using a homemade algorithm and the PEAKS' output files 'peptide.csv' and 'DB search psm.csv'.

### Bulk RNA-seq bioinformatic analysis

Bulk RNA-seq raw FastQ files were quality checked using FastQC, and raw counts were obtained by performing Salmon[48] 'quant' pseudoalignment (mm10 reference mouse genome), with fragment-level GC bias correction (–gcBias), eight threads (-p 8) and selective alignment enabled (–validateMappings). The quant.sf files were used to import transcript-level quantification data into a Summarized Experiment using Tximeta[49]. The Summarized Experiment was imported into an R environment, and differential gene expression analysis was performed using DESeq2 (ref. 50). Counts were normalized by DESeq2's median of ratios method[51], and normalized counts were transformed to *z* score for data visualization (ggplot2). For generating BigWig files, raw FastQ files were mapped onto a reference mouse genome (mm10) using Bowtie2 (ref. 52) to generate SAM files. SAM files were converted to BAM files using SAMTools[53], skipping alignments with MAPQ smaller than 37 (samtools view -bS -q 37). BAM files were sorted and filtered using Sambamba[54], eliminating unmapped and duplicated reads (-F '[XS] == null and not unmapped and not duplicate'). Index BAI files were generated using SAMTools 'index', and BigWig files were created using deepTools[55] 'bamCoverage' (–binSize 20–normalizeUsing BPM–ignoreForNormalization chrX–extendReads 150–centerReads–smoothLength 60). Visualization of BigWig files was performed using IGV[56].

GSEA was performed using the software[57] provided by the Broad Institute at https://www.gsea-msigdb.org/ with RNA-seq normalized count values and default parameters. Gene Ontology terms were obtained with the Enrichr tool[58].

## Article

## ChIP–seq bioinformatic analysis

ChIP–seq raw FastQ files were quality checked using FastQC and aligned onto a reference mouse genome (mm10) using Bowtie2 (ref. [52]) to generate SAM files. SAM files were converted to BAM files using SAMTools[53], skipping alignments with MAPQ smaller than 37 (samtools view -bS -q 37). BAM files were sorted and filtered using Sambamba[54], eliminating unmapped and duplicated reads (-F '[XS] == null and not unmapped and not duplicate'). Index BAI files were generated using SAMTools 'index', and BigWig files were created using deepTools[55] 'bamCoverage' (−binSize 20 −normalizeUsing BPM −ignoreForNormalization chrX −extendReads 150 −centerReads −smoothLength 60). Visualization of BigWig files was performed using IGV[56]. Peak calling was performed using MACS2 (ref. [59]) 'callpeak' (-f BAM −nomodel −extsize 20 -g mm -B) taking into account both immunoprecipitation (-t) and input (-c) samples for each peak calling. Only significant peaks ($q < 0.05$) were included in downstream analyses. Differential peak analysis was performed with ChIPpeakAnno, and heat maps were generated with DiffBind or plotHeatmap.

## Statistics and reproducibility

All statistical analyses were performed using GraphPad Prism 8.0.1. The statistical tests performed for data analysis are indicated in the corresponding figure legends. Individual $P$ values of significant data are indicated in the figures. Results were similarly replicated in at least two independent experiments, and results pooled from independent experiments are indicated in the corresponding figure legends. For experiments involving mice, each replicate corresponds to an individual mouse, and no animals or data points were excluded. RNA-seq, ChIP–seq and proteomics replicates were generated on different days. No statistical methods were used to predetermine sample sizes. Data distribution was assumed to be normal, but this was not formally tested. Only recipient mice for transplantation experiments were randomized because the rest of the experiments required mouse genotyping and were performed on genetically identical mice. Data collection and analysis were not performed blind to the conditions of the experiments.

## Reporting summary

Further information on research design is available in the Nature Portfolio Reporting Summary linked to this article.

## Data availability

All Pax5[WT], Pax5[K198Q] and Pax5[K198R] ChIP–seq data and RNA-seq data from wild-type and *Sirt7*[-/-] BM pro-B and pre-B cells; *Pax5*[-/-] B cell progenitors retrovirally expressing empty vector, Pax5[WT], Pax5[K198Q] or Pax5[K198R] and wild-type or *Sirt7*[-/-] B cell progenitors expressing empty vector, Pax5[WT] or Pax5[K198R] have been deposited at the National Center for Biotechnology Information Gene Expression Omnibus and are available under the accession number GSE246370. All sequencing data were aligned to the mm10 reference mouse genome. Proteomics data from wild-type and *Sirt7*[-/-] BM pre-B cells and Pax5 acetylation experiments have been deposited to the ProteomeXchange Consortium via the PRIDE partner repository with the dataset identifier PXD046457. Source data are provided with this paper.

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

## Acknowledgements

We are grateful to M. Parra and O. de Barrios (IJC) for providing B-ALL cell lines. We also thank the members of the laboratory of A.V. for stimulating discussions. This work was supported by a grant from the Scientific Foundation of the Spanish Association Against Cancer (AECC; grant number PROYE20042VAQU), the Spanish Ministry of Economy and Competitiveness (MINECO; SAF2017-88975R) and a grant PID2020-117284RB-I00 funded by MICIU/AEI/10.13039/501100011033 to A.V., the Spanish National Institute of Health Carlos III-ISCIII (ProteoRed) to A.V. and C.D.L.T., CP19/00176 to J.L.S. (cofunded by the European Social Fund 'Investing in your future'), FI-AGAUR FI_B 00293 to A.G.-G., the European Commission's Horizon research and innovation programme Marie Skłodowska-Curie (2020, 895979 to B.N.V.; 2021, 101065013 to A.I.), Beatriu de Pinos fellowship (2016-BP-00250 to B.N.V.), the Catalan Government Agency AGAUR (2017-SGR-148, 2021-SGR-01378 to A.V.; 2021-SGR-01494 to M.E.), the Human Genetics Institute of New Jersey (to L.S. and J.A.T.) and the Swedish Cancer Foundation (20-1153), the Childhood Cancer Foundation (2022-0019) and the Swedish Research Council (2021-02379) to M.S. Research at the laboratory of J.L.S. was supported by grant PID2019-111243RA-I00 funded by MICIU/AEI/10.13039/501100011033. The IJC Proteomics Unit is part of the Spanish Platform of Molecular and Bioinformatics Resources (ProteoRed), Instituto de Salud Carlos III (PT13/0001) and

Proyecto Proteoma del Cáncer Participación Española en el CPTAC Internacional, The National Cancer Institute's Clinical Proteomic Tumour Analysis Consortium. We also thank the CERCA Programme/Generalitat de Catalunya for institutional support.

## Author contributions

A.G.-G., B.N.V. and A.V. planned the project, designed the experiments and wrote the manuscript. A.V. and B.N.V. supervised the work, and M.S. provided critical reagents and key discussions on the experimental design and the manuscript. A.G.-G., M.E.-A., B.N.V., M.S., A.M.-D., E.A.-P., C.R., C.B., P.K., A.I. and L.S. performed experiments and/or analyzed data. A.G.-G., A.B.-C., J.L.S. and J.K.T. analyzed RNA-seq and ChIP–seq experiments. J.J.B. and C.D.L.T. performed and analyzed the proteomics experiments. M.E., L.S., J.L.S., A.I., T.B. and J.A.T. provided reagents and their expertise and participated in the discussion.

## Competing interests

The authors declare no competing interests.

## Additional information

**Extended data** is available for this paper at https://doi.org/10.1038/s41590-024-01995-7.

**Correspondence and requests for materials** should be addressed to Berta N. Vazquez or Alejandro Vaquero.

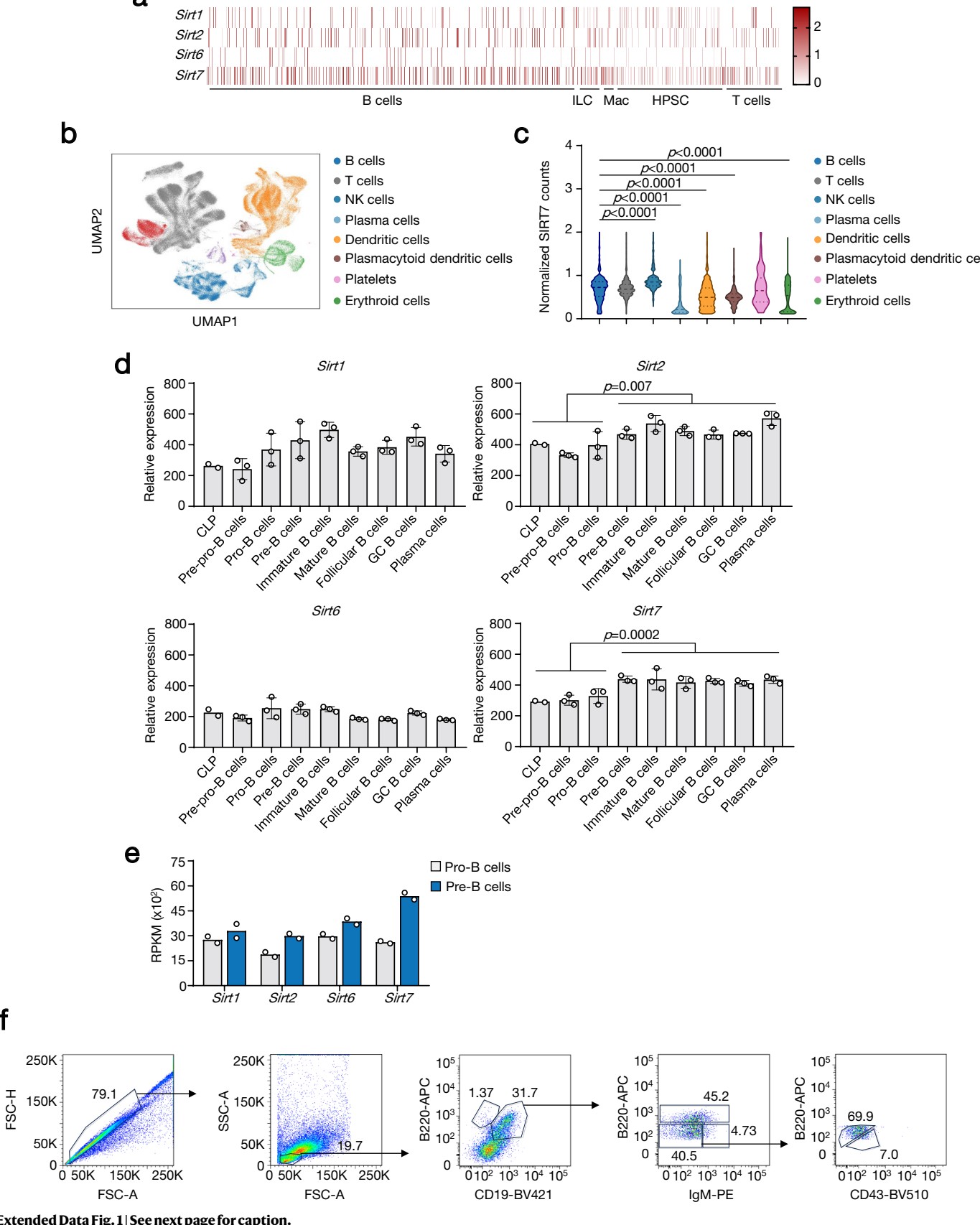

**Extended Data Fig. 1 | See next page for caption.**

**Extended Data Fig. 1 | SIRT7 is upregulated in B cell progenitors.**
**a**, Heatmap displaying the single-cell expression of *Sirt1*, *Sirt2*, *Sirt6* and *Sirt7* in the populations (B cells, innate lymphoid cells (ILCs), macrophages (Mac), hematopoietic and progenitor stem cells (HPSC) and T cells) annotated by sc-RNA-seq from purified murine BM Lin⁻ cells. Data was obtained from singlecell. broadinstitute.org (study SCP978). **b,c**, sc-RNA-seq feature plot (**b**) and single-cell expression of *Sirt7* (**c**) in B cells, T cells, NK cells, plasma cells, dendritic cells, plasmacytoid dendritic cells, platelets and erythroid cells of human BM. Data was obtained from singlecell.broadinstitute.org (study SCP101). In panel **c**, only those cells with detectable counts are plotted, and the dashed lines indicate the median and the first and fourth quartiles. Statistical significance was assessed by one-way ANOVA with Dunnet comparison. **d**, Microarray expression of *Sirt1* (top left), *Sirt2* (top right), *Sirt6* (bottom left) and *Sirt7* (bottom right) in B lymphopoiesis stages (n = 3; common lymphoid progenitors (CLP), n = 2). Data are presented as mean ± s.d. A two-tailed *t*-test was performed comparing the expression in pooled CLP–pro-B and in pooled pre-B–plasma cells. Data were obtained from the Immgen consortium (GSE15907). **e**, RNA-Seq expression patterns of *Sirt1*, *Sirt2*, *Sirt6* and *Sirt7* in pro-B cells and pre-B cells. Data are presented as the mean of the two replicates (n = 2). **f**, Gating strategy defining B cell subsets (B220⁺CD19⁻ pre-pro-B cells, B220⁺CD19⁺IgM⁻CD43⁺ pro-B cells, B220⁺CD19⁺IgM⁻CD43⁻ pre-B cells, B220⁺CD19⁺IgM⁺ immature B cells and B220^hiCD19⁺ mature B cells) for intracellular flow cytometry measurement of SIRT7 protein levels in fixed mouse BM.

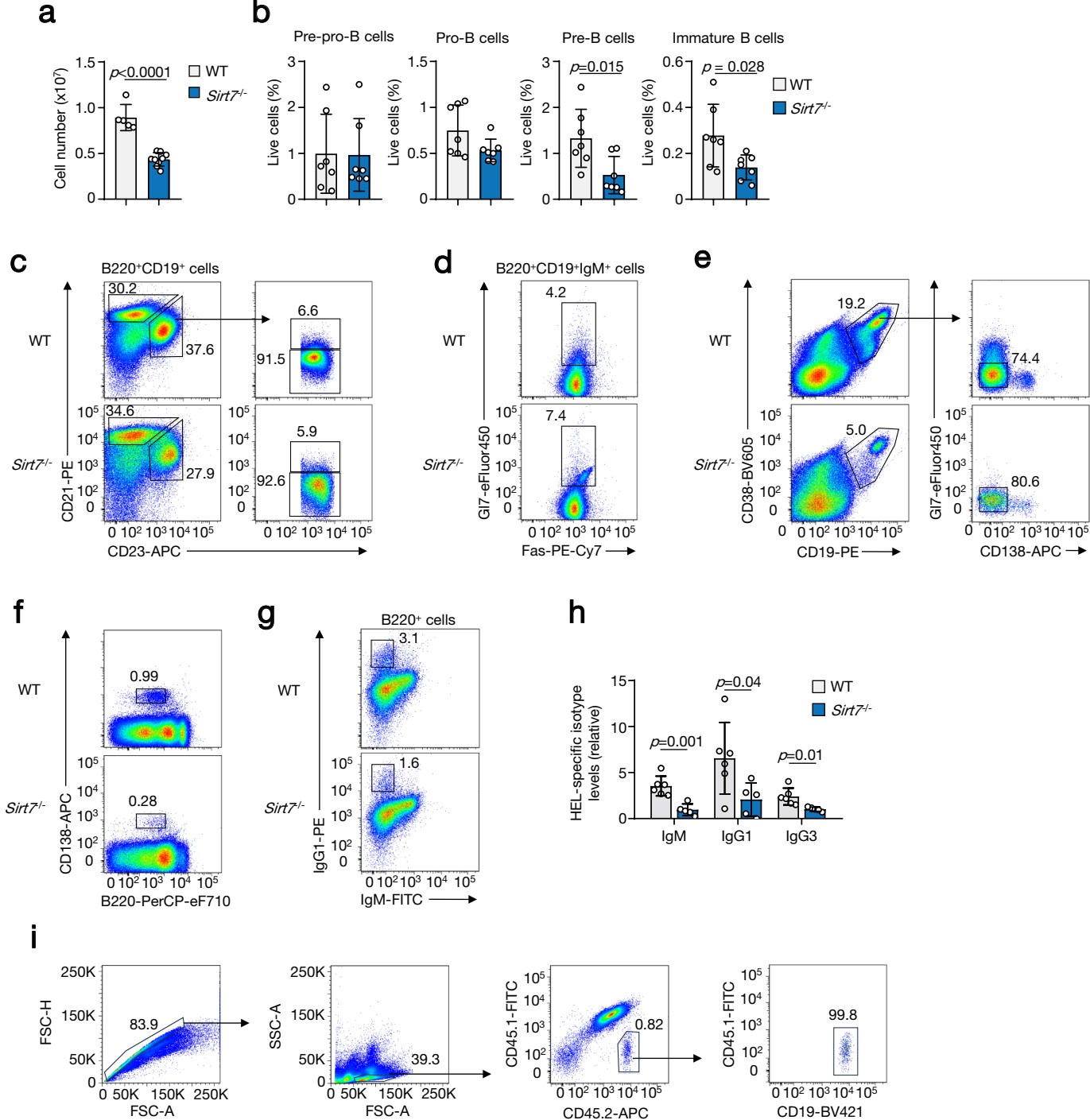

**Extended Data Fig. 2 | SIRT7 is required for normal B cell development.**
**a,b**, Number of B220⁺CD19⁺ B cells (**a**) and percentages of B220⁺CD19⁻ pre-pro-B cells, B220⁺CD19⁺IgM⁻CD43⁺ pro-B cells, B220⁺CD19⁺IgM⁻CD43⁻ pre-B cells and B220⁺CD19⁺IgM⁺ immature B cells (**b**) determined by flow cytometry in the BM of wild-type and $Sirt7^{\Delta4\text{-}9}$ C57BL/6 mice. Data are presented as mean ± s.d. and significance was assessed by two-tailed *t*-test (wild-type, n = 5 (**a**) and 7 (**b**); $Sirt7^{\Delta4\text{-}9}$, n = 9 (**a**) and 7 (**b**)). Pooled from two independent experiments. **c-g**, Representative FACS plots of splenic B220⁺CD19⁺CD21^high^CD23⁻ marginal zone B cells, B220⁺CD19⁺CD21⁺CD23⁺CD93⁻ follicular B cells and B220⁺CD19⁺CD21⁺CD23⁺CD93⁺ transitional B cells. (**c**), B220⁺CD19⁺IgM⁺Gl7⁺Fas⁺ germinal center B

cells (**d**), BM CD19⁺CD38⁺CD138⁻Gl7⁺ memory B cells (**e**), BM B220^lo^CD138⁺ plasma cells (**f**) and class-switched splenic B220⁺IgG1⁺ B cells (**g**) from wild-type and $Sirt7^{-/-}$ mice (n = 4). **h**, Levels of anti-HEL antibody isotypes in the sera of wild-type and $Sirt7^{-/-}$ mice 14 days after NP-HEL immunization, relative to naive mice, as determined by ELISA. Data are shown as in **a** (wild-type, n = 6; $Sirt7^{-/-}$, n = 5). One of two separate experiments is shown. **i**, Gating strategy used to identify donor-derived CD45.1⁻CD45.2⁺CD19⁺ mature B cells in the spleen of recipient CD45.1/CD45.2 mice four weeks after transplantation of CD45.2 wild-type or $Sirt7^{-/-}$ Lin⁻B220⁺CD19⁺IgM⁻ pro-B cells.

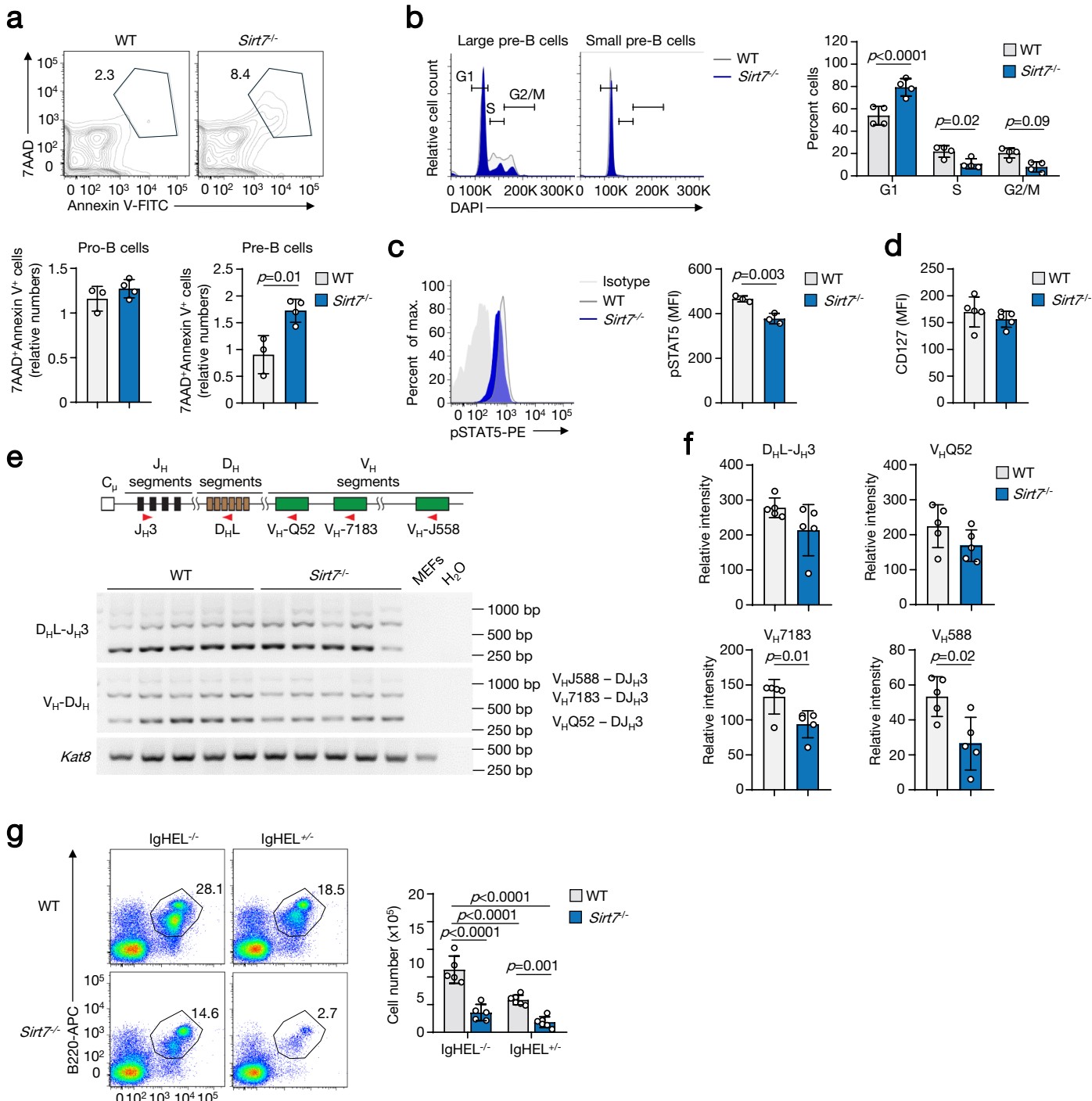

**Extended Data Fig. 3 | SIRT7 promotes B cell development independently of V(D)J. a**, Representative apoptosis FACS plots (top) and relative numbers (bottom) of 7AAD⁺Annexin V⁺ cells (gated on B220⁺CD19⁺IgM⁻CD43⁺ pro-B and B220⁺CD19⁺IgM⁻CD43⁻ pre-B cells) in the BM of wild-type (n = 3) and *Sirt7⁻/⁻* (n = 4) mice. Data was normalized to wild-type values. **b**, Representative histograms (left) and quantification (right) of the percentages of cells in G1, S and G2/M cell cycle stages of gated wild-type and *Sirt7⁻/⁻* B220⁺CD19⁺IgM⁻CD43⁻FSCʰⁱ large and B220⁺CD19⁺IgM⁻CD43⁻FSCˡᵒ small pre-B cells, determined by intracellular flow cytometry (n = 4 mice). Pooled from two separate experiments. **c**, Representative histogram (left) and p-STAT5^Y694 MFI (right) in gated wild-type and *Sirt7⁻/⁻* B220⁺CD19⁺IgM⁻CD43⁻ pre-B cells, measured by intracellular flow cytometry (n = 3 mice; p = 0.005). Pooled from three separate experiments **d**, Flow cytometric quantification of CD127 expression in gated B220⁺CD19⁺IgM⁻CD43⁻ pre-B cells from the BM of wild-type and *Sirt7⁻/⁻* mice (n = 5). Pooled from three

separate experiments. **e**, Schematic representation of the mouse *IgH* locus, showing constant (C), joining (J), diversity (D) and variable (V) gene segment organization (top) and semi-quantitative genomic DNA PCR using degenerate primers targeting DₕL-Jₕ3 and Vₕ-DJₕ gene segments in IgM⁺ B cells sorted from the BM of wild-type and *Sirt7⁻/⁻* mice. Mouse embryonic fibroblasts (MEFs) and H₂O were used as negative controls of PCR amplification. The independent locus *Kat8* was used as a loading control (n = 5 mice). **f**, Densitometric quantification of DₕL-Jₕ3, VₕJ588-DJₕ3, Vₕ7183-DJₕ3 and VₕQ52-DJₕ3 gene segment amplification in the mouse *IgH* locus normalized to *Kat8* in sorted splenic wild-type and *Sirt7⁻/⁻* IgM⁺ B cells (n = 5 mice). **g**, Representative flow cytometry plots (left) and number of B220⁺CD19⁺ B cells (right) in the BM of IgHEL⁻/⁻ (n = 5) IgHEL⁻/⁻ *Sirt7⁻/⁻* (n = 5), IgHEL⁺/⁻ *(n = 6)* and IgHEL⁺/⁻*Sirt7⁻/⁻* (n = 6) mice. Data in **a, b, c, d, f** and **g** are shown as mean ± s.d. Data in **a, c** and **f** were analyzed using one-tailed t-test, and data in **b** and **g** were analyzed using two-way ANOVA with Sidak multiple comparisons.

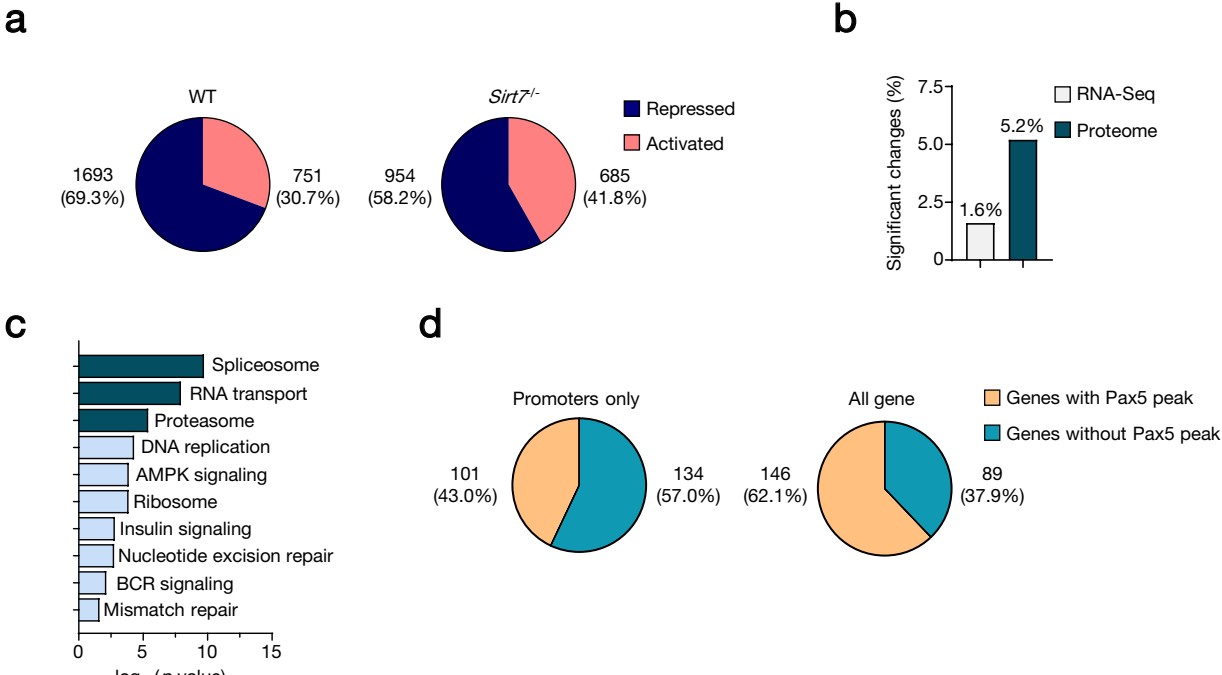

**Extended Data Fig. 4 | SIRT7 regulates Pax5 in B cell progenitors. a**, Pie charts showing the number and percentage of genes significantly induced and repressed ($|\log_2(\text{fold-change})| \geq 1.5$, FDR < 0.05) during the pre-B-to-pro-B cell transition in wild-type (left) and *Sirt7*[-/-] (right) mice (n = 2). **b**, Percentages of significantly regulated ($|\log_2(\text{fold change})| \geq 1$, FDR < 0.1) transcripts and proteins in *Sirt7*[-/-] vs wild-type B220[+]CD19[+]IgM[-]CD43[-] pre-B cells, relative to all the transcripts and proteins detected by RNA-Seq and proteomics, respectively.

**c**, Gene ontology analysis of significantly regulated proteins in *Sirt7*[-/-] vs wild-type B220[+]CD19[+]IgM[-]CD43[-] pre-B cells. **d**, Percentage of Cluster 3 genes from wild-type and *Sirt7*[-/-] B220[+]CD19[+]IgM[-]CD43[-] pre-B cell and B220[+]CD19[+]IgM[-]CD43[+] pro-B cell bulk RNA-Seq data with significant ($q$ < 0.05) Pax5 peaks[17] in their promoters only (±2Kb from TSS) (left panel) or in the whole gene (including ± 2Kb from TSS, gene bodies and 3'UTR, right panel).

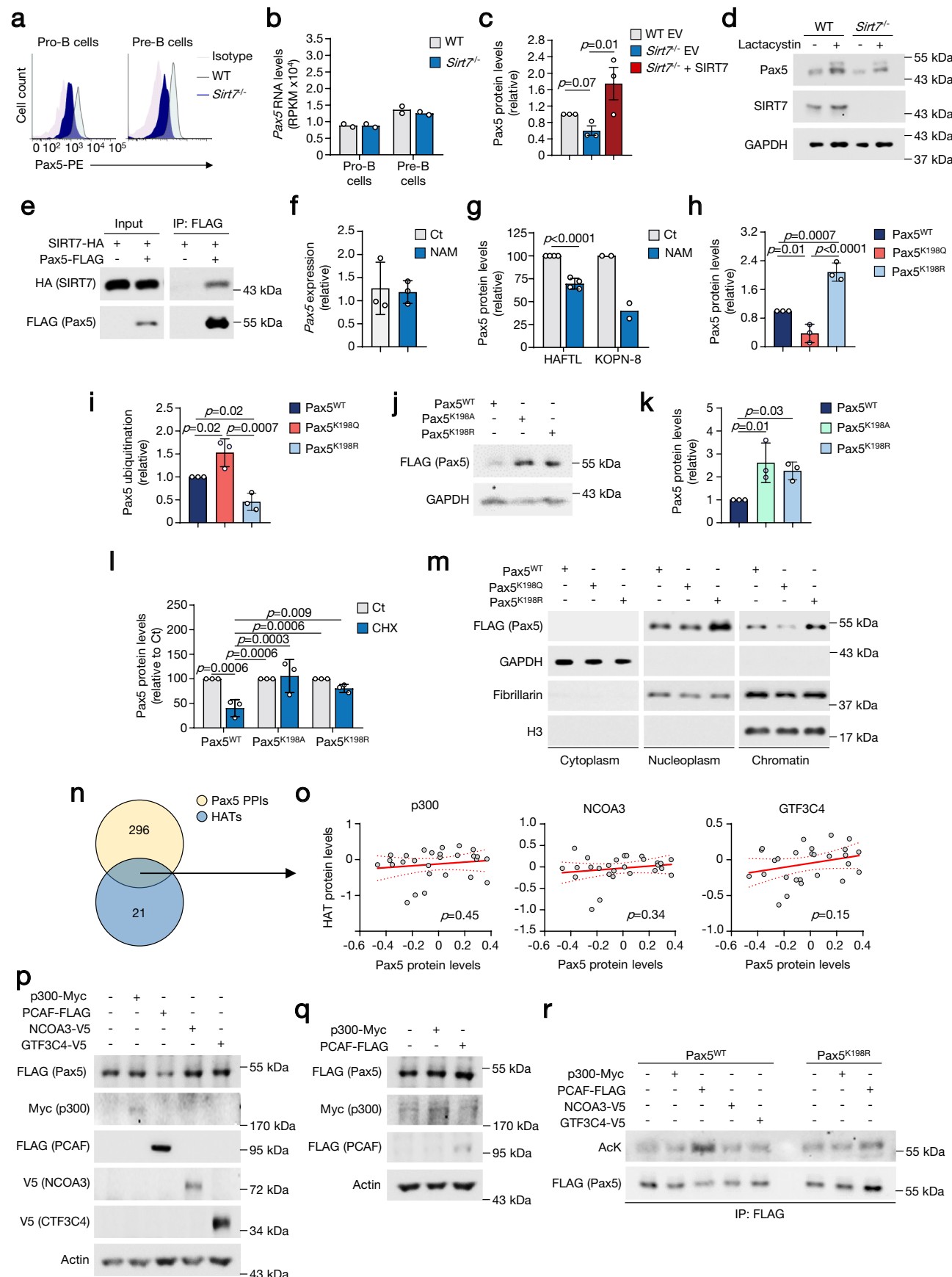

**Extended Data Fig. 5 | See next page for caption.**

**Extended Data Fig. 5 | SIRT7 regulates Pax5 stability by deacetylating K198. a**, Representative histograms of Pax5 expression in gated B220$^+$CD19$^+$IgM$^-$CD43$^+$ pro-B and B220$^+$CD19$^+$IgM$^-$CD43$^-$ pre-B cells from the BM of wild-type and Sirt7$^{-/-}$ mice (wild-type, n = 6; Sirt7$^{-/-}$, n = 8). Pooled from three separate experiments **b**, RNA-Seq analysis of Pax5 expression in sorted pro-B and pre-B cells defined as in **a**. Data are shown as mean of two replicates (n = 2). **c**, Pax5/GAPDH ratio in B220$^+$CD19$^+$IgM$^-$CD43$^+$ pro-B cells retrovirally expressing empty vector (EV) and SIRT7 and expanded ex vivo for four days with OP9 cells and 10 ng/ml IL-7, SCF and FLT3-L (n = 3). Pooled from 3 separate experiments. **d**, Immunoblots of Pax5 and SIRT7 protein from wild-type and CRISPR-Cas9-generated HAFTL$^{SIRT7KO}$ pre-B cells treated with vehicle (Ct) or 2 µM lactacystin for 8 h. **e**, FLAG-specific immunoprecipitation from HEK293F cells transiently co-expressing SIRT7-HA with EV or Pax5-FLAG using polyethyleneimine transfection, followed by FLAG and HA immunoblotting. **f**, RT-qPCR analysis of Pax5 expression relative to Hprt in HAFTL cells treated with vehicle (Ct) or 5 mM nicotinamide (NAM) for 48 h (n = 3). **g**, Pax5/H3 ratio in HAFTL (n = 4 biological replicates; p < 0.0001) and KOPN-8 (n = 2 biological replicates) cells treated as in **f**. **h,i**, FLAG-tagged Pax5$^{WT}$, Pax5$^{K198Q}$ and Pax5$^{K198R}$ levels (**h**, n = 3) and ubiquitination (**i**, n = 3). Expressed in

HEK293F cells as in **e**. Pooled from three separate experiments. **j,k**, Immunoblots (**j**) and quantification (**k**) of Pax5$^{WT}$, Pax5$^{K198A}$ and Pax5$^{K198R}$ forms expressed as in **e**. Data in **k** are pooled from three separate experiments. **l**, Relative levels of Pax5$^{WT}$, Pax5$^{K198A}$ and Pax5$^{K198R}$ forms in HEK293F cells treated with vehicle (Ct) or 100 µg/mL cycloheximide (CHX) for 8 h (n = 3, pooled from three separate experiments). **m**, Subcellular fractionation of HEK293F cells expressing Pax5$^{WT}$, Pax5$^{K198Q}$ and Pax5$^{K198R}$ forms as in **e**. GAPDH, Fibrillarin and H3; cytoplasmic, nuclear and chromatin controls. **n**, Venn diagram depicting the overlap between Pax5 protein–protein interactions (PPIs) and mammalian histone acetyltransferases[31] (HATs) (see methods). **o**, Scatter plots showing the correlation between the protein levels of Pax5, p300, NCOA3 and GTF3C4, as determined by proteomics in human B-ALL patients samples[29] (n = 27). Each point corresponds to one sample. Linear regression and 95% confidence intervals (dashed lines) are shown. **p-r**, Immunoblots depicting the expression (**p,q**) and acetylation levels (**r**) of Pax5$^{WT}$ and Pax5$^{K198R}$ proteins from HEK293F transiently co-expressing Pax5$^{WT}$ or Pax5$^{K198R}$ together with p300, PCAF, NCOA3 or GTF3C4. Data in **c**, **f**, **g**, **h**, **i** and **k** are shown as mean ± sd. Data in **c**, **h**, **l**, **k** were analyzed using one-way ANOVA with Fisher's LSD test, and data in **g** (HAFTL) were analyzed using two-tailed t-test.

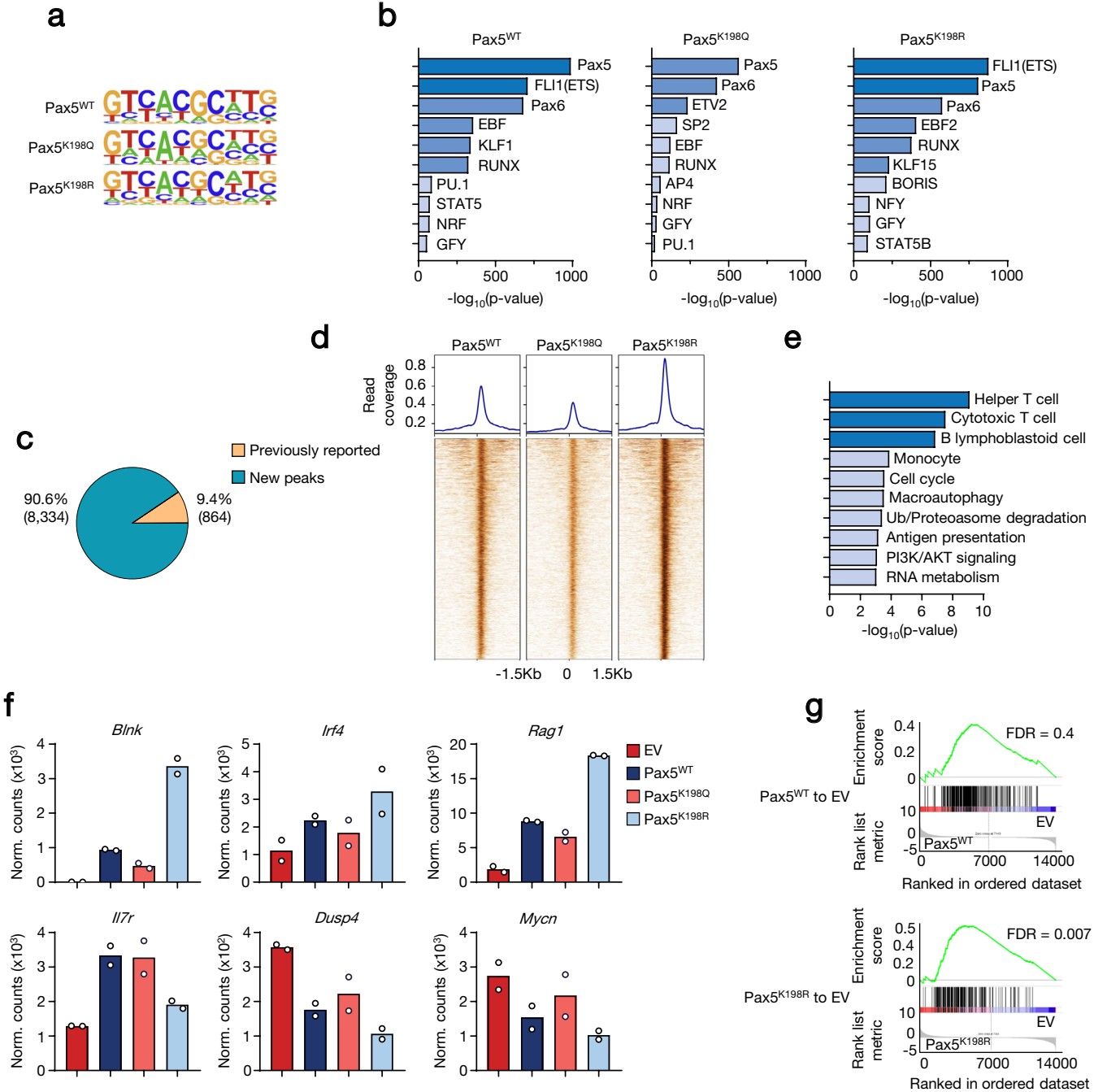

**Extended Data Fig. 6 | Pax5$^{K198}$ acetylation regulates gene expression. a**, DNA binding motifs of Pax5$^{WT}$, Pax5$^{K198Q}$ and Pax5$^{K198R}$ in Pax5$^{-/-}$ Lin$^-$B220$^+$IgM$^-$ pro-B cells retrovirally transduced with Pax5$^{WT}$, Pax5$^{K198Q}$ and Pax5$^{K198R}$ forms and expanded *ex vivo* for seven days with OP9 cells and 10 ng/ml IL-7, SCF and FLT3-L. **b**, Motif enrichment analysis of the significant peaks ($q < 0.05$) detected by ChIP-Seq of Pax5$^{WT}$, Pax5$^{K198Q}$ and Pax5$^{K198R}$ as in **a**. **c**, Comparison of the significant peaks bound by Pax5$^{K198R}$ as in **a** and those reported in Ref. 17. **d**, Genomic occupancies of Pax5$^{WT}$, Pax5$^{K198Q}$ and Pax5$^{K198R}$ forms in the unique Pax5$^{K198R}$ peaks.

Top panels, read coverage profiles. **e**, Gene ontology terms of the unique Pax5$^{K198R}$ peaks. Only terms with -log$_{10}$($p$-value)>3 are reported. **f**, RNA-Seq analysis of the expression of *Blnk*, *Irf4*, *Rag1*, *Il7r*, *Dusp4* and *Mycn* the indicated genes in Lin$^-$B220$^+$IgM$^-$ *Pax5$^{-/-}$* pro-B cells expressing an empty vector (EV), Pax5$^{WT}$, Pax5$^{K198Q}$ and Pax5$^{K198R}$ and expanded as in **a**. Data are presented as mean (n = 2). **g**, 'HALLMARK_E2F_TARGETS' GSEA of RNA-seq data from Pax5$^{WT}$ versus EV (upper) pro-B cells and PAX5$^{K198R}$ versus EV (lower) pro-B cells. FDR, false discovery rate.

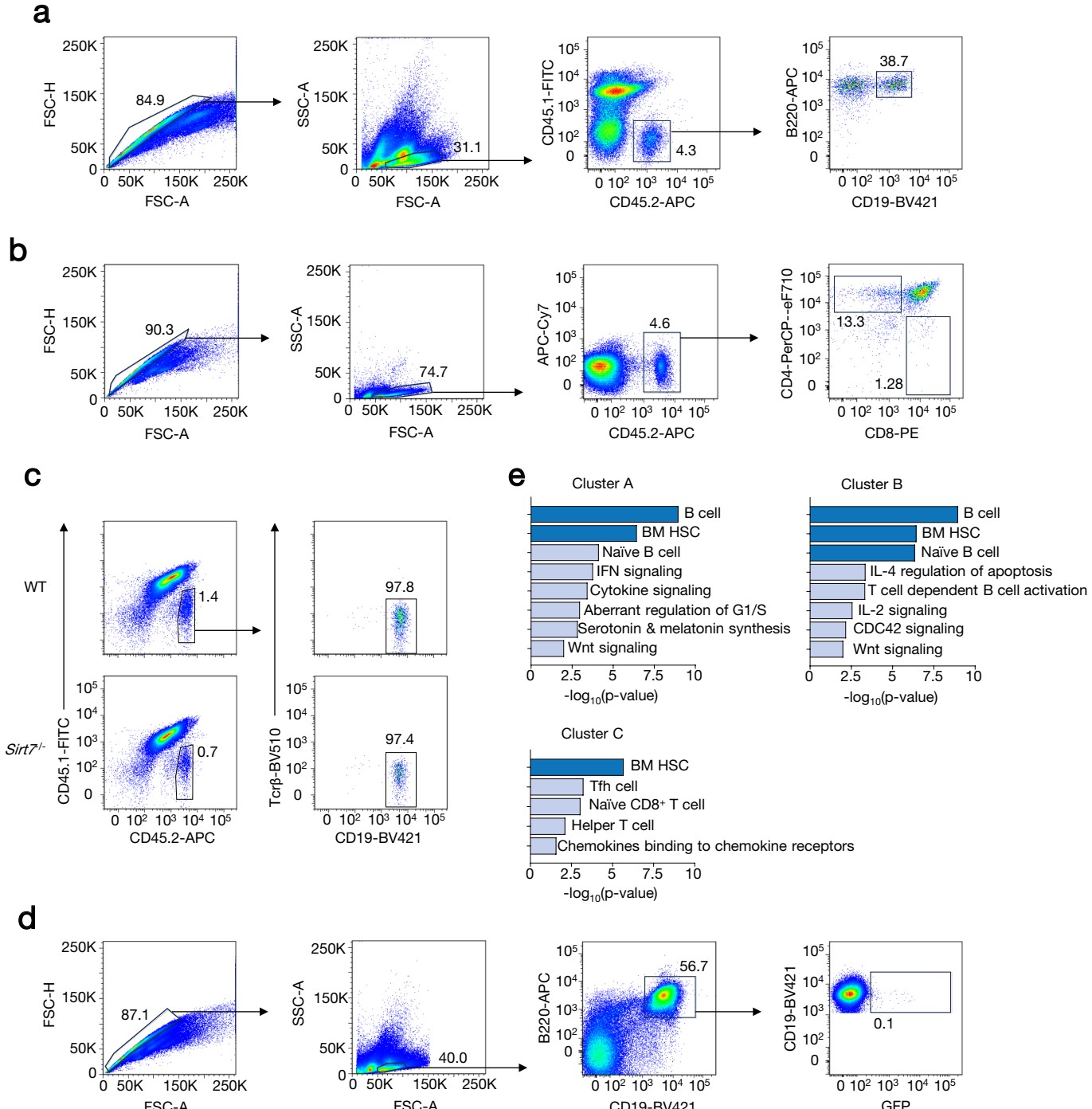

**Extended Data Fig. 7 | Pax5$^{K198}$ deacetylation regulates B cell identity.**
**a,b**, Gating strategies used to identify CD45.2$^+$B220$^+$CD19$^+$ B cells (**a**) or
CD45.2$^+$CD4$^+$ and CD45.2$^+$CD8$^+$ T cells (**b**) derived from donor CD45.2 *Pax5*$^{-/-}$
Lin$^-$B220$^+$IgM$^-$ pro-B cell expressing empty vector (EV), Pax5$^{WT}$, Pax5$^{K198Q}$ and
Pax5$^{K198R}$ by retroviral transduction in the bone marrow (**a**) or thymus (**b**)
of CD45.1 recipient mice 4 weeks after transplantation. **c**, Representative
histograms of the donor derived CD45.1$^-$CD45.2$^+$CD19$^+$ B cells in the spleens
of recipient CD45.1/CD45.2 mice 6 weeks after injection of wild-type or *Sirt7*$^{-/-}$

Lin$^-$IgM$^+$IgD$^+$ splenic B cells. **d**) Gating strategy used to identify donor-derived
B220$^+$CD19$^+$GFP$^+$ B cells in the spleens of CD45.1/CD45.2 recipient mice four
weeks after transplantation of *ex vivo* expanded (with OP9 cells and 10 ng/ml IL-7,
SCF and FLT3-L, for 7 days) wild-type and *Sirt7*$^{-/-}$ Lin$^-$B220$^+$CD19$^+$GFP$^+$ pro-B cells
retrovirally transduced with empty vector (EV, pMIG), Pax5$^{WT}$ or Pax5$^{K198R}$. **e**, Gene
ontology analysis of the genes in Clusters A, B and C generated by unsupervised
clustering analysis of bulk RNA-seq data from the same cells as in **d**. Only terms
with -log$_{10}$(p-value) > 1.5 are reported.

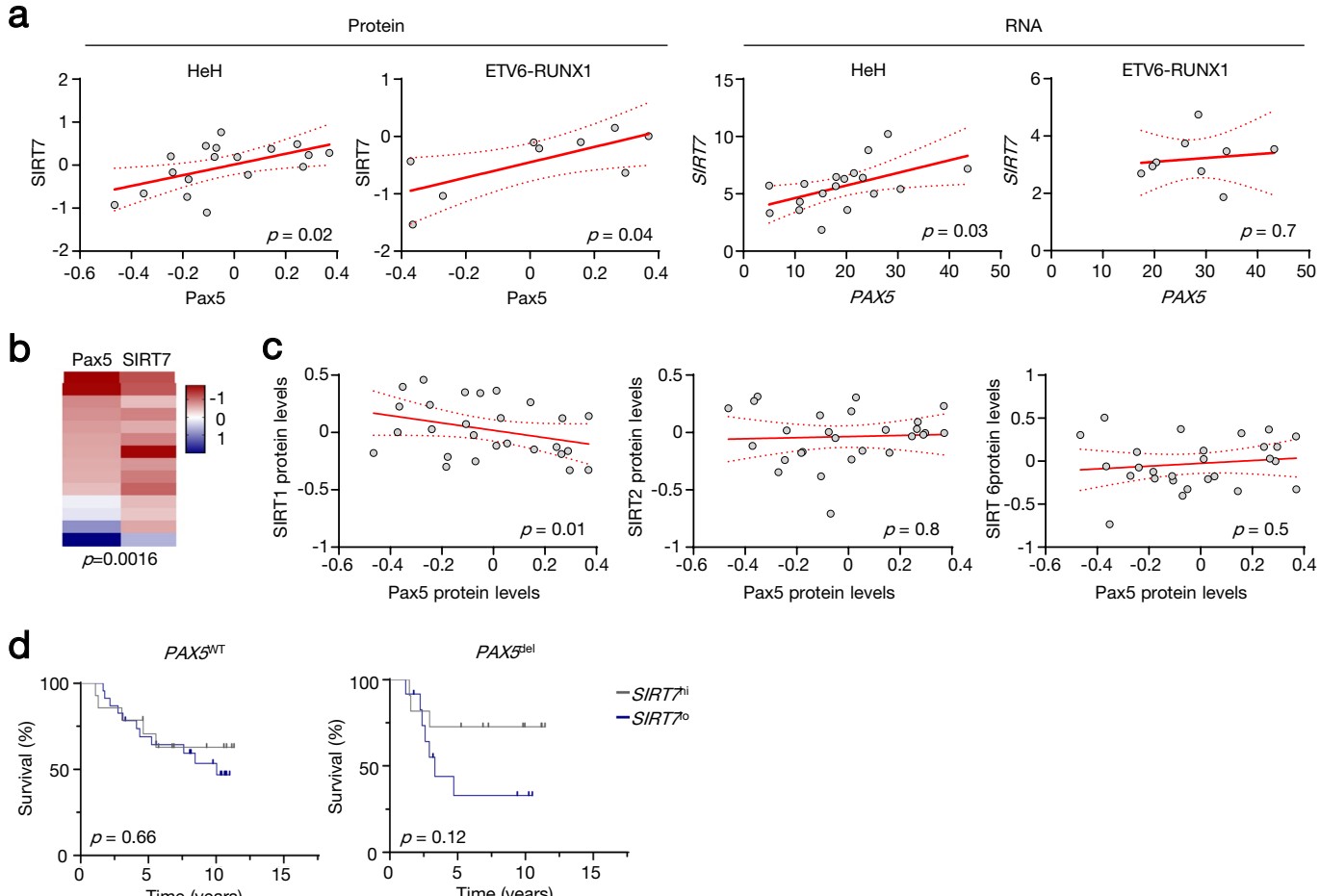

**Extended Data Fig. 8 | The PAX5/SIRT7 interplay is conserved in human B-ALL. a**, Scatter plots of Pax5 and SIRT7 levels derived from proteomics (left panels) and RNA-Seq (right panels) of human B-ALL HeH (n = 18) and ETV6-RUNX1 (n = 9) patient samples[29]. Each point corresponds to one sample. Linear regression and 95% confidence intervals (dashed lines) are shown in red. The p-values were calculated by one-tailed t-test. **b**, Heatmap showing the correlation between the protein levels of Pax5 and SIRT7 in B-CLL cell lines versus healthy donor B cells, as determined by proteomics[34]. The p-values were determined by one-tailed t-test (p = 0.0016). **c**, Correlation between the protein levels of Pax5 and SIRT1, SIRT2 and SIRT6 in human B-ALL patient samples[29], as determined by proteomics (n = 27). Data are presented and analyzed as in **a**. **d**, Kaplan-Meier survival curves of children with high-risk B-ALL (COG-P9906 study)[35,36] stratified into four groups based on higher- or lower-than-median *SIRT7* RNA expression levels and the occurrence of *PAX5* deletions[37]. Statistical significance was determined by log-rank test (*PAX5*WT*SIRT7*hi, n = 14; *PAX5*WT*SIRT7*lo, n = 15; *PAX5*del*SIRT7*hi, n = 11; *PAX5*del*SIRT7*lo, n = 12).

**Extended Data Table 1 | List of primer sequences used for RT-qPCR and *IgH* recombination semi-quantitative PCR**

| | |
|---|---|
| **RT-qPCR primers** | |
| Trbc1-Fw | 5'-ACCTTCTGGCACAATCCTCG-3' |
| Trbc1-Rev | 5'-GGCCTCTGCACTGATGTTCT-3' |
| Trbc2-Fw | 5'-TGGCACAATCCTCGAAACCA-3' |
| Trbc2-Rev | 5'-TGATTCCACAGTCTGCTCGG-3' |
| Thy1-Fw | 5'-GCTCTCCTGCTCTCAGTCTTG-3' |
| Thy1-Rev | 5'-TGTTATTCTCATGGCGGCAGT-3' |
| Cd3e-Fw | 5'-TCACTCTGGGCTTGCTGATG-3' |
| Cd3e-Rev | 5'-TTGCGGATGGGCTCATAGTC-3' |
| Zap70-Fw | 5'-GCTTGAAGGAGGTCTGTCCC-3' |
| Zap70-Rev | 5'-TATCCGTCCGAGTTCAGGGT-3' |
| Tigit-Fw | 5'-CTGTGCTGGGACTCATTTGCT-3' |
| Tigit-Rev | 5'-AGACTCCTCAGGTTCCATTCCT-3' |
| Pax5-Fw | 5'-GGAGGATCCAAACCAAAGGT-3' |
| Pax5-Rev | 5'-TTGTCACAGACTCGCTCTGC-3' |
| **_IgH_ recombination primers[43]** | |
| $D_H$L-Fw | 5'-GGAATTCGMTTTTTGTSAAGGGATCTACTACTGTG-3' |
| $V_H$5558-Fw | 5'-CGAGCTCTCCARCACAGCCTWCATGCARCTCARC-3' |
| $V_H$-Q52-Fw | 5'-CGGTACCAGACTGARCATCASCAAGGACAAYTCC-3' |
| $V_H$-7183-Fw | 5'-CGGTACCAAGAASAMCCTGTWCCTGCAAATGASC-3' |
| $J_H$3-Rev | 5'-GTCTAGATTCTCACAAGAGTCCGATAGACCCTGG-3' |
| Kat8-Fw | 5'-TATCTGCCTTTCTCTGTCAATGGG-3' |
| Kat8-Rev | 5'-AGGTGAGCCAGGTTAGGACTTGG-3' |

Berta N Vazquez

# Reporting Summary

## Statistics

For all statistical analyses, confirm that the following items are present in the figure legend, table legend, main text, or Methods section.

| n/a | Confirmed | |
|---|---|---|
| ☐ | ☒ | The exact sample size (*n*) for each experimental group/condition, given as a discrete number and unit of measurement |
| ☐ | ☒ | A statement on whether measurements were taken from distinct samples or whether the same sample was measured repeatedly |
| ☐ | ☒ | The statistical test(s) used AND whether they are one- or two-sided<br>*Only common tests should be described solely by name; describe more complex techniques in the Methods section.* |
| ☒ | ☐ | A description of all covariates tested |
| ☐ | ☒ | A description of any assumptions or corrections, such as tests of normality and adjustment for multiple comparisons |
| ☐ | ☒ | A full description of the statistical parameters including central tendency (e.g. means) or other basic estimates (e.g. regression coefficient) AND variation (e.g. standard deviation) or associated estimates of uncertainty (e.g. confidence intervals) |
| ☐ | ☒ | For null hypothesis testing, the test statistic (e.g. *F*, *t*, *r*) with confidence intervals, effect sizes, degrees of freedom and *P* value noted<br>*Give P values as exact values whenever suitable.* |
| ☒ | ☐ | For Bayesian analysis, information on the choice of priors and Markov chain Monte Carlo settings |
| ☒ | ☐ | For hierarchical and complex designs, identification of the appropriate level for tests and full reporting of outcomes |
| ☐ | ☒ | Estimates of effect sizes (e.g. Cohen's *d*, Pearson's *r*), indicating how they were calculated |

*Our web collection on statistics for biologists contains articles on many of the points above.*

## Software and code

Policy information about availability of computer code

| | |
|---|---|
| Data collection | Flow cytometry samples were run in a FACS Canto II (BD Biosciences) or sorted with a FACS Aria II (BD Biosciences). Data were collected with BD FACS Diva software 6.1.3.<br>Western blot images were obtained with an iBright 1500 (Invitrogen).<br>ELISA results were read in a Multiskan Sky (ThermoFisher) plate reader.<br>Gel filtration HPLC were performed with a GE AKTA Purifier 10 FPLC System.<br>RT-qPCR experiments were run on a QuantStudio 5 (ThermoFisher).<br>Chromatin shearing for ChIP experiments was performed with a Covaris M220 system.<br>Mass spectrometry data were acquired on an Orbitrap Fusion Lumos Tribrid system (ThermoFisher).<br>ChIP-Seq and RNA-Seq data were acquired on a DNBSEQ-G400 instrument. |
| Data analysis | Data presentation and statistical analyses: GraphPad Prism 8.0.1., ggplot 3.5.1. and Microsoft Excel 2309.<br>RT-qPCR data were processed with QuantStudio Design & Analysis Software 1.5.1.<br>Flow cytometry data analysis: FlowJo 7.6.<br>Western blot densitometric estimation and histological sections processing: ImageJ 1.52a.<br>Mass spectrometry data were processed with Xcalibur software (Proteome, version 4.0.27.10; PAX5 acetylome, version 4.2.28.14) (ThermoFisher).<br>sc-RNA-Seq visualization and data collection: Single Cell Portal (Broad Institute).<br>Gene set enrichment analysis: GSEA 4.1.0.<br>Heatmap visualization and clustering: Morpheus software (Broad Institute).<br>Gene ontolgy terms: Enrichr tool.<br>Visualization of ChIP-Seq and RNA-Seq tracks: Integrated Genome Viewer 2.7.2. |

For manuscripts utilizing custom algorithms or software that are central to the research but not yet described in published literature, software must be made available to editors and reviewers. We strongly encourage code deposition in a community repository (e.g. GitHub). See the Nature Portfolio guidelines for submitting code & software for further information.

Motif Analysis: HOMER v.5.0.1.
Differential Expression Analysis: DESeq2
ChIP-Seq binding profiles: DiffBind and plotHeatmap (deepTools v3.5.1)

For manuscripts utilizing custom algorithms or software that are central to the research but not yet described in published literature, software must be made available to editors and reviewers. We strongly encourage code deposition in a community repository (e.g. GitHub). See the Nature Portfolio guidelines for submitting code & software for further information.

## Data

Policy information about availability of data

All manuscripts must include a data availability statement. This statement should provide the following information, where applicable:
- Accession codes, unique identifiers, or web links for publicly available datasets
- A description of any restrictions on data availability
- For clinical datasets or third party data, please ensure that the statement adheres to our policy

All Pax5WT, Pax5K198Q and Pax5K198R ChIP-Seq data and RNA-Seq data from wild-type and Sirt7-/- bone marrow pro-B and pre-B cells; Pax5-/- B cell progenitors retrovirally expressing EV, Pax5WT, Pax5K198Q and Pax5K198R; and wild-type or Sirt7-/- B cell progenitors expressing EV, Pax5WT or Pax5K198R have been deposited at the National Center for Biotechnology Information Gene Expression Omnibus (GEO) and are available under the accession number: GSE246370. All sequencing data were aligned onto the mm10 reference mouse genome. Proteomics data from wild-type and Sirt7-/- bone marrow pre-B cells and Pax5 acetylation experiments have been deposited to the ProteomeXchange Consortium via the PRIDE partner repository with the dataset identifier PXD046457.

## Research involving human participants, their data, or biological material

Policy information about studies with human participants or human data. See also policy information about sex, gender (identity/presentation), and sexual orientation and race, ethnicity and racism.

| Reporting on sex and gender | N/A |
| Reporting on race, ethnicity, or other socially relevant groupings | N/A |
| Population characteristics | N/A |
| Recruitment | N/A |
| Ethics oversight | N/A |

Note that full information on the approval of the study protocol must also be provided in the manuscript.

## Field-specific reporting

Please select the one below that is the best fit for your research. If you are not sure, read the appropriate sections before making your selection.

☒ Life sciences          ☐ Behavioural & social sciences          ☐ Ecological, evolutionary & environmental sciences

For a reference copy of the document with all sections, see nature.com/documents/nr-reporting-summary-flat.pdf

## Life sciences study design

All studies must disclose on these points even when the disclosure is negative.

| Sample size | No sample size calculation was performed, as experiments were performed on genetically identical mice or cell lines. Sample size was based on previous experiments, as well as on the 3R principales to reduce the number of animal used. Sample sizes were sufficient to detect differences between experimental groups. |
| Data exclusions | No data was excluded from the manuscript. |
| Replication | All replicates were performed under the same conditions and are described in the manuscript. All attempts of replication were successful. Results were similarly replicated in at least two independent experiments, and results pooled from independent experiments are indicated in the corresponding figure legends. |
| Randomization | For in vivo transplantation experiments, recipient mice were randomized to exclude age- and sex-related effects. The rest of the experiments were performed with genetically identical mice, so randomization was not required. |
| Blinding | Investigators were not blinded, as most experiments were performed by a single person. In the case of sequencing and proteomics experiments, much of the bioinformatic analysis was performed by an independent person that was not involved in the performance of the experiment. |

# Reporting for specific materials, systems and methods

We require information from authors about some types of materials, experimental systems and methods used in many studies. Here, indicate whether each material, system or method listed is relevant to your study. If you are not sure if a list item applies to your research, read the appropriate section before selecting a response.

## Materials & experimental systems

| n/a | Involved in the study |
|-----|-----------------------|
| ☐ | ☒ Antibodies |
| ☐ | ☒ Eukaryotic cell lines |
| ☒ | ☐ Palaeontology and archaeology |
| ☐ | ☒ Animals and other organisms |
| ☒ | ☐ Clinical data |
| ☒ | ☐ Dual use research of concern |
| ☒ | ☐ Plants |

## Methods

| n/a | Involved in the study |
|-----|-----------------------|
| ☐ | ☒ ChIP-seq |
| ☐ | ☒ Flow cytometry |
| ☒ | ☐ MRI-based neuroimaging |

## Antibodies

| | |
|---|---|
| Antibodies used | For flow cytometry, all antibodies were used at a dilution of 1:400, unless otherwise specified:<br>anti-B220 (RA3-6B2, eBioscience, 17-0452-81)<br>anti-CD19 (HIB19, eBioscience, 562440)<br>anti-CD43 (eBioR2/60, eBioscience, 1:800 dilution, 11-0431-82)<br>anti-CD43 (1510, BD Biosciences, 1:800 dilution, 563377)<br>anti-IgM (II/41, eBioscience, 1:800 dilution, 553437)<br>anti-IgD (11-26c, eBioscience, 47-5993-80)<br>anti-CD21 (7G6, BDBiosciences, 1:800 dilution, 552957)<br>anti-CD23 (B3B4, BDBiosciences, 1:200 dilution, 553137)<br>anti-CD93 (AA4.1, eBioscience, 11-5892-82)<br>anti-GL7 (GL-7, eBioscience, 1:200 dilution, 48-5902-82)<br>anti-CD38 (90, eBioscience, 406-0381-80)<br>anti-CD138 (300506, Invitrogen, MA5-23553)<br>anti-Fas (SA367H8, Biolegend, 1:600 dilution, 152617)<br>anti-IgG1 (A85-1, BD Pharmigen, 1:600 dilution, 553441)<br>anti-CD127 (eBioSD/199, eBioscience, 1:200 dilution, 12-1273-81)<br>anti-CD45.1 (A20, eBioscience, 11-0453-81)<br>anti-CD45.2 (104, eBioscience, 1:200 dilution, 17-0454-81)<br>anti-TCRβ (H57-597, BD Biosciences, 1:200 dilution, 14-5961-82)<br>anti-NKp46 (29A1.4, Biolegend, 1:100 dilution, 137618)<br>anti-CD4 (GSK1.5, eBioscience, 1:800 dilution, 46-0041-80)<br>anti-CD8 (53-6.7, eBioscience, 1:800 dilution, 12-0081-81)<br>anti-hCD4 (RPA-T4, Biolegend, 300504)<br>anti-CD3e (145-2C11, BD Biosciences, 559971)<br>anti-Ly76 (TER-119, BD Biosciences, 51-09082J)<br>anti-CD11b (M1/70, BD Biosciences, 51-01712J)<br>anti-Gr1 (RB6-8C5, BD Biosciences, 51-01212J)<br>anti-PAX5 (1H9, eBioscience, 0.2 µg per sample, 14-9918-82)<br>anti-SIRT7 (D3K5A, Cell signaling, 0.25 µg per sample, 5360)<br>anti-STAT5-pY694 (47/Stat5(pY694), 20 µL per sample, BD Biosciences, 612567)<br>anti-Rabbit IgG (H+L) polyclonal secondary antibody (Invitrogen, 0.25 µg per sample, 11034)<br><br>For magnetic purification of CD19+ cells:<br>anti-CD19 (1D3, BD Biosciences, 0.1 µg per 10^6 cells, 553784)<br><br>For ChIP-Seq:<br>anti-PAX5 (Abcam, 10 µg per sample, ab183575)<br><br>For Western Blot, all antibodies were used at a dilution of 1:1000 in PBS-0.1% Tween, unless otherwise specified:<br>anti-SIRT7 (D3K5A, Cell signaling, 5360S)<br>anti-PAX5 (D19F8, Cell signaling, 8970S)<br>anti-H3 (ab1791, Abcam)<br>anti-Fibrillarin (B1, Santa Cruz Biotechnology, sc-166001)<br>anti-Acetyl-lysine (Ac-K2-100, Cell signaling, 1:200 dilution, 9814)<br>anti-FLAG (M2, Sigma-Aldrich, 1:10,000 dilution, A8592)<br>anti-HA (Sigma, 1:5,000 dilution, H6908)<br>anti-Myc-Tag (9B11, Cell signaling, 2276)<br>anti-V5 (ab9116, Abcam)<br>anti-Actin (AC-15, Sigma-Aldrich, 1:5000 dilution, A1978) |

For ELISA:
rat anti-mouse IgM (Ig Isotyping Mouse Uncoated ELISA Kit, Invitrogen, 1/250 dilution, 88-50630)
rat anti-mouse IgG1 (Ig Isotyping Mouse Uncoated ELISA Kit, Invitrogen, 1/250 dilution, 88-50630)
rat anti-mouse IgG3 (Ig Isotyping Mouse Uncoated ELISA Kit, Invitrogen, 1/250 dilution, 88-50630)
goat anti-rat IgG conjugated to HRP (Sigma-Aldrich, 1:5000 dilution, AP136P)

For immunoprecipitation:
anti-FLAG (M2, Sigma, 20ul per sample, A2220)

| | |
|---|---|
| Validation | All the antibodies used are commercially available and were validated by the manufacturer for the intended applications and species (validation information available at vendor's website), with one exception: the anti-SIRT7 (D3K5A, Cell signaling) antibody has not been reported for flow cytometry, but we have validated it in our lab using proper isotype controls as well as wild-type, SIRT7-deficient, and SIRT7-overexpressing mouse cells to determine its specificity. |

## Eukaryotic cell lines

Policy information about cell lines and Sex and Gender in Research

| | |
|---|---|
| Cell line source(s) | KOPN-8, NALM-20, REH, TANOUE, SD-1, TOM-1 and SEM cells were purchased from the DSMZ-German Collection of Microorganisms and Cell Cultures GmbH and were kindly provided by M. Parra (IJC, Barcelona, Spain). HAFTL cells are a fetal liver-derived, Ha-Ras-transformed mouse pre-B cell line that have been previously described (Alessandrini A. et al. Continuing rearrangement of immunoglobulin and T-cell receptor genes in a Ha-ras-transformed lymphoid progenitor cell line.Proc Natl Acad Sci U S A 84(7): 1799–1803 (1987) and were provided by Dr. Maribel Parra (IJC, Barcelona, Spain). OP9, HEK293F cells were purchased from the American Type Culture Collection (ATCC). Platinum E cells were purchased from Cell Biolabs. Primary pro-B cells were obtained as detailed in the Methods section from both male and female mice. |
| Authentication | Cell lines were not authenticated. |
| Mycoplasma contamination | All tested cell lines were negative for mycoplasma contamination. |
| Commonly misidentified lines (See ICLAC register) | None of the cell lines used were found in the Commonly misidentified lines database |

## Animals and other research organisms

Policy information about studies involving animals; ARRIVE guidelines recommended for reporting animal research, and Sex and Gender in Research

| | |
|---|---|
| Laboratory animals | Wild-type, Sirt7-/- and IgHEL+ mice were in the 129Sv background, unless otherwise specified in the manuscript. In these cases, C57BL/6 wild-type and Sirt7-/- were used to confirm our observations on a different background. Pax5-/- and CD45.2 mice were in the C57BL/6 background. Heterozygous CD45.1+CD45.2+ mice were generated by crossing C57BL/6 CD45.1 and Wt 129Sv CD45.2 mice for one generation. Both sexes were included in all experiments, and only mice of 8-16 weeks were used, except for isolation of Pax5-/- progenitors, that were obtained from murine fetal livers. |
| Wild animals | Study did not involve wild animals. |
| Reporting on sex | Both sexes were included in all experiments, and no sex comparisons have been performed. |
| Field-collected samples | Study did not involve field-collected samples. |
| Ethics oversight | Animal studies were conducted at IJC (Spain) according to national authorities and institutional ethics committees (Germans Trias i Pujol Reserach Institute Ethics Committee). The collection of bone marrow samples from C57BL/6 wild-type and Sirt7-/- mice, and the generation of Pax5-/- mouse B cell progenitors were conducted according to national authorities and institutional ethics committees at MPI-HLR (Germany) and Lund University (Sweden), respectively. |

Note that full information on the approval of the study protocol must also be provided in the manuscript.

## Plants

| | |
|---|---|
| Seed stocks | *Report on the source of all seed stocks or other plant material used. If applicable, state the seed stock centre and catalogue number. If plant specimens were collected from the field, describe the collection location, date and sampling procedures.* |
| Novel plant genotypes | *Describe the methods by which all novel plant genotypes were produced. This includes those generated by transgenic approaches, gene editing, chemical/radiation-based mutagenesis and hybridization. For transgenic lines, describe the transformation method, the number of independent lines analyzed and the generation upon which experiments were performed. For gene-edited lines, describe the editor used, the endogenous sequence targeted for editing, the targeting guide RNA sequence (if applicable) and how the editor was applied.* |
| Authentication | *Describe any authentication procedures for each seed stock used or novel genotype generated. Describe any experiments used to assess the effect of a mutation and, where applicable, how potential secondary effects (e.g. second site T-DNA insertions, mosiacism, off-target gene editing) were examined.* |

# ChIP-seq

## Data deposition

☒ Confirm that both raw and final processed data have been deposited in a public database such as GEO.

☒ Confirm that you have deposited or provided access to graph files (e.g. BED files) for the called peaks.

**Data access links**
*May remain private before publication.*

ChIP-Seq data are available at GEO under the accession number: GSE246370 (https://www.ncbi.nlm.nih.gov/geo/query/acc.cgi?acc=GSE246370)

**Files in database submission**

GSM7867743 Wt_proB_1
GSM7867744 Wt_preB_1
GSM7867745 Sirt7-/-_proB_1
GSM7867746 Sirt7-/-_preB_1
GSM7867747 Wt_proB_2
GSM7867748 Wt_preB_2
GSM7867749 Sirt7-/-_proB_2
GSM7867750 Sirt7-/-_preB_2
GSM7867751 Pax5_EV_1_RNA
GSM7867752 Pax5_EV_2_RNA
GSM7867753 Pax5_Wt_1_RNA
GSM7867754 Pax5_Wt_2_RNA
GSM7867755 Pax5_K198Q_1_RNA
GSM7867756 Pax5_K198Q_2_RNA
GSM7867757 Pax5_K198R_1_RNA
GSM7867758 Pax5_K198R_2_RNA
GSM8376251 Input_Pax5_Wt_1
GSM8376252 Input_Pax5_Wt_2
GSM8376253 Input_Pax5_K198Q_1
GSM8376254 Input_Pax5_K198Q_2
GSM8376255 Input_Pax5_K198R_1
GSM8376256 Input_Pax5_K198R_2
GSM8376257 ChIP_Pax5_Wt_1
GSM8376258 ChIP_Pax5_Wt_2
GSM8376259 ChIP_Pax5_K198Q_1
GSM8376260 ChIP_Pax5_K198Q_2
GSM8376261 ChIP_Pax5_K198R_1
GSM8376262 ChIP_Pax5_K198R_2
GSM8376263 WT_EV_1
GSM8376264 WT_EV_2
GSM8376265 Sirt7-/-_EV_1
GSM8376266 Sirt7-/-_EV_2
GSM8376267 Sirt7-/-_Pax5Wt_1
GSM8376268 Sirt7-/-_Pax5Wt_2
GSM8376269 Sirt7-/-_Pax5K198R_1
GSM8376270 Sirt7-/-_Pax5K198R_2

**Genome browser session**
(e.g. UCSC)

*Provide a link to an anonymized genome browser session for "Initial submission" and "Revised version" documents only, to enable peer review. Write "no longer applicable" for "Final submission" documents.*

## Methodology

**Replicates**

Two biological replicates for each sample.

**Sequencing depth**

Clean reads per sample (SE50 sequencing):
Input_Pax5_Wt_1: 25,309,841
Input_Pax5_Wt_2: 25,294,390
Input_Pax5_K198Q_1: 25,291,043
Input_Pax5_K198Q_2: 25,280,864
Input_Pax5_K198R_1: 25,271,600
Input_Pax5_K198R_2: 25,308,898
ChIP_Pax5_Wt_1: 25,214,352
ChIP_Pax5_Wt_2: 25,271,937
ChIP_Pax5_K198Q_1: 25,174,210
ChIP_Pax5_K198Q_2: 25,201,975
ChIP_Pax5_K198R_1: 25,205,736
ChIP_Pax5_K198R_2: 25,209,517

**Antibodies**

anti-PAX5 (Abcam, 10 µg per sample, ab183575, Lot #1037689-1)

**Peak calling parameters**

Peak calling was performed using MACS2 "callpeak" (-f BAM --nomodel --extsize 20 -g mm -B) taking into account both IP (-t) and input (-c) samples for each peak calling.

| Data quality | All the raw .fastq files were quality checked using FastQC. Universal adapters were trimmed using Trim Galore. After Bowtie2 alignment, we selected from the sorted BAM files only those reads that were succesfully aligned to the genome, eliminating all th.e PCR duplicates and filtering reads by MAPQ > 36 using SAMtools and SAMbamba. BigWig files were generated using DeepTools |
| Software | Windows 10 with "Ubuntu on Windows" (v22.04.2 LTS), with LAN access to our High-Performance Computer (HPC) server (Josep Carreras Leukaemia Research Institute supercomputer). Ubuntu terminal using Bash as standard language with all the associated packages retrieved from our internal HPC server (Bowtie2, STAR, MACS2, Salmon, SAMbamba, SAMtools, FastQC, Trim Galore, plotHeatmap, DeepTools). R-Studio for using R-Language and associated packages (Tximeta, DiffBind, ggplot2, DESeq2, ChIPpeakAnno). Anaconda3 "Conda" environment to execute Python associated packages (Scanpy). |

# Flow Cytometry

## Plots

Confirm that:

☒ The axis labels state the marker and fluorochrome used (e.g. CD4-FITC).

☒ The axis scales are clearly visible. Include numbers along axes only for bottom left plot of group (a 'group' is an analysis of identical markers).

☒ All plots are contour plots with outliers or pseudocolor plots.

☒ A numerical value for number of cells or percentage (with statistics) is provided.

## Methodology

| Sample preparation | Bone marrow, spleen and thymus samples were collected from wild-type, Sirt7-/-, IgHEL+, Sirt7-/-IgHEL+, CD45.1+CD45.2+ and CD45.2+ mice. Bone marrow samples were crushed in staining buffer (3% fetal bovine serum (FBS), 2 mM EDTA in phosphate buffer saline (PBS)) to obtain single cell suspensions. Spleen and thymus samples were similarly processed. Red blood cells were lysed in ACK buffer (Gibco), and the reaction was stopped by adding five volumes of staining buffer. Cells were filtered through 40-µm sterile strainers and incubated with Fc-block (eBioscience) before staining for 30 min on ice with the corresponding antibodies. Stained cells were then washed and analyzed or sorted. |
| Instrument | FACS Canto II and LSRFortessa SORP (BD Biosciences). |
| Software | Data were collected with BD FACS Diva software 6.1.3. and analyzed with FlowJo 7.6. sofware. |
| Cell population abundance | Sorted populations were re-run on a flow cytometer to ensure their purity. |
| Gating strategy | In all samples, singlets were gated first by FSC-H and FSC-A. Progenitors/Lymphocytes were subsequently gated by FSC VS SSC. |

☒ Tick this box to confirm that a figure exemplifying the gating strategy is provided in the Supplementary Information.

