## [Peer Review File · Nature Immunology]

A SIRT7-dependent acetylation switch regulates early B cell differentiation and lineage commitment through Pax5

Corresponding Author: Dr Alejandro Vaquero

Version 0:

Decision Letter:

15th Dec 2023

Dear Dr. Vaquero,

Your Article, "A SIRT7-dependent acetylation switch regulates early B-cell differentiation and lineage commitment through PAX5" has now been seen by 3 referees. While we find your work of considerable potential interest, the reviewers have raised substantial concerns that must be addressed. As such, we cannot accept the current version of the manuscript for publication, but would be happy to consider a revised version that addresses these concerns, as long as novelty is not compromised in the interim.

Please revise the manuscript to address all issues raised by the referees. At resubmission, please include a point-by-point "Response to referees" detailing how you have addressed each referee comment (please specify page and figure number where the new data can be found in the revised manuscript). This response will be sent back to the referees along with the revised manuscript.

In addition, please include a revised version of any required reporting checklist. It will be available to referees (and, potentially, statisticians) to aid in their evaluation if the manuscript goes back for peer review. A revised checklist is essential for re-review of the paper. The Reporting Summary can be found here: <https://www.nature.com/documents/nr-reporting-summary.pdf>

Link Redacted

We hope to receive a suitably revised manuscript within 6 months. If you cannot send it within this time, please let us know. We will be happy to consider your revision so long as nothing similar has been accepted for publication at Nature Immunology or published elsewhere.

Nature Immunology is committed to improving transparency in authorship. As part of our efforts in this direction, we are now requesting that all authors identified as 'corresponding author' on published papers create and link their Open Researcher and Contributor Identifier (ORCID) with their account on the Manuscript Tracking System (MTS), prior to acceptance. ORCID helps the scientific community achieve unambiguous attribution of all scholarly contributions. You can create and link your

ORCID from the home page of the MTS by clicking on 'Modify my Springer Nature account'. For more information please visit please visit www.springernature.com/orcid.

Thank you for the opportunity to review your work.

Sincerely,

Ioana Visan, Ph.D.
Senior Editor
Nature Immunology

Tel: 212-726-9207
Fax: 212-696-9752
www.nature.com/ni

Reviewers' Comments:

Reviewer #1:

Remarks to the Author:

The Authors showed that SIRT7 is important for B cell lymphopoiesis by deacetylating and stabilizing the B cell lineage transcription factor PAX5. SIRT7 deficient cells have about 50% decrease in B cells. They identified the potential lysine residue at position 198 (K198) and the expression of acetylation mimic can similarly destabilize PAX5. In human B-ALL, similar to PAX5, SIRT7 expression associates with good prognosis. While the results are potentially interesting, detailed mechanisms remained to be identified. For instance, it is unknown: 1) how K198 acetylation promotes PAX5 degradation and how the modification links to ubiquitin ligase and deubiquitinase; 2) the responsible acetyltransferase(s); 3) if SIRT7 has functional significance in B cell immunity.

1. In Fig 1g-j, the splenic B cells decreased by half in the SIRT7-KO. However, the percentages of IgM+ and IgD+ cells were significantly lower. What are the phenotypes of the B cells in the spleen and lymph nodes? For instance, transitional, marginal zone, germinal center, memory, and plasma cells. What are the IgM- IgD- B cells? Are they class-switched or not expressing cell-surface Ig?

2. Does SIRT7 deficiency affect serum antibody isotypes (basal and immunization)? As SIRT7-KO decreased PAX5 protein level, do the mice have accelerated germinal center and plasma cell differentiation?

3. Does SIRT7 deficiency peripheral B cells experience lineage instability?

4. Adoptive ProB cells transfer only resulted in very minimal reconstitution and lead to highly variable results (Fig. 1l-n; Fig. 5i-j). Bone marrow chimera and conditional SIRT7 KO should be used to confirm the effect of PAX5 mutants on the development of B cells and lineage skewing with larger sample sizes.

5. Previous studies have shown that P300-mediated acetylation on other lysine residues promotes PAX5 activity (He et al., 2011). Given the negative role of K198 acetylation, is K198 acetylation differentially regulated and uncoupled with the acetylation on other lysine residues?

6. In Fig. 2b-e, while it was suggested that PAX5 is anti-proliferative, the Authors should shed light on why SIRT7 KO B cells have fewer proliferating cells when they have lower level of PAX5?

7. Fig. 3p, Does the re-expression of SIRT7 and PAX5 restore B cell development and cell number?

8. Does overexpress or knockout SIRT7 affect the protein level and transactivation activity of PAX5K198 mutants?

9. In Fig. 4P, PAX5 acetylated mimics has lower protein levels in HEK293F cells. However, the Authors should include control proteins to indicate the transfection efficiency. It will be best if the reporter proteins are linked to PAX5 via 2A peptide to ensure the original ratio between reporter and PAX5 is 1:1.

10. In line 220 and Fig. 4m-n, the experiment was done in HEK293F cells, suggesting PAX5 is acetylated by ubiquitous acetyltransferases. With this amenable system, the Authors should be able to identify the responsible acetyltransferases.

11. In Fig. 3m, what are the other changes in the proteome? Are there any common features shared between these proteins and PAX5 (e.g., the motif around K198)?

12. In Fig. 3f, how many of the genes in Cluster 3 actually have PAX5 peaks around the promoters (correlating data from Fig. 5).

13. Additional proof for the interaction between SIRT7 and PAX5 in the cells (proximity ligation).
14. Which lysine(s) on PAX5 got Ub when K198 is Ac?
15. Does the acetylation status at K198 affect sub-cellular localization of PAX5 (e.g., nuclear vs cytoplasmic)?
16. In Fig. 4a, can SIRT7 overexpression rescue PAX5 level in SIRT7 KO primary B cells?
17. In Fig. 4f-g, how does SIRT7 deficiency affects PAX5 stability in mouse B cells?
18. Western blot should be quantified. For example, Fig. 4q, the amount of HA-Ub should be normalized to the PAX5 level (especially K198R seems to have less FLAG).
19. In line 243, how much of the decreased chromatin binding is due to decreased PAX5 proteins or its intrinsic DNA binding? In another words, does PAX5 K198 Ac differentially regulate PAX5 activity independent of protein stability?
20. In Fig. 5b, Does K198Q and K198R affect the DNA motifs that PAX5 binds to? What kind of regions are shown? Are there any differential peaks?
21. The role of SIRT7 and PAX5 acetylation should be tested using normal mouse and human B cells. Does SIRT7 KD decrease PAX5 level in other B cell lines and primary B cells? Similarly, does expression of SIRT7 increase PAX5?

Reviewer #2:

Remarks to the Author:

This is an outstanding study that identifies a novel mechanism of control over B cell development and helps to understand the dual role of Pax5 in controlling B cell lineage commitment by repressing and activating gene expression. I was impressed by the breadth and depth of the analyses.

I only have one key question that I think needs answering:

The overarching hypothesis of the authors is that the B cell development defect observed in SIRT7 KO cells is directly attributed to its role in Pax5 acetylation. However, Sirtuins are implicated in several different roles, the alteration of which could give rise to the SIRT7 KO phenotype. To support their hypothesis the authors should perform a key experiment which is to rescue the SIRT7 KO phenotype with the deacetylated Pax5 mutant (K198R). This would be a similar experiment to that performed in Figure 5 which introduced Pax5 wt and mutants into Pax5^{-/-} cells. RNAseq analysis as performed in Figure 5 should also be used to understand how lineage commitment was rescued by the K198R mutant at the transcriptome level.

Minor comments:

In line 94 the authors claim that SIRT7 is "specifically" upregulated. However, SIRT2 and to some extent SIRT1 are also upregulated during B cell development. I think the authors should tone this down and just say SIRT7 is upregulated.

Line 188 change "demonstrate" to "suggest" that SIRT7 and Pax5 collaborate.

Reviewer #3:

Remarks to the Author:

Vaquero and colleagues investigated the Sirt subfamily of histone deacetylase in B cell development and identified Sirt7 as a critical regulator that promotes pro to preB maturation, in a cell intrinsic, and deacetylase enzyme activity-dependent manner. The authors further showed that Sirt7 is critical for suppressing T cell potential in proB cells and promoting cell proliferation, with the former mediated by deacetylation and stabilization of Pax5. Through mass spec studies, the authors identified K198 in Pax5 as a major target for deacetylation by Sirt7. Pax5-K198Q mutant (simulating acetylated form) destabilized Pax5 and failed to suppress T lineage potential while Pax5-K198R mutant (resistant to acetylation) showed an opposite effect. The reverse correlation between Sirt7 and Pax5 expression was also observed in B-ALL cells. These studies are mostly well designed, and the findings are of substantial interest, and the Sirt7-Pax5 regulatory axis via post-translational modification is a novel advance. However, the scope of the study in the current form is somehow limited. The authors may gain broadened insights into the regulatory activities of Sirt7 by addressing the following questions:

1. Pax5 protein pool. In Sirt7KO cells, Pax5 protein was reduced but substantial amount was retained (Fig. 4b, d, and 4f after CHX treatment). The authors also made an observation that Pax5-K198R mutant failed to rescue B cell development in Pax5KO mice, in spite of its ability to activate and suppress Pax5 target genes. Taken these together, one interpretation is that in a B cell, there should be a pool of Pax5 protein that coexist in the K198 acetylated form and unacetylated form, and both forms may have balanced acts, leading to B cell maturation. This hypothesis is worth testing, but forced expression of both the K198Q and K198R mutants at the 1:1 ratio (or various ratios).

The use of Arginine (R) to replace in place of K is not an ideal choice, because R is subject to many modifications itself as well (<https://pubmed.ncbi.nlm.nih.gov/18603028/>). Pax5-K198R mutant might gain some new functions besides becoming resistant to deacetylation by Sirt7. This is based on the data in Fig. 5d, where Pax5-K198R appeared to a super-Pax5, that over-repressed cluster 1 genes, and excessively induced cluster 2 genes. Testing a more neutral mutation such as Pax5-K198A, alone or in combination with Pax5-K198Q could provide more insights.

Because Pax5-K198R is more stable, the absolute protein amount/cell was likely higher than WT. The ChIP-seq signal in Fig. 5b should be normalized by the protein amount. In addition to quantitative measurement of peak strength, there are potentially qualitative changes, after proper normalization, such as Pax5-K198Q failed to bind specific sites occupied by WT Pax5, and Pax5-K198R may gain new targets, which may account for its unusual gene activation/repression patterns.

Fig. 1l, m, n, the authors used mutant Sirt7 to rescue Sirt7 deficiency. Testing Pax5 mutants would be fruitful efforts as well to see how Pax5-K198R (or Pax5-K198A) in a different cell context. This experiment needs an additional control, WT EV.

2. Sirt7 activity. Sirt7 deficiency had profound impact on B cell development and target gene expression. These had shared features with Pax5 deficiency, but they are clearly not phenocopies of each other. The authors need to clearly document the distinct features between the two, besides their similarity. The DEGs due to Sirt7 deficiency should be compared with Pax5-activated or -repressed genes to examine what portion of Sirt7 target regulation is ascribed to Pax5.

Sirt7 is a histone deacetylase. Yet its impact on the histone modification in proB cells, on overall level or specific gene loci was not addressed at all. The singular focus on deacetylation of a non-histone protein Pax5 is a sharp focus but appeared to be narrow. The direct impact of Sirt7 on B cell epigenetic regulation, if biologically meaningful, should not be ignored in a comprehensive study.

3. Data presentation. The authors performed impressively extensive biochemical and genetic studies, yet the data description is generally very sketchy, without sufficient explanation. For a paper to appeal to non-experts and trainees in the field, some better description is needed. Some examples for improvement include:

- 1) Sirt7 KO, Cre or germline? No description in any section besides references. Two strains were mentioned, which one was used in this study, or both?
- 2) Figure 3e, mark numbers of differentially expressed genes, criteria for DEGs.
- 3) What genes are used for Figure 3f, clustering analysis? Direct comparison of wt and KO in pro and pre-B cells was not documented well. Cluster 2, specifically induced by Sirt7 in pre-B, it does not seem to be the case on heatmap.
- 4) Figure 3o, motif analysis on what? DEG promoters?
- 5) Fig. 4e, only 2 data points, no Standard error or standard deviation can be deduced.
- 6) Fig. 4h, what is the immunoblot antibody?
- 7) Fig. 4i, native PAGE was used? This is essential info to evaluate co-elution
- 8) Fig. 4l, IP Pax5, followed by IB anti-acetylated lysine?
- 9) Fig. 4m, more explanation are needed. In the peptide fragment, Kac is the location of K198? How can one tell the removal of 42 dalton acetyl group from the graph?
- 10) 5d, what DEGs are used for clustering?
- 11) How Sirt7^{-/-} HEK293F was made?
- 12) Peak calling conditions. IgG control or Pax5 KO control.
- 13) Figure 2f, impaired V to DJ recombination, could be secondary to reduced survival. Decouple the events (by Bcl2 transgene or other means) or acknowledge this possibility.

Version 1:

Decision Letter:

Dear Dr. Vaquero,

Thank you for your response to the reviewers' comments on your manuscript "A SIRT7-dependent acetylation switch regulates early B-cell differentiation and lineage commitment through PAX5". We are happy to inform you that if you revise your manuscript appropriately in response to the referees' comments and our editorial requirements your manuscript should be publishable in Nature Immunology.

Please revise your manuscript to include a detailed discussion of the possibility that a pool of acetylated and deacetylated Pax5 might have specific physiological relevance during B cell development. At resubmission, please include a point-by-point response to the referees' comments, noting the pages and lines where the changes can be found in the revision. Please highlight the changes in the revised manuscript as well.

We are trying to improve the quality and transparency of methods and statistics reporting in our papers (please see our editorial in the May 2013 issue). Please update the Life Sciences Reporting Summary, and supplements if applicable, with any information relevant to any new experiments and upload it (as a Related Manuscript File) along with the files for your revision. If nothing in the checklist has changed, please upload the current version again.

TRANSPARENT PEER REVIEW

Nature Immunology offers a transparent peer review option for new original research manuscripts submitted from 1st December 2019. We encourage increased transparency in peer review by publishing the reviewer comments, author rebuttal letters and editorial decision letters if the authors agree. Such peer review material is made available as a supplementary peer review file. **Please state in the cover letter 'I wish to participate in transparent peer review' if you want to opt in, or 'I do not wish to participate in transparent peer review' if you don't.** Failure to state your preference will result in delays in accepting your manuscript for publication.

ORCID

Nature Immunology is committed to improving transparency in authorship. As part of our efforts in this direction, we are now requesting that all authors identified as 'corresponding author' on published papers create and link their Open Researcher and Contributor Identifier (ORCID) with their account on the Manuscript Tracking System (MTS), prior to acceptance. ORCID helps the scientific community achieve unambiguous attribution of all scholarly contributions. For more information please visit www.springernature.com/orcid.

Before resubmitting the final version of the manuscript, if you are listed as a corresponding author on the manuscript, please follow the steps below to link your account on our MTS with your ORCID. If you don't have an ORCID yet, you will be able to create one in minutes. If you are not listed as a corresponding author, please ensure that the corresponding author(s) comply.

1. From the home page of the [MTS](https://mts-ni.nature.com/cgi-bin/main.plex) click on **'Modify my Springer Nature account'** under **'General tasks'**.
2. In the **'Personal profile'** tab, click on **'ORCID Create/link an Open Researcher Contributor ID(ORCID)'**. This will redirect you to the ORCID website.
- 3a. If you already have an ORCID account, enter your ORCID email and password and click on **'Authorize'** to link your ORCID with your account on the MTS.
- 3b. If you don't yet have an ORCID, you can easily create one by providing the required information and then click on **'Authorize'**. This will link your newly created ORCID with your account on the MTS.

IMPORTANT: All authors identified as 'corresponding authors' on the manuscript must follow these instructions. Non-corresponding authors do not have to link their ORCIDs, but please note that it will not be possible to add/modify ORCIDs at proof. Thus, if they wish to have their ORCID added to the paper, they must also follow the above procedure prior to acceptance.

To support ORCID's aims, we only allow a single ORCID identifier to be attached to one account. If you have any issues attaching an ORCID identifier to your Manuscript Tracking System account, please contact the [Platform Support Helpdesk](http://platformsupport.nature.com/).

We hope that you will support this initiative and supply the required information. Should you have any query or comments, please do not hesitate to contact me.

Nature Immunology has now transitioned to a unified Rights Collection system which will allow our Author Services team to quickly and easily collect the rights and permissions required to publish your work. Once your paper is accepted, you will receive an email in approximately 10 business days providing you with a link to complete the grant of rights. If you choose to publish Open Access, our Author Services team will also be in touch at that time regarding any additional information that may be required to arrange payment for your article.

For information regarding our different publishing models please see our [Transformational Journals](https://www.springernature.com/gp/open-research/transformational-journals) page. If you have any questions about costs, Open Access requirements, or our legal forms, please contact ASJournals@springernature.com.

In recognition of the time and expertise our reviewers provide to Nature Immunology's editorial process, we would like to formally acknowledge their contribution to the external peer review of your manuscript entitled "A SIRT7-dependent acetylation switch regulates early B-cell differentiation and lineage commitment through PAX5". For those reviewers who give their assent, we will be publishing their names alongside the published article.

When you are ready to submit your revised manuscript, please use the URL below to submit the revised version:

Link Redacted

We hope to receive your revised manuscript in 4-5 days, by 21st Aug 2024. Please let us know if circumstances will delay submission beyond this time. If you have any questions please do not hesitate to contact me.

Sincerely,

Ioana Staicu, Ph.D.
Senior Editor
Nature Immunology

Tel: 212-726-9207
Fax: 212-696-9752
www.nature.com/ni

Reviewer #1 (Remarks to the Author):

Appreciate the efforts that the Authors have put into the revision, which has made the story more complete and compelling. Congratulations on your very interesting findings!

Reviewer #2 (Remarks to the Author):

I congrats the authors on excellent responses to the reviewer's comments and on the important findings reported in this paper. I thinking this manuscript should be accepted.

Reviewer #3 (Remarks to the Author):

The authors have improved the manuscript, through identification of PCAF that catalyzes Pax5 acetylation in addressing reviewer 1's questions. The authors also partly address this reviewer and Reviewer 2's concerns by rescue of Sirt deficiency with Pax5 K198R.

With regard to this reviewer's second point about the role of Sirt7 on the epigenome, the authors shared their unpublished data on this subject. The reviewer can accept the authors' argument and respect their wishes to publish that part in an independent manuscript.

In the first major point, the suggestion was to evaluate the requirement for both acetylated and unacetylated forms of Pax5 in rectifying Pax5 deficiency in B cell development. The authors cited technical difficulties in performing this study, especially to express K198Q and K198R mutants at 1:1 ratio. Since Pax5 is a relatively small protein (<50kDa), it is possible to express these two proteins in one retroviral/lentiviral construct through 2A peptide linkage at reasonable high levels. The authors showed rescue of Sirt7 deficiency by overexpressing Pax5 K198R, but this addressed a different question. The authors also provided some indirect evidence to this point, but conclusive supporting evidence is still missing.

Version 2:

Decision Letter:

Our ref: NI-A36869B

26th Aug 2024

Dear Dr. Vaquero,

Thank you for submitting your revised manuscript "A SIRT7-dependent acetylation switch regulates early B-cell differentiation and lineage commitment through PAX5" (NI-A36869B). We are happy to inform you that if you revise your manuscript appropriately according to our editorial requirements, your manuscript should be publishable in Nature Immunology.

I will now pre-edit the current version of your paper. We will also perform detailed checks on your paper and will send you a checklist detailing our editorial and formatting requirements in about two weeks. Please do not upload the final materials and make any revisions until you receive this additional information from us.

If you had not uploaded a Word file for the current version of the manuscript, we will need one before beginning the editing process; please email that to immunology@us.nature.com at your earliest convenience.

In the meantime however, please deposit all omic and code data into public repositories so that the accession codes are readily available to be added in the revised manuscript. We cannot accept the paper without the codes. In addition, please check that the ORCID of all corresponding authors is linked to their Nature account, as this frequently causes delays at acceptance. Should you have any query or comments about ORCID, please do not hesitate to contact our editorial assistant at immunology@us.nature.com.

Thank you again for your interest in Nature Immunology. Please do not hesitate to contact me if you have any questions.

Sincerely,

Ioana Staicu, Ph.D.
Senior Editor
Nature Immunology

Tel: 212-726-9207
Fax: 212-696-9752
www.nature.com/ni

Version 3:

Decision Letter:

In reply please quote: NI-A36869C

Dear Dr. Vaquero,

I am delighted to accept your manuscript entitled "A SIRT7-dependent acetylation switch regulates early B cell differentiation and lineage commitment through Pax5" for publication in an upcoming issue of Nature Immunology.

Over the next few weeks, your paper will be copyedited to ensure that it conforms to Nature Immunology style. Once your paper is typeset, you will receive an email with a link to choose the appropriate publishing options for your paper and our Author Services team will be in touch regarding any additional information that may be required.

Please note that *Nature Immunology* is a Transformative Journal (TJ). Authors may publish their research with us through the traditional subscription access route or make their paper immediately open access through payment of an article-processing charge (APC). Authors will not be required to make a final decision about access to their article until it has been accepted. [Find out more about Transformative Journals](https://www.springernature.com/gp/open-research/transformative-journals).

Your paper will be published online soon after we receive your corrections and will appear in print in the next available issue.

Also, if you have any spectacular or outstanding figures or graphics associated with your manuscript - though not necessarily included with your submission - we'd be delighted to consider them as candidates for our cover. Simply send an electronic version (accompanied by a hard copy) to us with a possible cover caption enclosed.

If you have not already done so, we strongly recommend that you upload the step-by-step protocols used in this manuscript to protocols.io. protocols.io is an open online resource that allows researchers to share their detailed experimental know-how. All uploaded protocols are made freely available and are assigned DOIs for ease of citation. Protocols can be linked to any publications in which they are used and will be linked to from your article. You can also establish a dedicated workspace to collect all your lab Protocols. By uploading your Protocols to protocols.io, you are enabling researchers to more readily reproduce or adapt the methodology you use, as well as increasing the visibility of your protocols and papers. Upload your Protocols at <https://protocols.io>. Further information can be found at <https://www.protocols.io/help/publish-articles>.

Please note that we encourage the authors to self-archive their manuscript (the accepted version before copy editing) in their institutional repository, and in their funders' archives, six months after publication. Nature Portfolio recognizes the efforts of funding bodies to increase access of the research they fund, and strongly encourages authors to participate in such efforts. For information about our editorial policy, including license agreement and author copyright, please visit www.nature.com/ni/about/ed_policies/index.html

Sincerely,

Ioana Staicu, Ph.D.
Senior Editor
Nature Immunology

Tel: 212-726-9207
Fax: 212-696-9752
www.nature.com/ni

Click here if you would like to recommend Nature Immunology to your librarian
<http://www.nature.com/subscriptions/recommend.html#forms>

** Visit the Springer Nature Editorial and Publishing website at http://editorial-jobs.springernature.com?utm_source=ejP_NImm_email&utm_medium=ejP_NImm_email&utm_campaign=ejp_NImm for more information about our career opportunities. If you have any questions please click [here](mailto:editorial.publishing.jobs@springernature.com).

Reviewers' Comments Gámez-García et al. :

We thank all three reviewers for their excellent comments and suggestions. The new data and modifications have improved considerably the overall quality of the work and strengthen our claims.

Reviewer #1:

Remarks to the Author:

The Authors showed that SIRT7 is important for B cell lymphopoiesis by deacetylating and stabilizing the B cell lineage transcription factor PAX5. SIRT7 deficient cells have about 50% decrease in B cells. They identified the potential lysine residue at position 198 (K198) and the expression of acetylation mimic can similarly destabilize PAX5. In human B-ALL, similar to PAX5, SIRT7 expression associates with good prognosis. While the results are potentially interesting, detailed mechanisms remained to be identified. For instance, it is unknown: 1) how K198 acetylation promotes PAX5 degradation and how the modification links to ubiquitin ligase and deubiquitinase; 2) the responsible acetyltransferase(s); 3) if SIRT7 has functional significance in B cell immunity.

1. In Fig 1g-j, the splenic B cells decreased by half in the SIRT7-KO. However, the percentages of IgM+ and IgD+ cells were significantly lower. What are the phenotypes of the B cells in the spleen and lymph nodes? For instance, transitional, marginal zone, germinal center, memory, and plasma cells. What are the IgM- IgD- B cells? Are they class-switched or not expressing cell-surface Ig?

We thank the reviewer for pointing this out. First of all, and thanks to this observation, we have realized that there was a mistake in Fig 1j regarding the IgM⁺ and IgD⁺ cells. While in the original manuscript we stated that these were splenic B-cells, these data were in fact derived from bone marrow samples. These were recirculating B cells rather than splenic B cells, which explains why their percentages did not match the percent of total CD19⁺ splenic B cells. To address the reviewer's point, we have now repeated the experiment (n = 4 mice per genotype) and, as requested, broadened the analysis to include a much more comprehensive characterization of the phenotype produced by SIRT7 deficiency in mature B cell subsets. Specifically, we have included the analysis of transitional, marginal zone, germinal center, memory, plasma, and class-switched IgG1⁺ cells (**Fig.1k** and **Extended Data Fig1h-l**, page 5 of the updated manuscript). Overall, we observed a significant reduction in most mature B-cell subsets, which closely resembled the phenotype observed in bone marrow B-cell progenitors. This was true for all the populations we analyzed except for the germinal center (GC) B cells. Unexpectedly, we found that while the number of these cells was similar in *Wt* and *Sirt7*^{-/-} mice, the percentages of all downstream populations (memory B, plasma and class-switched IgG1⁺ cells) were still reduced. A possible explanation for this increase in GC cells is that SIRT7 may also be required for germinal center B-cell differentiation, which would explain the downstream reduction in plasma cells, class-switched and memory B-cells. This strongly suggests that SIRT7 plays different roles in B-cell progenitors and mature B-cells, which is further supported by the new data we have generated to address the point 3 raised by this reviewer.

Although very interesting, we believe that this subject is out of the scope of the current manuscript and should be developed in a further study. Nevertheless, we have mentioned these observations in the new version of the manuscript. Given that the new characterization of

splenic B-cells is much more informative, we have removed the IgM⁺ and IgD⁺ data from the manuscript, but included these data in this document for the reviewer's perusal (**Reply Fig. 1**).

Reply Fig. 1. Numbers of B220⁺IgM⁺IgD⁻ and B220⁺IgM⁻IgD⁺ cells in the spleen of *Wt* and *Sirt7*^{-/-} mice. Data are presented as mean ± SEM and were analyzed by one-tailed *t*-test (n = 4).

2. Does SIRT7 deficiency affect serum antibody isotypes (basal and immunization)? As SIRT7-KO decreased PAX5 protein level, do the mice have accelerated germinal center and plasma cell differentiation?

The hypothesis that *Sirt7*^{-/-} mature B-cells display a favored germinal center and plasma cell differentiation due to reduced PAX5 levels is certainly of great interest. However, in our characterization of mature B-cell subsets, we found no signs of accelerated GC and PC differentiation, as all the populations downstream of the GC, including PC, were reduced in *Sirt7*^{-/-} mice. These observations indicate that SIRT7 may be also required for late B-cell differentiation, rather than opposing it by increasing the levels of PAX5.

To further support this idea, as well as to answer the other question raised in this point, we immunized *Wt* and *Sirt7*^{-/-} mice with the T-cell-dependent antigen hen egg lysozyme, coupled to NPP hapten (NP-HEL). Seven days after immunization, we collected blood from these mice to measure the levels of HEL-specific IgM, IgG1 and IgG3 antibody isotypes in serum. As expected, the levels of all three isotypes were significantly reduced in immunized *Sirt7*^{-/-} mice, which confirmed that SIRT7 is required for proper B-cell immunity (**Extended Data Fig. 1m**, page 5). Together with the characterization of mature B-cell subsets, our data suggest that PAX5 downregulation in *Sirt7*^{-/-} mice does not promote germinal center or plasma cell differentiation.

Regarding the basal antibody isotypes, we have previously published that switching from IgM to IgG1 and IgG3 is impaired in SIRT7 deficient splenic B-cells, likely due to defective end-joining (Vazquez *et al.* (2016), EMBO J). This is in line with our observations here that the numbers of IgM⁺, IgD⁺ and IgG1⁺ are reduced, with our new characterization the phenotypes in mature B-cell subsets and with the defective response to immunization by *Sirt7*^{-/-} mice. Therefore, these data strongly support the functional relevance of SIRT7 in B-cell immunity.

3. Does SIRT7 deficiency peripheral B cells experience lineage instability?

We thank the reviewer for pointing out this interesting issue. This was certainly a very appealing hypothesis to test. To our knowledge, only the conditional *Pax5* knock-out in mature B cells has been described to revert lineage commitment. We have replicated the seminal experiments of Cobaleda *et al.* (Nature, 2007) by sorting Lin⁻IgM⁺IgD⁺ splenic B cells from our *Wt* and *Sirt7*^{-/-} mice and injecting these cells into CD45.1⁺CD45.2⁺ mice (n = 3 mice per genotype). The results, now included in **Extended Data Fig. 5h** (page 15), show no signs of transdifferentiation in mice injected with either *Wt* or with *Sirt7*^{-/-} mature B cells, strongly suggesting that SIRT7 is relevant for the establishment of B cell commitment rather than its maintenance. As discussed in the

previous point, these data also suggest that SIRT7 may play different roles in developing and mature B-cells.

4. Adoptive ProB cells transfer only resulted in very minimal reconstitution and lead to highly variable results (Fig. 1l-n; Fig. 5i-j). Bone marrow chimera and conditional SIRT7 KO should be used to confirm the effect of PAX5 mutants on the development of B cells and lineage skewing with larger sample sizes.

We agree with the reviewer that the degree of reconstitution we achieved in our *in vivo* experiments was limited and that there was some variability. However, we observed very strong and reproducible phenotypes in all the *in vivo* experiments that we performed, and the data clearly supports all our claims. The original manuscript included several transfer experiments that clearly showed (i) the severe block in differentiation in *Sirt7*^{-/-} pro-B cells that could be rescued by reintroduction of *Wt* but not catalytically inactive SIRT7^{H187Y} (Fig. 1m,n), and (ii) that the deacetylated PAX5^{K198R} rescued lineage commitment but not differentiation (Fig. 5k-m). To support these data, we have now included new *in vivo* experiments with *Sirt7*^{-/-} pro-B cells stably expressing *Wt* PAX5 and PAX5^{K198R} (Fig. 5q, page 15), that not only point in the same direction, but also provide new mechanistic information to understand how the SIRT7/PAX5 interplay regulates B-cell development (see point 7). We believe that, taken together, all these observations strongly support the reliability of these experiments.

Regarding bone marrow chimeras, we agree with the reviewer that these are an excellent approach to assess competition between two different phenotypes in a shared environment. However, our laboratory has previously published these competitive bone marrow transplantation experiments, using a 1:1 mixture of CD45.1 *Wt* and CD45.2 *Sirt7*^{-/-} bone marrow-derived cells (Vazquez *et al.* (2016), EMBO J). In these experiments, SIRT7-deficient bone marrow cells showed a reduced ability to repopulate the lymphoid compartment, supporting our conclusions. Furthermore, it was published in 2015 (Mohrin *et al.*, Science) that SIRT7 prevents age-related myeloid skewing in hematopoietic stem cells. We therefore believe that performing bone marrow chimeras with *Sirt7*^{-/-} mice might not be advisable to study B-cell intrinsic defects, as these experiments might also be influenced by HSC-derived effects. Regarding the use of conditional knock-out models, although very useful, it was not feasible to use them at this point due to time constraints. Furthermore, the main information that this system would provide is to understand whether SIRT7 plays a B-cell intrinsic role in development, which is clearly supported by our data in Fig. 1m and 5q. For these reasons, we believe that the considerable investment that establishing a new mouse colony and the associated work would represent in terms of time, effort, and resources was not justified in this case.

5. Previous studies have shown that P300-mediated acetylation on other lysine residues promotes PAX5 activity (He et al., 2011). Given the negative role of K198 acetylation, is K198 acetylation differentially regulated and uncoupled with the acetylation on other lysine residues?

We thank the reviewer for this excellent point. According to our mass spectrometry data, K198 is the only detected residue differentially regulated by SIRT7. In these experiments, we detected 5 different acetylated lysines, all of which were previously identified by He *et al.* Of these lysines, only K198 acetylation was lost upon SIRT7 expression, which demonstrated that among these residues, K198 was the only target of SIRT7 (Fig. 4m,n).

Further supportive evidence is also provided by the identification of the main HAT responsible for K198 acetylation, requested by the reviewer in point 10. As described later in that point, our data (now included in Fig 4s-u and Extended Data Fig.4n-r, pages 11-12), clearly indicate that

PCAF (KAT2B) acetylates PAX5 at K198R and promotes a decrease in PAX5 protein levels through K198 (Fig. 4t,u), in agreement with a key unique role for K198ac in the control of PAX5 stability. Altogether, these observations strongly suggest a unique and crucial role of K198 in the regulation of PAX5 function.

6. In Fig. 2b-e, while it was suggested that PAX5 is anti-proliferative, the Authors should shed light on why SIRT7 KO B cells have fewer proliferating cells when they have lower level of PAX5?

We thank the reviewer for this excellent observation. The anti-proliferative effect of PAX5 in B-cell progenitors was proposed to be due to *Myc* repression by PAX5 (Somasundaram *et al.*, Blood 2021). Importantly, our RNA-Seq data shows indeed that *Myc* is one of the top upregulated genes in *Sirt7*^{-/-} pre-B cells, which is consistent with the downregulation of PAX5 in these cells (Reply Fig. 2).

Reply Fig. 2. RNA-Seq analysis of *Myc* expression in *Wt* and *Sirt7*^{-/-} pre-B cells. Data are shown as mean \pm s.d. (n = 2).

However, as the reviewer points out, we show in the manuscript that SIRT7 promotes pre-B cell proliferation, which clearly indicates that there may be other PAX5-independent functions of SIRT7 in cell cycle control that could explain these observations. In fact, SIRT7 has been shown to have a general ubiquitous role as a promoter of cell proliferation and cell cycle in multiple models (Ford *et al.* (2006), Genes Dev; Iyer-Bierhoff *et al.* (2018), Cell Reports; Kim *et al.* (2013), Hepatology; Chen *et al.* (2017), J Cell Physiol; Zhao *et al.* (2020), Oncol Rep), through the regulation of rDNA transcription, ribosome biogenesis and DNA replication. In any case, we believe that this effect on proliferation does not have any impact in the major claims of the manuscript. In fact, in this new version of the manuscript and as we discuss in the next point, we have found that the reintroduction of PAX5 in *Sirt7*^{-/-} pro-B cells largely rescues the B-cell development arrest shown by these cells, indicating that SIRT7-mediated regulation of the cell cycle is secondary when PAX5 function is rescued.

7. Fig. 3p, Does the re-expression of SIRT7 and PAX5 restore B cell development and cell number?

We thank the reviewer for this excellent point, which has also been raised by the other two reviewers. This experiment is key to supporting our major claim that SIRT7 promotes B cell development through PAX5. In the original version of the manuscript, we had already demonstrated in transplantation experiments that the reintroduction of *Wt* SIRT7, but not catalytically inactive SIRT7-H198Y, rescued B cell development in *Sirt7*^{-/-} pro-B cells, indicating that the deacetylase activity of SIRT7 is required for normal B cell development (Fig. 1n). Following the reviewer's suggestion, we have now performed a similar experiment in which we reintroduced either *Wt* PAX5 or PAX5^{K198R} into *Sirt7*^{-/-} pro-B cells. In agreement with our model, expression of either PAX5 *Wt* or K198R resulted in a rescue of B-cell development and cell

number (**Fig. 5q**, page 15), strongly supporting our conclusion that SIRT7 promotes B cell development through deacetylation and stabilization of PAX5.

8. Does overexpress or knockout SIRT7 affect the protein level and transactivation activity of PAX5K198 mutants?

We thank the reviewer for bringing up this point. Regarding the protein levels, we agree that it could be interesting to explore whether SIRT7 regulates PAX5 stability through other mechanisms besides K198 deacetylation. However, we believe that our data clearly indicate that K198 deacetylation is the primary mechanism by which SIRT7 regulates B-cell development. This is supported by several of our results: first, catalytically inactive SIRT7-H187Y was unable to drive B-cell development, indicating that the catalytic activity of SIRT7 is required for this process (**Fig. 1n**). In addition, our proteomic analyses of PAX5 acetylation showed that SIRT7 specifically targets K198, and our biochemical experiments with PAX5^{K198} mutants closely recapitulated the effects of SIRT7 deficiency on PAX5 stability. Finally, our new *in vivo* data showed that rescuing PAX5 levels by reintroducing either *Wt* PAX5 or PAX5^{K198R} into SIRT7-deficient cells was sufficient to almost completely reverse the differentiation block. Therefore, although there may be other mechanisms of PAX5 regulation by SIRT7, we believe that our data support the notion that K198 deacetylation is the main mechanism by which SIRT7 regulates B-cell development and commitment.

To explore the effect of the K198 mutants on PAX5 transactivation activity, we have performed luciferase experiments using the CD19-LUC construct reported by the Busslinger lab (Dörfler and Busslinger (1996), EMBO J). In these experiments, *Wt* PAX5 and the K198 mutants displayed a very similar effect, indicating that K198 acetylation does not impact the transcriptional activity of PAX5 *per se* (**Reply Fig. 3a**). In the same line, our RNA-Seq data indicated that SIRT7 was not involved in *Cd19* regulation in B-cell progenitors (**Reply Fig. 3b**). Importantly, He *et al.* (2011) performed similar experiments and observed that only K67 and K87/89 were involved in PAX5 transcriptional activity, while K198 and others were found to be dispensable for PAX5 CD19-Luciferase activity and for *Cd19* expression in B cells. This was somewhat expected, since K67 and K87/89 are located in the Paired-box domain of PAX5, whereas K198 is in a putative intrinsically disordered region, so their regulation and functions were likely to be independent.

Reply Fig. 3. a, Luciferase assay in HEK293F cells expressing CD19-Luc together with GFP and either an empty vector or the indicated PAX5 forms. GFP-normalized luciferase signal is shown as mean ± s.d. (n = 3 replicates). Representative of two independent experiments. **b**, RNA-Seq analysis of *Cd19* expression in *Wt* and *Sirt7*^{-/-} pre-B cells. Data are shown as mean ± s.d. (n = 2).

9. In Fig. 4P, PAX5 acetylated mimics has lower protein levels in HEK293F cells. However, the Authors should include control proteins to indicate the transfection efficiency. It will be best if the reporter proteins are linked to PAX5 via 2A peptide to ensure the original ratio between reporter and PAX5 is 1:1.

We have performed a control experiment using our pMSCV-IRES-hCD4 backbone to quantify the levels of PAX5 relative to a hCD4 control protein. In this vector, PAX5 and the hCD4 selection marker are expressed in a single RNA containing an internal ribosome entry site (IRES), which ensures a proper control to monitor equivalent expression of all three PAX5 forms. We have used these vectors to infect *Pax5*^{-/-} pro-B cells and quantified by flow cytometry the levels of PAX5 and hCD4 to determine the PAX5/hCD4 ratio (**Reply Fig. 4**). Although detecting differences in protein expression by flow cytometry is often challenging, we observed reduced protein levels of the PAX5 acetylated mimic and increased levels of the deacetylated one, confirming that differences in the mutants are due to their different stability.

Reply Fig 4. PAX5 MFI relative to hCD4 MFI in *Pax5*^{-/-} pro-B cells expressing the indicated constructs, as determined by flow cytometry. Data are presented as mean ± s.d and were analyzed by one-way ANOVA with Fisher's LSD test (n = 3).

10. In line 220 and Fig. 4m-n, the experiment was done in HEK293F cells, suggesting PAX5 is acetylated by ubiquitous acetyltransferases. With this amenable system, the Authors should be able to identify the responsible acetyltransferases.

As requested by the reviewer, we have been able to identify PCAF as the HAT responsible for K198 acetylation. Although p300 was previously described by He *et al.* as a HAT able to acetylate PAX5, we decided to use an unbiased approach to comprehensively screen for potential PAX5 acetyltransferases (HATs) based on two criteria: First, the candidate should be a PAX5 interactor; and second, since K198 acetylation reduces PAX5 protein stability, the candidate's protein expression should be negatively correlated with PAX5. For that purpose, we compared a list of all mammalian HATs (Hyndmann and Knepper (2017), *Am J Physiol Renal Physiol*) with another list of known PAX5 interactors. In this analysis, we identified four PAX5 interactors with acetyltransferase activity: p300, PCAF (KAT2B), GTF3C4 and NCOA3 (**Extended Data Fig. 4n,o**, pages 11-12). Next, we correlated the protein levels of these candidates with those of PAX5 using publicly available proteomics data from 27 B-ALL patients. Among the four candidates, only PCAF showed a negative significant correlation with PAX5, suggesting that it may be the HAT responsible for K198 acetylation (**Figure 4s** and **Extended Data Figure 4o**). We then co-expressed all four candidates (p300, PCAF, GTF3C4 and NCOA3) with PAX5 in HEK293F cells. Consistent with the proteomics data, PAX5 protein levels were strongly reduced upon co-expression with PCAF, but not with any of the other HATs. Importantly, this effect was lost when we expressed PCAF with the PAX5^{K198R} mutant, indicating that PCAF reduces PAX5 stability through K198 acetylation (**Fig. 4r** and **Extended Data Fig. 4q,r**). Finally, we co-expressed PAX5-

FLAG together with the candidate acetyltransferases, purified PAX5 by anti-FLAG immunoprecipitation and measured global PAX5 acetylation by anti-pan-acetyl-lysine (AcK) immunoblotting. Surprisingly, we could not reproduce the results published by He *et al.*, as p300 did not increase PAX5 acetylation in our analysis. Of note, only PCAF significantly increased PAX5 acetylation, but it had no effect on the K198R mutant (**Figure 4s** and **Extended Data Figure 4s**). Taken together, these data strongly suggest that PCAF specifically acetylates K198 and regulates PAX5 stability, which indicates that it is the major K198 acetyltransferase that counteracts SIRT7-mediated K198 deacetylation.

11. In Fig. 3m, what are the other changes in the proteome? Are there any common features shared between these proteins and PAX5 (e.g., the motif around K198)?

We thank the reviewer for bringing up this issue. A gene ontology analysis of the proteins regulated in *Sirt7*^{-/-} pre-B cells identified several pathways (**Extended Data Fig. 3c**). Among them, we identified the spliceosome, DNA replication, and signal transduction pathways, which is in agreement with previous reports describing a functional association of SIRT7 in these pathways in other tissues or cell types.

As suggested by the reviewer, we have also searched for any common motifs between the proteins regulated by SIRT7. For this analysis, we used as a reference the 21-residue peptide centered at K198 of mPAX5, including 10 residues upstream and downstream. We then generated a 21-mer for each of the lysine residues present in all the differentially regulated proteins from the proteome, resulting in a list of 5378 peptides, and analyzed their homology with the K198 21-mer. Of all these peptides, only 157 show more than 25% homology to the reference, with the top homologous peptide showing a 38.1% similarity to the K198 reference. Finally, we aligned the K198 21-mer with these 157 peptides to search for a common motif (see **Reply Fig. 5**), which yielded a very weak motif, suggesting that SIRT7 does not target a common motif in the proteins that it regulates in B-cell progenitors.

Reply Fig. 5. Motif analysis computed with the lysine-centered 21-mers that presented at least 25% homology to the PAX5^{K198} 21-mer. Only the 21-mers from differentially expressed proteins between *Wt* and *Sirt7*^{-/-} pre-B cells were used.

To highlight some of the top differentially expressed proteins (including Smarcd1, Ezh2, the MLL component Wdr48 or Irf2), in the new version of the manuscript we have now labeled them in the volcano plot (**Fig. 3m**, pages 8-9). Additionally, we have now included a Supplementary Table (**Supplementary Table 2**) containing the list of all significantly regulated proteins.

12. In Fig. 3f, how many of the genes in Cluster 3 actually have PAX5 peaks around the promoters (correlating data from Fig. 5).

To address the reviewer's request, we have now crossed cluster 3 genes with a publicly available list of significant ($q < 0.05$) PAX5 peaks (Hill et al. (2020), Nature), considering gene bodies or only their promoters. The results, which have been included in **Extended Data Fig. 3d** (page 9), show that 43% of the genes in Cluster 3 have significant PAX5 peaks around their promoters, and 62% display PAX5 peaks around the gene (including promoter regions, gene bodies and 3'UTR), supporting that most of the genes in cluster 3 are direct targets of PAX5.

13. Additional proof for the interaction between SIRT7 and PAX5 in the cells (proximity ligation).

Unfortunately, proximity ligation assays cannot be performed with SIRT7 due to technical reasons, as there are no bona-fide antibodies against SIRT7 suitable for immunofluorescence. In fact, we have tested many commercially available antibodies against SIRT7, but none of them have passed our quality controls. Nevertheless, we believe that we have provide strong evidence for the interaction of SIRT7 and PAX5 in the manuscript: First, even though endogenous interactions are often very challenging, we have shown by immunoprecipitation that SIRT7 interacts with endogenous PAX5 in HAFTL pre-B cells. Second, we have also observed a clear interaction between transiently expressed SIRT7 and PAX5 in HEK293F cells, a model that lacks B-lymphoid-specific proteins, again suggesting a direct interaction between both proteins. Third, in size-exclusion experiments in which we separated protein complexes based on their molecular weight, PAX5 and SIRT7 coeluted together in high molecular weight fractions, which suggests that they interact in the context of a large protein complex. This is consistent with several reports demonstrating that PAX5 works in concert with other transcription factors and epigenetic regulators. Finally, we have shown that SIRT7 deacetylates PAX5, both *in vitro* and *in vivo*, which also involves a physical interaction between them. Altogether, we believe that we have provided compelling evidence through four different experimental approaches supporting the interaction between SIRT7 and PAX5.

14. Which lysine(s) on PAX5 got Ub when K198 is Ac?

We thank the reviewer for bringing up this issue. Following the reviewer's suggestion, we have analyzed our mass spectrometry data (**Fig. 4k**) to identify ubiquitinated residues in PAX5. Specifically, we looked for Gly-Gly dimers and for the Leu-Arg-Gly-Gly sequence, the two hallmarks of lysine ubiquitination identification by mass spectrometry. Unfortunately, we could not identify any evidence of ubiquitination in PAX5. We did identify two interesting non-canonical ubiquitinated residues (Thr6 and Ser15) that were lost upon expression of SIRT7, suggesting that these residues may be involved in K198Ac-mediated PAX5 degradation. However, these detected modifications had a low AScore, indicating low confidence. Furthermore, although non-lysine ubiquitination has been reported to participate in ERAD-mediated protein degradation, it is still poorly understood how non-canonical ubiquitination mediates protein degradation. Therefore, the relationship between PAX5 acetylation and ubiquitination, and how it is linked to its degradation seems to be at this point quite complicated and in our opinion is beyond the scope of this study. For these reasons, we believe that it should be the subject of further research in a follow-up study.

Although we have not been successful identifying the residue/s ubiquitinated, we have also screened for the E3 ligase(s) and deubiquitinase(s) (DUBs) responsible for PAX5 degradation and stabilization. We have followed a similar approach to the one used to identify PCAF as a PAX5 acetyltransferase. Specifically, we cross-referenced the list of all the mammalian proteins

harboring E3 or DUB activities with the list of PAX5 interactors, and compared the protein expression levels of each candidate with those of PAX5 in B-ALL samples. PAX5 interactors with E3 or DUB activities were selected as candidates for validation if they correlated negatively with PAX5 in the case of E3 ligases or positively in the case of DUBs.

Reply Fig 6. a, Venn diagram depicting the overlap between PAX5 PPIs and mammalian deubiquitinases (DUBs). PAX5 PPIs integrate the interactors described by Okuyama *et al.* (2019, PLOS Genetics) and those compiled in the BioGRID repository. The list of mammalian DUBs was obtained from <https://esbl.nihbi.nih.gov/Databases/KSBP2/Targets/Lists/DUBs/>. **b**, Scatter plots showing the correlation between the protein levels of PAX5 and the indicated DUBs determined by proteomics in human B-ALL patients samples. Each point corresponds to one sample. Linear regression and 95% confidence intervals (dashed lines) and Spearman's rank correlation coefficient (R^2) and significance are shown ($n = 27$). **c**, Venn diagram depicting the overlap between PAX5 PPIs and mammalian E3 ubiquitin ligases. The list of mammalian E3 ligases was obtained from <https://esbl.nihbi.nih.gov/Databases/KSBP2/Targets/Lists/E3-ligases/>. **d**, Table summarizing the correlation between the protein levels of PAX5 and the indicated E3 ligases. **e**, Densitometric quantification of the relative protein levels of PAX5 in HEK293F cells expressing the indicated E3 ligases. Data are shown as mean \pm s.d. **f,g**, Representative immunoblots (**f**) and quantification (**g**) of the ubiquitination levels of FLAG-tagged PAX5 purified from HEK293F cells expressing the indicated constructs. In **g**, data are presented as in mean \pm s.d.

In the case of the DUBs, we found two PAX5 interactors with deubiquitinase activity: BAP1 and PSMD14. However, the protein levels of both BAP1 and PSMD14 were negatively correlated with those of PAX5, indicating that while they may deubiquitinate PAX5, they are unlikely to be involved in its ubiquitination-mediated proteasomal degradation (**Reply Fig. 6a,b**). Regarding the E3 ligase, we identified three different candidates (MDM4, DTX2 and SCAF11) that were

negatively correlated with PAX5, suggesting that they may target PAX5 for degradation (**Reply Fig. 6c,d**). Of these three candidates, we obtained expression plasmids for MDM4 and DTX2, but we were unable to clone the SCAF11 cDNA, probably due to its very large size and the existence of multiple isoforms. In fact, we could not find any published work using such a construct, so we assumed that ectopic expression of this protein would be problematic. Therefore, we decided to focus our analysis on DTX2 and MDM4. To do this, we co-expressed PAX5, HA-Ubiquitin and each of these E3 ligases and measured PAX5 levels by immunoblotting, and its ubiquitination by anti-FLAG immunoprecipitation followed by anti-HA immunoblotting. We observed that both DTX2 and, to a lesser extent, MDM4 slightly decreased PAX5 levels and increased its ubiquitination (**Reply Fig. 1e-g**). However, these effects were very weak and variable, so we believe that we cannot claim that any of these enzymes is involved in PAX5 degradation. Therefore, we have not included these new data in the updated manuscript. As we discussed above, PAX5 proteasomal degradation appears to be a complex and tightly regulated process. Although interesting, this issue is not at this point relevant for the main conclusions of our work, and therefore we believe that should be the focus of further research in a follow-up study.

15. Does the acetylation status at K198 affect sub-cellular localization of PAX5 (e.g., nuclear vs cytoplasmic)?

This is an excellent question. To address it, we have performed sub-cellular fractionation of HEK293F cells transiently expressing PAX5 *Wt*, K198Q or K198R. Specifically, we have studied the localization profile of the three proteins in the isolated cytoplasm, nucleoplasm and chromatin fractions. The results, included in **Extended Data Fig. 4m** (page 11), showed that none of the PAX5 forms localized to the cytoplasm, whereas we observed reduced levels of K198Q and increased levels of the K198R in the nucleoplasm and chromatin. These results are consistent with our ChIP-Seq data in pro-B cells (**Fig. 5b**) and indicate that K198 acetylation does not have any impact in PAX5 sub-cellular location.

16. In Fig. 4a, can SIRT7 overexpression rescue PAX5 level in SIRT7 KO primary B cells?

Following the reviewer's request, we have performed this experiment and included it in the manuscript as **Fig. 4d** (page 9; quantification in **Extended Data Fig. 4c**). Similar to the B-cell development defect and the de-repression of lineage-inappropriate genes, reintroduction of SIRT7 also rescued the protein levels of PAX5. Together with the new data indicating that reintroduction of PAX5 in *Sirt7*^{-/-} primary B-cell progenitors also rescues B-cell development, our data strongly indicates that PAX5 mediates the main effects of SIRT7 in B-cell development and commitment.

17. In Fig. 4f-g, how does SIRT7 deficiency affects PAX5 stability in mouse B cells?

We have demonstrated the decrease in stability of PAX5 in SIRT7-deficient cells in several experiments. As **Fig. 4a-c** suggest, PAX5 stability is decreased in SIRT7-deficient B-cells. Furthermore, we have analyzed the stability of PAX5 in CHX experiments in both mouse pro-B HAFTL cells (**Fig. 4f,g**) and in the non-B cell line HEK293F (**Fig. 4r**). In both cases we obtained a similar result suggesting that PAX5 degradation is mostly regulated by ubiquitous regulators. In HAFTL, endogenous PAX5 has a half-life of about 10h, which is reduced to 5.5h in the absence of SIRT7. Similarly, in HEK293F, ectopically expressed *Wt* has a strikingly similar half-life of 8.7h, whereas the half-life of the PAX5^{K198Q} mutant is reduced to 3.4h, which closely recapitulates our observations in *Sirt7*^{-/-} HAFTL cells. In primary *Sirt7*^{-/-} pre-B cells (**Fig. 4b** and **4e**), the reduction in the protein levels of PAX5 is also very similar to that of the K198Q mutant in HEK293F (**Fig. 4p**); and the ChIP-Seq signal of this mutant is also very similarly reduced, relative to *Wt* PAX5.

Taken together, we believe that the regulation of PAX5 stability in B-cells is similarly regulated throughout B-cell development

18. Western blot should be quantified. For example, Fig. 4q, the amount of HA-Ub should be normalized to the PAX5 level (especially K198R seems to have less FLAG).

We thank the reviewer for pointing out this issue. Following the reviewer's request, we have now quantified the immunoblots in **Fig. 4p** and **4q** and performed a third replicate for **Fig. 4e** to calculate statistical significance. We have also quantified and analyzed the new **Fig. 4d, Extended Data Fig. 4j** (study of K198A mutant requested by reviewer 3), **Fig. 6d** (displaying the levels of PAX5 in B-ALL cell lines overexpressing SIRT7), and the PCAF-associated analysis (**Fig. 4t,u**; pages 9-11).

19. In line 243, how much of the decreased chromatin binding is due to decreased PAX5 proteins or its intrinsic DNA binding? In another words, does PAX5 K198 Ac differentially regulate PAX5 activity independent of protein stability?

As we already anticipated above, we have performed CD19-Luc experiments with *Wt* PAX5 and the K198 mutants. In these experiments, K198 mutations to Q and R did not affect the ability of PAX5 to induce expression of the luciferase reporter, suggesting that K198 acetylation does not affect PAX5 DNA binding or transcriptional activity *per se*, at least at the *Cd19* promoter region. Consistently, He *et al.* (2011) performed similar luciferase assays with a K198A mutant (an alternative to mimic deacetylated lysines) with the same result. Furthermore, in cell fractionation experiments, we observed similar differences in the protein levels of the mutants in nucleoplasm and chromatin, suggesting that these differences are due to differential protein levels rather than differential DNA binding ability (**Extended Data Fig. 4m**, page 11). Finally, as discussed below, we have analyzed the DNA motif bound by PAX5 in our ChIP-Seq experiments, and we did not find any major differences between all three PAX5 forms (**Extended Data Fig. 5a**, page 13).

20. In Fig. 5b, Does K198Q and K198R affect the DNA motifs that PAX5 binds to? What kind of regions are shown? Are there any differential peaks?

We thank the reviewer for these important questions. In our original PAX5 ChIP-Seq experiments, the relatively limited number of peaks detected made difficult to provide a conclusive answer to these questions. To address this issue and to improve the overall quality of these ChIP-seqs, we successfully repeated these experiments, which have now replaced the previous ones in the manuscript. This new ChIP-Seqs perfectly reproduced the results of the first experiment (see **Fig. 5b**, page 13), but this time we were able to identify around 10,000 significant peaks ($q < 0.05$), which is enough to reliably analyze the differential binding of the mutants and their motifs. As shown in the new **Fig. 5c**, the constitutively deacetylated K198R mutant gained 3,093 (28.2%) new peaks not occupied by *Wt* PAX5, while it lost 1,656 (15.1%). With respect to the K198Q mutant, it lost almost 80% of the peaks occupied by the *Wt* and PAX5^{K198R} forms, in agreement with its decreased genomic occupancy (**Fig. 5b**), and with our new cell fractionation experiments showing the same effect (**Extended Data Fig. 4m**).

To address whether these differences affect the motifs to which the different PAX5 forms bind, we also performed HOMER analysis with each of the three mutants (page 13). Regarding the PAX5 motif, it was largely preserved in K198 acetylation mimics, indicating that K198 does not affect the DNA binding ability of PAX5 (**Extended Data Fig. 5a**). As shown in **Extended Data Fig. 5b**, our motif enrichment analysis also identified other motifs, corresponding to putative binding sites of other transcription factors (such as PU.1, ETV2 or FLI1). Interestingly, we found that

some of these motifs were significantly affected by both the K198Q and K198R mutations, suggesting that K198 acetylation by SIRT7 may affect PAX5 ability to cooperate with some of these transcription factors at specific loci.

Regarding the question of which type of regions are shown, the plot in **Fig. 5b** shows 1000 randomly subsampled significant peaks (peak center \pm 1500 bp, whereas the analysis in **Fig. 5c** was performed using all the significant peaks ($q < 0.05$) that we detected. We have included this information in the figure legend (page 40).

21. The role of SIRT7 and PAX5 acetylation should be tested using normal mouse and human B cells. Does SIRT7 KD decrease PAX5 level in other B cell lines and primary B cells? Similarly, does expression of SIRT7 increase PAX5?

We believe we have already included in the manuscript compelling evidence to demonstrate not only the role of SIRT7 and PAX5 acetylation in mouse B-cells, but also that their interplay is largely conserved in human B-ALL. In addition to the large amount of data already included, in the new version of the manuscript we have added two more experiments directly related to the reviewer's point: First, as replied in point 16 of the reviewer, the new data included in **Fig. 4d** (page 9) demonstrates that the decreased levels of PAX5 in SIRT7-deficient primary B-cell progenitors can be rescued by re-expression of SIRT7. Second, as requested by the reviewer here, we have overexpressed SIRT7 in two different human B-ALL cell lines that express low levels of PAX5 (see **Fig. 6a**). Importantly, in both cell lines, SIRT7 overexpression significantly increased PAX5 protein levels (**Fig. 6d,e**, page 16), further supporting our previous observations and suggesting an exciting new strategy to enhance PAX5 functions in B-ALL.

Reviewer #2:

Remarks to the Author:

This is an outstanding study that identifies a novel mechanism of control over B cell development and helps to understand the dual role of Pax5 in controlling B cell lineage commitment by repressing and activating gene expression. I was impressed by the breadth and depth of the analyses.

I only have one key question that I think needs answering:

The overarching hypothesis of the authors is that the B cell development defect observed in SIRT7 KO cells is directly attributed to its role in Pax5 acetylation. However, Sirtuins are implicated in several different roles, the alteration of which could give rise to the SIRT7 KO phenotype. To support their hypothesis the authors should perform a key experiment which is to rescue the SIRT7 KO phenotype with the deacetylated Pax5 mutant (K198R). This would be a similar experiment to that performed in Figure 5 which introduced Pax5 wt and mutants into Pax5^{-/-} cells. RNAseq analysis as performed in Figure 5 should also be used to understand how lineage commitment was rescued by the K198R mutant at the transcriptome level.

We sincerely thank the reviewer for the very positive assessment of our work. Completing this study has been an enormous collective effort, and we greatly appreciate such encouraging feedback.

We thank the reviewer for this excellent suggestion. We completely agree that a rescue experiment by reintroducing PAX5^{K198R} in *Sirt7*^{-/-} cells should provide strong support to the hypothesis that PAX5 mediates, at least in part, the effects of SIRT7 in B-cell development. Reviewers #1 and #3 also requested a similar experiment with *Wt* PAX5, so we have performed the requested *in vivo* rescue experiment in *Sirt7*^{-/-} cells with both *Wt* PAX5 and PAX5^{K198R}. Importantly, the reintroduction of both PAX5 forms almost completely rescued the B-cell development defect shown by SIRT7 (**Fig. 5q**, page 15 of the updated manuscript), which strongly supports the central hypothesis of this paper. Additionally, as requested, we have also performed RNA-Seq on the same cells used in these transplant experiments. The results showed that PAX5 ectopic expression partially or fully reversed 30% of the genes upregulated in *Sirt7*^{-/-} cells (specially PAX5^{K198R}), while it did not affect the genes downregulated in SIRT7-deficient cells (**Fig. 5r**, see page 16). Importantly, the upregulated genes reverted by PAX5 expression were enriched in lineage-inappropriate genes (such as *Nfatc2*, *Thy1*, *Alox5ap* or *Cyrr1*) (**Extended Data Fig. 5i**). This is in line with our RNA-Seq data in **Figures 3** and **5** indicating that PAX5 mediates the SIRT7-dependent repression of lineage plasticity.

Interestingly, other pieces of evidence generated to address other comments of reviewers #1 and #3, have provided very valuable information to understand the specific mechanisms by which the PAX5^{K198R} (but not PAX5^{K198Q}) rescued lineage commitment in *Pax5*^{-/-} cells. Our new ChIP-Seq analysis of PAX5 mutants, performed to improve our original data, provided additional information on how PAX5^{K198R} rescues commitment (see page 13). Thus, in these experiments we have detected about 10,000 significant PAX5 peaks, which enabled us to perform detailed analyses (see **Fig. 5**). We found that the PAX5 acetylation mimic K198Q lost almost 80% of its genomic binding sites. Conversely, deacetylation of K198 results in the gain of 3,093 (28.2%) new peaks and the loss of 1,656 (15.1%) peaks relative to *Wt* PAX5 (**Fig. 5c**). Gene ontology analysis of the newly acquired peaks revealed that PAX5^{K198R} binds to regions strongly associated with other lineages, such as T-cells and monocytes (**Extended Data Fig. 4e**), consistent with its strong ability to repress these genes and restore lineage commitment. By comparing our dataset

with a previously published PAX5 ChIP-Seq, we found that most of the peaks bound by PAX5^{K198R} (90.6%) have been previously described to be bound by PAX5 (**Extended Data Fig. 5c**). Therefore, these data suggest that PAX5 deacetylation by SIRT7 promotes PAX5 robust binding to regions involved in the expression of lineage-inappropriate genes. This provides a plausible mechanism for how the K198R mutant rescues lineage commitment.

Minor comments:

In line 94 the authors claim that SIRT7 is “specifically” upregulated. However, SIRT2 and to some extent SIRT1 are also upregulated during B cell development. I think the authors should tone this down and just say SIRT7 is upregulated.

Line 188 change “demonstrate” to “suggest” that SIRT7 and Pax5 collaborate.

We thank the reviewer for pointing this out. We have now performed these corrections, as requested (pages 4 and 9, respectively).

Reviewer #3:

Remarks to the Author:

Vaquero and colleagues investigated the Sirt subfamily of histone deacetylase in B cell development and identified Sirt7 as a critical regulator that promotes pro to preB maturation, in a cell intrinsic, and deacetylase enzyme activity-dependent manner. The authors further showed that Sirt7 is critical for suppressing T cell potential in proB cells and promoting cell proliferation, with the former mediated by deacetylation and stabilization of Pax5. Through mass spec studies, the authors identified K198 in Pax5 as a major target for deacetylation by Sirt7. Pax5-K198Q mutant (simulating acetylated form) destabilized Pax5 and failed to suppress T lineage potential while Pax5-K198R mutant (resistant to acetylation) showed an opposite effect. The reverse correlation between Sirt7 and Pax5 expression was also observed in B-ALL cells. These studies are mostly well designed, and the findings are of substantial interest, and the Sirt7-Pax5 regulatory axis via post-translational modification is a novel advance. However, the scope of the study in the current form is somehow limited. The authors may gain broadened insights into the regulatory activities of Sirt7 by addressing the following questions:

1. Pax5 protein pool. In Sirt7KO cells, Pax5 protein was reduced but substantial amount was retained (Fig. 4b, d, and 4f after CHX treatment). The authors also made an observation that Pax5-K198R mutant failed to rescue B cell development in Pax5KO mice, in spite of its ability to activate and suppress Pax5 target genes. Taken these together, one interpretation is that in a B cell, there should be a pool of Pax5 protein that coexist in the K198 acetylated form and unacetylated form, and both forms may have balanced acts, leading to B cell maturation. This hypothesis is worth testing, but forced expression of both the K198Q and K198R mutants at the 1:1 ratio (or various ratios).

We thank the review for this suggestion. As pointed out, neither K198R nor K198Q were able to rescue the B-cell developmental arrest exhibited by PAX5-deficient cells, suggesting that both K198 acetylation and deacetylation are required for B-cell differentiation. Unfortunately, we have been unable to perform successfully the requested *in vivo* transplantation experiments of reintroducing simultaneously both K198Q and K198R mutants in the same cells at 1:1 ratio, due to several technical problems. The most relevant of them was the impossibility of controlling reliably simultaneous 1:1 *in vivo* expression levels of both mutants, which made very difficult to draw conclusions from these experiments. However, we did perform another critical *in vivo* experiment that was requested by all three reviewers, which we believe provides important insight into this issue. We reintroduced either *Wt* PAX5 or the PAX5^{K198R} mimic into *Sirt7*^{-/-} B-cell progenitors and performed *in vivo* transplantations with these cells. Notably, both forms of PAX5 similarly rescued B-cell development to levels almost identical to those of *Wt* cells (Fig. 5q, see page 15 of the updated manuscript). This strongly supports the idea that SIRT7 promotes B-cell development through the deacetylation of PAX5-K198. The fact that in SIRT7-deficient cells, in which PAX5 should be hyperacetylated at K198, overexpression of K198R can rescue B-cell development, suggests that it has a dominant effect over the acetylated form.

In the new version of the manuscript, we also include further evidence supporting the functional relevance of dynamic K198 acetylation/deacetylation. As requested by reviewer #2, we also performed RNA-Seq analysis of these cells, which showed that overexpression of K198R rescues 30% of the upregulated genes in SIRT7-deficient cells, but also results in the abnormal expression of many other genes (Fig. 5r, Clusters B-C), such as genes related with IL2-, IL-4-, or Wnt signaling (Extended Data Fig. 5i, see page 16). This suggests that, although the overexpression of K198R rescues B-cell differentiation in *Sirt7*^{-/-} mice, a proper balance between

acetylated and deacetylated PAX5 is required for the optimal regulation of B-cell transcriptional programs. The relevance of K198ac in B-cell development is also supported by additional evidence included in the revised version of the manuscript. As requested by reviewer #1, we have identified the acetyltransferase (HAT) responsible for K198 acetylation. A combination of computational approaches and experimental validation led us to identify PCAF (or KAT2B) as able to significantly increase the global acetylation of PAX5 and strongly reduce its protein levels (**Fig. 4s-u** and **Extended Data Fig. 4n-r**, see pages 11-12). Of note, these effects were completely abrogated by the PAX5^{K198R} mutation, suggesting that PCAF specifically destabilizes PAX5 by acetylating K198. This is particularly interesting considering a recent report showing that the conditional deletion of PCAF, and its closely related homolog GCN5, in B-cell progenitors impairs B-cell development at the pro-B cell stage, similar to the effects produced by SIRT7 deficiency (Oksenysh *et al.* (2021), *Biomolecules*). Thus, the requirement of PCAF for B-cell development, together with our observation that it reduces the stability of PAX5 by opposing SIRT7-mediated deacetylation of K198, offers compelling evidence that acetylated PAX5 is also required for proper B-cell maturation.

Altogether these observations support that the dynamic regulation of K198 acetylation/deacetylation results in an optimal and balance expression of key signaling genes during the process of B-cell development. Considering these issues we have now expanded the discussion of this subject in the manuscript (pages 15-16 and 19).

The use of Arginine (R) to replace in place of K is not an ideal choice, because R is subject to may modification itself as well (<https://pubmed.ncbi.nlm.nih.gov/18603028/>). Pax5-K198R mutant might gain some new functions beside becoming resistant to deacetylation by Sirt7. This is based on the data in Fig. 5d, where Pax5-K198R appeared to a super-Pax5, that over-repressed cluster 1 genes, and excessively induced cluster2 genes. Testing a more neutral mutation such as Pax5-K198A, alone or in combination with Pax5-K198Q could provide more insights.

We thank the reviewer for this point and agree that PAX5-K198R behaves as a hyperactive form of PAX5 in terms of its ability regulate gene expression. We believe that this is most likely because PAX5-K198R cannot be acetylated, which explains its stronger effect compared to the dynamically regulated *Wt* form. In these experiments, we mutated K to R because this is the gold standard to mimic lysine acetylation with a minimal structural impact. However, as the reviewer mentions, we cannot discard completely the possibility that the mutation to R could be altered by other arginine-dependent modifications. Following the reviewer's suggestion, we mutated K198 to Alanine and tested the protein levels and the stability of this mutant. Overall, PAX5-K198A behaved as PAX5-K198R, with both displaying increased protein levels and stability compared to *Wt* PAX5 (**Extended Data Fig. 4 j-l**, page 11 of the reviewed manuscript). These results strongly suggest that the improved functions of the deacetylated PAX5 mimics are due to the increased stability of the deacetylated form, rather than to any potential additional arginine modification.

Because Pax5-K198R is more stable, the absolute protein amount/cell was likely higher than WT. The ChIP-seq signal in Fig. 5b should be normalized by the protein amount. In addition to quantitative measurement of peak strength, there are potentially qualitative changes, after proper normalization, such as Pax5-K198Q failed to bind specific sites occupied by WT Pax5, and Pax5-K198R may gain new targets, which may account for its unusual in gene activation/repression patterns.

We thank the reviewer for pointing out these key issues. As the reviewer suggested, we normalized the ChIP-seq signal by the protein amounts shown by PAX5 mutants in our western

blots (quantification in **Extended Data Figure 4h**, page 11). To do so, we used the “Scaling factor” tool provided by the ChIPseqSpikelnFree software. Due to the nature of this mathematical approach, which cannot distinguish specific signals from background noise, the baseline was also affected by this normalization. Nevertheless, the peak profile intensity of all three PAX5 forms was nearly identical after normalization (**Reply Fig. 7**). This strongly suggests that the differential stability of the mutants accounts for their differential strength binding. However, to our knowledge, such an approach has never been used before, and we believe that the reliability of scaling ChIP-Seq signals based on the protein amount detected by western is very limited. Therefore, we have decided to exclude these data from the updated manuscript and have performed further analysis to address this issue, as discussed below.

Reply Fig. 7. ChIP-Seq analysis of the indicated PAX5 forms normalized using as a scaling factor the densitometric quantification of the levels of PAX5 mutants in **Fig. 4p**.

Regarding the analysis of qualitative differences in binding by PAX5 mutants, this was an important question that we were unable to address in the initial version of the manuscript, because our original ChIP-Seq experiments detected only a very limited number of significant peaks. To solve this problem, we repeated the ChIP-Seq experiments, and included the new data in the manuscript (see page 13). Importantly, the new ChIP experiments closely reproduced the results of the first one (see **Fig. 6b**), but this time we detected approximately 10,000 significant ($q < 0.05$) PAX5 peaks. This new dataset allowed us to analyze the differential binding of the mutants in detail. As we show in **Fig 5c**, the K198R mutant acquired 3,093 (28.2%) new peaks. Unexpectedly, it also lost binding to 1,656 (15.1%) peaks occupied by *Wt* PAX5. To further explore whether PAX5^{K198R} bound to new regions or enhanced the binding to canonical PAX5 regions, we compared the peaks bound by PAX5^{K198R} with a publicly available list of PAX5 peaks (Hill *et al.* (2020), Nature). We found that most of the significant PAX5^{K198R} peaks (90.6%) overlapped with previously reported peaks (**Extended Data Fig. 5c**). This strongly suggests that, rather than causing PAX5 redistribution, K198 deacetylation by SIRT7 promotes robust PAX5 binding to regions where it binds weakly. However, the fact that PAX5^{K198R} lost 15% of the peaks occupied by *Wt* PAX5 suggests that K198 acetylation also regulates PAX5 target specificity to some extent.

Fig. 1l, m, n, the authors used mutant Sirt7 to rescue Sirt7 deficiency. Testing Pax5 mutants would be fruitful efforts as well to see how Pax5-K198R (or Pax5-K198A) in a different cell context. This experiment needs an additional control, WT EV.

This is an excellent suggestion. We performed this experiment using *Wt* EV cells or *Sirt7*^{-/-} expressing either an EV, *Wt* PAX5 or PAX5^{K198R}. As discussed above, the results provided strong evidence supporting the hypothesis that SIRT7 promotes B-cell development through the deacetylation of PAX5 (Fig 5l).

2. Sirt7 activity. Sirt7 deficiency had profound impact on B cell development and target gene expression. These had share feature with Pax5 deficiency, but they are clearly not phenocopy of each other. The authors need to clearly document the distinct features between the two, besides their similarity. The DEGs due to Sirt7 deficiency should be compared with Pax5-activated or -repressed genes to examine what portion of Sirt7 target regulation is ascribed to Pax5.

We fully agree with the reviewer that in the original manuscript we only emphasized the common features of the transcriptomes of PAX5 and SIRT7 deficiencies. We have now reported some of the differences in the updated manuscript (see Fig. 3 and page 9). Regarding the proportion of the genes regulated by SIRT7 that depend on PAX5, the new RNA-Seq of the rescue experiment reintroducing *Wt* PAX5 or PAX5^{K198R} into *Sirt7*^{-/-} cells provides key evidence to reply to this issue. In this experiment, PAX5 rescued approximately 30% of the genes that were upregulated in SIRT7-deficient cells, but it did not affect the downregulated genes (Fig. 5r, page 16). Interestingly, as mentioned earlier, PAX5^{K198R} appeared to enhance the upregulation of Cluster B genes in SIRT7-deficient cells, which may have been an indirect effect of PAX5 expression. However, it did not appear to impede PAX5^{K198R} from rescuing B-cell development in *Sirt7*^{-/-} cells, suggesting that the repression of Cluster B genes may be dispensable for B-cell development.

Sirt7 is a histone deacetylase. Yet its impact on the histone modification in proB cells, on overall level or specific gene loci was not addressed at all. The singular focus on deacetylation of a non-histone protein Pax5 is a sharp focus but appeared to be narrow. The direct impact of Sirt7 on B cell epigenetic regulation, if biologically meaning, should not be ignored in a comprehensive study.

This is a crucial point. Indeed, we are currently working on a parallel story to describe the impact of SIRT7 in B-cell epigenetics. We have compelling data on this topic and would like to confidentially share some of our findings with the reviewer, but we respectfully believe that addressing the epigenetic regulation by SIRT7 in B-cell progenitors is at this point out of the focus of this paper.

3. Data presentation. The authors performed impressively extensive biochemical and genetic studies, yet the data description is generally very sketchy, without sufficient explanation. For a paper to appeal to non-experts and trainees in the field, some better description is needed. Some examples for improvement include:

We thank the reviewer for this helpful suggestion. Due to character limitations, we tried to be very compact when writing the manuscript, without sacrificing its clarity. However, we fully agree that a more detailed explanation of some points would make it easier for non-experts to understand this work. We have therefore improved the manuscript following the suggestions below.

1) Sirt7 KO, Cre or germline? No description in any section besides references. Two strains were mentioned, which one was used in this study, or both?

Both SIRT7-deficient mouse strains used in this paper are germline *Sirt7* knockouts. In the paper, the experiment included in Extended Data Fig. 1f,g was the only one performed with the

C57BL/6 *Sirt7*^{-/-} strain. In fact, this experiment aimed to confirm that defective B-cell development was not exclusive to the 129Sv strain that we normally use in the laboratory. All other experiments with *Sirt7*^{-/-} mice were performed with the 129Sv strain.

We have stated in the Methods section, page 21: “ Germline *Sirt7*^{-/-} mice²⁵ and *IgHEL*³⁰ mice (CD45.2) were on the 129Sv background unless otherwise specified, while *Pax5*^{-/-} mice⁵ and CD45.2 mice were on the C57BL/6 background. Germline C57BL/6 *Sirt7*^{-/-} mice were described previously²⁴.” We have also indicated in **Extended Data Fig. 1f,g** and its figure legend that it was performed with the C57BL/6 strain (Supporting information pages 2-3)). However, to avoid redundancy, this information has not been included in all other experiments.

2) Figure 3e, mark numbers of differentially expressed genes, criteria for DEGs.

We have included the number of DEGs in the Volcano plots in **Fig. 3e**. We have also included the criteria for DEGs in the figure legend (page 37).

3) What genes are used for Figure 3f, clustering analysis?

Fig. 3f contains the top ≈2000 DEGs ($p < 0.03$) among all four conditions. We have now clarified the significance threshold in the figure legend (page 37). We used this criterion because in this RNA-Seq experiment contained two variables: genotype and cell type. While the genotype significantly regulated some hundreds of genes, thousands of genes were regulated from pro-B to pre-B cells. Therefore, we decided to represent a middle-term number of genes to get a representative vision of the effect of the genotype on each cell type and in the pro-B-to-pre-B cell transition.

3.1) Direct comparison of WT and KO in pro and pre-B cells was not documented well.

In the original manuscript, we focused on the gene ontology and GSEA terms regulated by SIRT7 in pre-B cells, as well as on how SIRT7 regulated the pro-B-to-pre-B cell transition. We have now expanded this point to document better the individual effect of SIRT7 deficiency in pro-B and pre-B cells (page 7). We have also included Supplementary Tables with the DEGs of all the RNA-Seq experiments, as well as the proteome.

3.2) Cluster 2, specifically induced by Sirt7 in pre-B, it does not seem to be the case on heatmap.

We thank the reviewer for pointing this out. Cluster 2 genes are actually downregulated in *Sirt7*^{-/-} pre-B cells, with a minor effect in pro-B, so we have toned down the way we refer to these genes and now mention Cluster 2 as genes downregulated in *Sirt7*^{-/-} in pre-B cells (page 8).

4) Figure 3o, motif analysis on what? DEG promoters?

We have included this information in the figure legend (page 37). Motif enrichment was performed on the promoter regions (TSS -2000 bp upstream) of Cluster 3 DEGs.

5) Fig. 4e, only 2 data points, no Standard error or standard deviation can be deduced.

We have now performed a third replicate and calculated the statistical significance of the data (page 10).

6) Fig. 4h, what is the immunoblot antibody?

These immunoblots were performed with anti-PAX5 and anti-SIRT7 antibodies. We have clarified this in the figure legend to avoid confusion with the anti-FLAG antibody used for the immunoprecipitation (page 38). The references of these antibodies can be found in the Methods section and in the Reporting Summary.

7) Fig. 4i, native PAGE was used? This is essential info to evaluate co-elution

In this case, co-elution information is not derived from the PAGE step itself, but from the previous chromatography. Cells are lysed under native conditions with a high salt concentration, to ensure that only highly stable multi-protein complexes are retained. Native cell lysates are then subjected to gel filtration chromatography. The individual fractions eluted from the chromatography contain size-excluded proteins and complexes, which are finally denatured and analyzed by regular SDS-PAGE. Thus, the purpose of the immunoblot in this case is not to separate protein complexes, but to detect the proteins that were present in each chromatographic fraction. We have explained it in more detail in the Methods section (page 26).

8) Fig. 4l, IP Pax5, followed by IB anti-acetylated lysine?

We have improved the explanation in the updated version of the manuscript (page 10). This is an *in vitro* deacetylation assay using SIRT7 as the enzyme and PAX5 as the substrate. Here, we separately purified PAX5-FLAG and SIRT7-FLAG from HEK293F cells, eluted them in native conditions and incubated them together in the presence of NAD⁺ to test whether SIRT7 deacetylates PAX5 *in vitro*. PAX5 deacetylation was then analyzed by immunoblot with anti-acetylated lysine (AcK).

9) Fig. 4m. more explanation are needed. In the peptide fragment, Kac is the location of K198? How can one tell the removal of 42 dalton acetyl group from the graph?

We have improved the explanation in the updated manuscript (page 11): the cluster of peaks spanning from 555.79 to 555.54 m/z corresponds to the acetylated KacRDEGIQ(+0.98)ESPVPNGHSLPGR PAX5 peptide, where Kac indicates acetylated K198 (as we have now clarified in the figure legend). Therefore, the loss of these peaks indicates that this peptide was not identified in the PAX5 + SIRT7 condition, where we identified the non-acetylated peptide.

10) 5d, what DEGs are used for clustering?

Only significant DEGs with a false discovery rate (FDR)<0.05 were included in the clustering. We have now indicated this info in the figure legend (page 40).

11) How Sirt7^{-/-} HEK293F was made?

As referenced in the Methods section (page 24), *Sirt7^{-/-}* HEK293F was previously generated and described by our lab (Simonet et al., 2020, Sci. Adv).

12) Peak calling conditions. IgG control or Pax5 KO control.

For the peak calling, we used a significance cutoff of $q < 0.05$, as indicated in the updated manuscript (page 32). All the details of the code used for MACS2 peak calling are described in the Methods section: “Peak calling was performed using MACS2⁶⁵ “callpeak” (-f BAM --nomodel --extsize 20 -g mm -B) taking into account both IP (-t) and input (-c) samples for each peak calling”.

These ChIP-Seq experiments did not include neither an IgG control nor a *Pax5*^{-/-} control. However, the antibody we used has been previously validated for ChIP (Okuyama *et al.* (2019), PLOS Genetics). To ensure the specificity of our ChIP-Seq, we compared the peaks of our ChIP with those published by the Buslinger lab (Hill *et al.*, 2020, Nature) using a different antibody. As shown in Reply **Fig. 9**, 92.6% of the peaks detected in our ChIP (Wt PAX5) overlapped with peaks found in the cited paper, strongly supporting the specificity of both antibodies.

Reply Fig. 9. Comparison of the significant PAX5 peaks identified by ChIP-Seq in this study and those identified in a previously published PAX5 ChIP-Seq (Hill *et al.* (2020), Nature).

13) Figure 2f, impaired V to DJ recombination, could be secondary to reduced survival. Decouple the events (by Bcl2 transgene or other means) or acknowledge this possibility.

As shown in Figure 2a, we observed reduced survival specifically in pre-B cells, whereas pro-B cells do not have this defect *in vivo*. Since V-to-DJ recombination occurs in late pro-B cells, and pre-B cells already express a productive pre-BCR, we believe that reduced survival may be secondary to impaired V-to-DJ recombination, rather than being its cause. We have previously published that SIRT7 participates in NHEJ-mediated repair by collaborating with 53BP1 (Vazquez *et al.* (2016), EMBO J), a protein essential for the repair of long-range V-to-DJ breaks (Difilippantonio *et al.* (2008), Nature). Consistent with this, our data in **Fig. 2f** suggest that SIRT7 is specifically involved in long-range V-to-DJ recombination, probably via 53BP1-mediated repair. However, when we crossed our *Sirt7*^{-/-} mice with mice carrying an IgHEL transgene, this failed to rescue the B-cell developmental arrest induced by SIRT7 deficiency (**Fig. 2j**), which actually decouples survival from defective V-to-DJ recombination. If reduced survival was secondary to impaired V(D)J, the IgHEL transgene would have expected to rescue the B-cell developmental defect. In summary, our data clearly suggests that while defective V-to-DJ may have multiple underlying causes, it does not seem to be a major mechanism driving SIRT7-mediated B cell development.

Point-by-point response to Reviewers' Comments - Gámez-García et al.

We thank reviewers for their positive feedback and constructive criticism, which has helped us deepen into the interplay between SIRT7 and PAX5. The points raised by the three reviewers have led us to successfully complete a large number of supporting experiments that we believe have strengthened all of our major conclusions. We are very pleased that all the reviewers recognize the considerable effort we put into addressing the issues raised. We are especially happy that Reviewers 1 and 2 found the revised version of the manuscript suitable for publication in Nature Immunology.

Reviewer #1 (Remarks to the Author):

Appreciate the efforts that the Authors have put into the revision, which has made the story more complete and compelling. Congratulations on your very interesting findings!

Reviewer #2 (Remarks to the Author):

I congrats the authors on excellent responses to the reviewer's comments and on the important findings reported in this paper. I thinking this manuscript should be accepted.

Reviewer #3 (Remarks to the Author):

The authors have improved the manuscript, through identification of PCAF that catalyzes Pax5 acetylation in addressing reviewer 1's questions. The authors also partly address this reviewer and Reviewer 2's concerns by rescue of Sirt deficiency with Pax5 K198R.

We thank the reviewer for the positive assessment of the new version of the manuscript. Even though there is always one more experiment that can be done, we do believe that the new data presented in the manuscript are conclusive and provide solid support for our conclusions.

With regard to this reviewer's second point about the role of Sirt7 on the epigenome, the authors shared their unpublished data on this subject. The reviewer can accept the authors' argument and respect their wishes to publish that part in an independent manuscript.

We appreciate the reviewer's understanding on accepting to leave out results regarding the epigenetic roles of SIRT7 in B cell development. These findings will undoubtedly lead to novel insights in the near future in the form of another manuscript.

In the first major point, the suggestion was to evaluate the requirement for both acetylated and unacetylated forms of Pax5 in rectifying Pax5 deficiency in B cell development. The authors cited technical difficulties in performing this study, especially to express K198Q and K198R mutants at 1:1 ratio. Since Pax5 is a relatively small protein (<50kDa), it is possible to express these two proteins in one retroviral/lentiviral construct through 2A peptide linkage at reasonable high levels. The authors showed rescue of Sirt7 deficiency by overexpressing Pax5 K198R, but this addressed a different question. The authors also provided some indirect evidence to this point, but conclusive supporting evidence is still missing.

We thank the reviewer for pointing out this issue. In the manuscript, we have shown that PAX5 mediates the main effects of SIRT7 in B-cell development, and we describe how the acetylation and deacetylation dynamics of PAX5 regulate its functions. We agree with the reviewer that it is possible that a pool of acetylated and deacetylated PAX5 is required for optimal B-cell differentiation. However, we respectfully disagree regarding the 1:1 experiment.

As the reviewer discusses, several open questions remain regarding PAX5 acetylation dynamics. For instance, whether PAX5 acetylation is differentially regulated at different progenitor stages, and whether acetylated and non-acetylated PAX5 co-exist at the same time, as suggested by the reviewer. We believe that these are very compelling questions that will eventually enhance our understanding of PAX5 regulation. Considering that K198R can prevent lineage conversion but neither K198R nor K198Q can rescue B-cell development defects in PAX5 KO experiments, we suggest that during early B-cell differentiation PAX5 is present in two different sub-populations: First, a subpopulation of PAX5 that remains hypoacetylated throughout the differentiation stages, and that is responsible for repressing the expression of inappropriate genes (mainly T-cell and NK lineages). The second one, a pool of dynamically acetylated/deacetylated PAX5 to allow fine-tuning of B-cell differentiation.

However, we believe that these questions cannot be addressed through this 1:1 ratio experiment, as this assay has severe limitations and is neither as straightforward nor obvious as suggested. Thus, we have already demonstrated in several experiments throughout the manuscript that PAX5 can be expressed at reasonably high levels, and we believe that using a 2A peptide approach is an excellent suggestion to ensure the initial 1:1 expression of both proteins. This was indeed one of the options we considered to address the reviewer's point. However, the main challenge we encountered is that the PAX5-K198R and K198Q mutants have markedly different half-lives (14.4h and 3.4h, respectively). Therefore, even using a 2A construct, the steady-state levels of the two proteins would differ significantly. In this case, the conclusions drawn from an experiment where we cannot control the protein levels would be very limited, especially when our readout is the repopulation or not of the B-cell lineage.

Although we agree that this point is important to understand the regulation of PAX5 dynamics throughout B-cell development, it is not central to the major conclusions of the work. Therefore, we believe that the considerable amount of additional work, effort and resources that would require to reply appropriately this issue is, at this point, not justified. However, to acknowledge the reviewer's point in more detail, we have now included in the text the possibility of this "pool" connecting it with the data supporting the relevance of both acetylated and deacetylated K198 in B-cell differentiation. This data has been included in the discussion section (page 20, lines 482-503) and we believe that it significantly helps to integrate our evidence, and the possible explanations discussed above.